



# Correction of temperature and relative humidity biases in ERA5 by bivariate quantile mapping: Implications for contrail classification.

Kevin Wolf[1], Nicolas Bellouin[1,2], Olivier Boucher[1], Susanne Rohs[3], and Yun Li[3]

[1]Institut Pierre-Simon Laplace, Sorbonne Université / CNRS, Paris, France
[2]Department of Meteorology, University of Reading, Reading, United Kingdom
[3]Institute of Energy and Climate Research – Troposphere (IEK-8), Forschungszentrum Jülich, Jülich, Germany

**Correspondence:** Kevin Wolf (kevin.wolf@ipsl.fr)

**Abstract.**

The skill of the atmospheric reanalysis ERA5 from the European Centre for Medium-Range Weather Forecasts (ECMWF) at simulating upper atmospheric temperature and relative humidity is assessed by using five years of In-service Aircraft for a Global Observing System (IAGOS) observations. IAGOS flight trajectories are used to extract co-located meteorological conditions - temperature, relative humidity, and wind speed - and are compared with the IAGOS measurements. This assessment is particularly relevant to the study of contrail formation, so focuses on the highly frequented air space that spans the Eastern United States over the North Atlantic and to central Europe. The comparison is performed in terms of mean, median, probability density functions, and a confusion matrix. For temperature a good agreement is identified with a maximum bias of −0.4 K at the 200 hPa level. Larger biases are found for relative humidity with up to −5.5 % at the 250 hPa level. To remove the systematic biases, which mostly tend towards too dry and cold, a bias correction method, based on a multivariate quantile technique, is proposed and applied. After the correction the bias in temperature is reduced to below 0.1 K and in relative humidity to below −1.5 %. To estimate the representation of contrail occurrence in ERA5, data points from IAGOS as well as corrected and uncorrected data points from ERA5 of temperature and relative humidity are flagged for contrail formation using the Schmidt-Appleman–criterion. In the IAGOS data set 39.2 % and 16.9 % of the samples represent conditions for non-persistent contrails and persistent contrails, respectively. The corresponding numbers for original ERA5 analyses are 40.8 and 17.5 %, respectively, indicating good agreement overall. Applying a proposed quantile mapping correction method and removing the biases in temperature and relative humidity has only a small effect on the distributions but leads to an overestimation of non-persistent contrail occurrence (44.0 %) and underestimation of persistent contrails (16.8 %). Differences in contrail occurrence that remain after the bias correction are traced back to the underling biases in temperature and relative humidity, indicating that ERA5 is either too dry and warm or cold and moist with largest differences at 250 hPa and decreasing with increasing altitude.





## 1 Introduction

Aviation contributes to global climate warming (Lee et al., 2021). The total contribution by aviation is commonly split into
two parts. One fraction is directly attributable to carbon-dioxide ($CO_2$), and is well quantified. For the year 2018, aviation was
estimated to be responsible for 2.5 to 2.6 % of global $CO_2$ emissions (Friedlingstein et al., 2019; Lee et al., 2021; Boucher
et al., 2021). The other contributing fraction to aviation-induced climate change comes from byproducts resulting from fossil
fuel combustion, like nitrogen oxides ($NO_x$), sulfur dioxide ($SO_2$), and aerosol particles. Furthermore, the combustion of all
fuels, regardless whether they are fossil or synthetic, lead to the emission of water vapor (WV) as long as they contain hydrogen
bonds.

The effects of WC are receiving increasing attention in recent years as the emitted WV in the engine exhaust allows and
triggers the formation of condensation trails, also called contrails (Schumann, 1996; Kärcher, 2018). Optically thin cirrus and
contrails are known to have a net warming effect on the climate (Burkhardt and Kärcher, 2011; Schumann et al., 2015; Lee
et al., 2021). The influence of a perturbation, here contrails, on the Earth's atmosphere and its radiative transfer is quantified
by the radiative forcing (RF). The RF is defined as the difference in the net irradiance at the top of atmosphere under perturbed
and unperturbed conditions. The aviation-induced $CO_2$-related RF is estimated to be around 30 $mWm^{-2}$ (Lee et al., 2021;
Boucher et al., 2021). Contrail RF is estimated to be stronger, at about 60 $mWm^{-2}$ but is subject to much larger uncertainties
(Burkhardt and Kärcher, 2011; Lee et al., 2021).

Contrail formation depends on the ambient conditions, which have to be sufficiently cold and moist. The thresholds of
temperature, below which a contrail forms, and relative humidity above which a contrail can form, are estimated with the
Schmidt–Appleman criterion (SAc, Schmidt, 1941; Appleman, 1953). For a contrail to be persistent, defined as having a
lifetime longer than 10 minutes, the ambient air has to fulfill the SAc and must also be supersaturated with respect to ice.
When these criteria are fulfilled and persistent contrails have formed, they can remain for hours, spread, merge, and increase
the total cirrus cloud cover. Employing climate simulations and analyzing satellite observations, Burkhardt and Kärcher (2011)
and Quaas et al. (2021) estimated an increase in total cloud cover due to contrail formation of 6 to 10 % in the mid-latitudes of
the northern hemisphere, where most of the flights occur.

To lower the climate impact of aviation it is important to reduce $CO_2$ as well as non-$CO_2$ effects. One approach to minimize
contrail radiative forcing is flight re-routing to avoid areas where contrails are likely to form and persist. Therefore, a precise
knowledge of the temporal and spatial occurrence of contrail is required. This can be obtained in four different ways.

The first approach are ground-based observations. For example, Schumann et al. (2013) used a roof-top camera to infer cirrus
properties. However, this approach is limited to a single or few locations. In a second approach, satellite observations provide
a top-down view with the required global coverage but come with some drawbacks (Meyer et al., 2002; Minnis et al., 2013).
Depending on the sensor and the satellite platform, the temporal or spatial resolutions are often insufficient to detect young
contrails with low cloud optical thickness (Kärcher et al., 2009). Furthermore satellite observations, similarly to ground-based
observations, can be compromised by underlying cloud layers between the surface and the cirrus.



In a third approach, contrail occurrence can be assessed by model simulations. However, the assessment strongly relies on the accurate representation of the temperature and humidity fields at high altitudes, as well as that of ice cloud amount and microphysical properties, in the model. Contrail modeling can be done interactively or offline. Interactive contrail models are typically implemented in climate models (e.g. Bickel et al. (2020)) by simulating ice supersaturated regions and calculating contrail cirrus cover based on aircraft emission inventories. Offline contrail models, such as (CoCiP, Schumann, 2012) use meteorological fields to predict contrail formation and evolution to contrail cirrus. A frequent source of meteorological data is ERA5 (Hersbach et al., 2020), a state-of-the-art global modeling system from the European Centre for Medium-Range Weather Forecasts (ECMWF). ERA5 builds on the decade-long improvements of the Integrated Forecasting System (IFS) of ECMWF and replaces its predecessor ERA-interim (Dee et al., 2011). Previous studies showed that the IFS scheme and the associated data assimilation predict well the temperature field, as was verified against radiosonde and satellite observations (Dyroff et al., 2015; Carminati et al., 2019). Slightly less accurate is the prediction and re-analysis of relative humidity, which is generally challenging due to the high temporal and spatial distribution of WV. Specific issues have been identified in the upper troposphere and lower stratosphere, as well as with the general representation of ice supersaturation. For example, Dyroff et al. (2015) and Bland et al. (2021) identified a moist bias in the lower stratosphere that also leads to a cold bias due to an excessive longwave radiative cooling. Contrarily, the upper troposphere is subject to an underestimation of water vapor concentrations and ice supersaturation in ERA-interim and ERA5 (Kunz et al., 2014; Gierens et al., 2020; Schumann et al., 2021).

To mitigate the dry bias in ERA-interim and ERA5 for contrail estimation, studies have applied either multiplication factors (Schumann and Graf, 2013; Schumann et al., 2015) or parameterized corrections (Teoh et al., 2022a). However, these proposed corrections do not consider spatial variations in the bias, particularly at different pressure levels.

In situ measurements are the fourth approach. They directly probe and investigate contrail properties. Dedicated measurement campaigns, for instance Voigt et al. (2017), are rare and lack spatial representation. Fortunately, the IAGOS data set is different in the way that it covers large areas of North America, the North Atlantic, and Europe, which have now been sampled for around two decades including its predecessor Measurement of OZone and water vapour on Airbus In-service airCraft (MOZAIC; Marenco et al., 1998; Petzold et al., 2017).

In this study we propose a correction for ERA5 data that is based on a bivariate quantile mapping (QM), which is a standard method of model bias correction (Cannon et al., 2015; Cannon, 2016, 2018). The QM method allows the removal of biases based on the statistical distributions of an observed and modeled quantity, for example temperature and relative humidity. Here, the QM is trained on 3.5 years of IAGOS observations and collocated ERA5 data of temperature and relative humidity. The QM method is then applied on 5.5 years of ERA5 data and compared with IAGOS. Subsequently, we determine the impact of the correction on the representation of non-persistent and persistent contrails with respect to IAGOS. In case of false classifications the underlying differences in simulated and observed temperature and relative humidity are determined to identify systematic shortcoming in ERA5.

Subsequent to this introduction, Sec. 2 describes the data and methods used in this study. After that the results are presented in Sec. 3 and summarized in Sec. 4. The appendix provides detailed information about the Schmidt–Appleman criterion and a IAGOS data analysis.



## 2 Data and Methods

### 2.1 In-service Aircraft for a Global Observing System

The In-service Aircraft for a Global Observing System (IAGOS; Petzold et al., 2015) is a framework of commercial aircraft that are equipped with a set of sensors for in situ measurements of meteorological conditions, trace gas concentrations, and cloud properties. Since 2015, all aircraft within the IAGOS framework have been equipped with the 'Package 1' (P1) instrument package system that includes a backscatter cloud probe (BCP) to measure the particle number concentration $N_{\text{ice}}$, and a dedicated sensor 'ICH' that measures temperature $T_{\text{P1}}$ and relative humidity $r_{\text{P1}}$. The BCP is a single particle backscattering optical spectrometer to detect cloud particles with sizes between 5 and 75 $\mu$m. Light with 658 nm wavelength is emitted by a light emitting diode and directed through a quartz window to the outside of the aircraft fuselage. The light is focused on a narrow range of 4 cm that represents the target area. Cloud particles within the focus backscatter the radiation to a sensor. The intensity of the radiation is proportional to the size, the refractive index, and the shape of the particles as well as the angle under which the particles were hit by the beam. Directly from these measurement the particles size and the particle number concentration can be derived. More details on the BCP can be found in Beswick et al. (2014).

The ICH package is comprised of a capacitive sensor (Humicap-H, Vaisala, Finland) for measurements of relative humidity (defined over liquid water) and a collocated PT-100 platinum sensor for temperature measurements. Both sensors are mounted within a Model 102 BX housing of Rosemount Inc. (Aerospace Division, USA) to minimize heating from solar radiation and thermodynamic effects. The recorded data is post-processed by the IAGOS consortium to correct the raw data following Helten et al. (1998) and Boulanger et al. (2018, 2020). Hereby an "in-flight calibration method" (IFC) correcting an offset drift during the course of the deployment period is applied (Smit et al., 2008; Petzold et al., 2017).

Post-processed data of $T_{\text{P1}}$ and $r_{\text{P1}}$ are stored for every four seconds. However, the response time $t_{1-1/e}$ of a sensor is an important characteristic as it directly affects the measurements. $t_{1-1/e}$ is commonly defined as time that is required by a sensor to adapt to $1 - \frac{1}{e} = 0.6\bar{3}$ of an abrupt change in the measured quantity. The temperature sensor is characterized by a response time $t_{1-1/e}$ of 4 s and an accuracy of $\pm 0.5$ K . The IAGOS humidity sensor is characterized by an average uncertainty of $\pm 6$ %. Including uncertainties from sensor calibration and data post-processing, the uncertainty ranges between 5 % and up to 10 % and increases with decreasing temperature (Helten et al., 1998). The humidity sensor's response time $t_{1-1/e}$ was determined to be 1 s at 293 K and increases to several minutes at 233 K (Neis et al., 2015). $t_{1-1/e}$ of the relative humidity sensor increases due to reduced molecular diffusion into and out of the sensors polymer substrate. In a first order approximation, the distance between two IAGOS measurements of $T_{\text{P1}}$ and $r_{\text{P1}}$ is 0.96 km at a cruise speed of 240 m s$^{-1}$. However, $t_{1-1/e}$ of the relative humidity sensor averages these measurements over a distance that ranges between 15 km (253 K) and 50 km (233 K) at cruise altitude.

In this study, we use only the IAGOS measurements that fulfill the following criteria :

- IAGOS quality flag of $T_{\text{P1}}$ and $r_{\text{P1}}$ is 'good' and 'limited'

- measurements are located between 30°N and 70°N; 110°W and 30°E



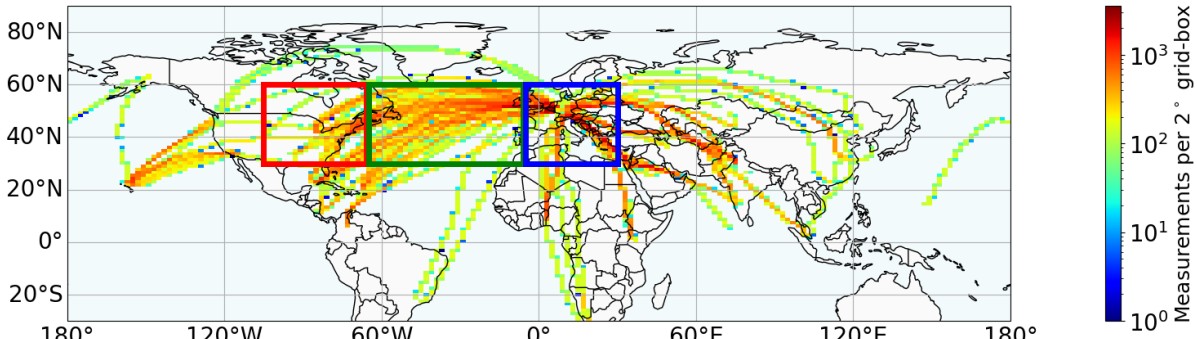

**Figure 1.** Number of measurements per 2°x2° grid-box of analyzed IAGOS measurements for the years 2015 to 2021 (inclusive). The measurements are filtered for data quality and pressure levels. This study uses measurements in the three boxes: United States (US, red), North Atlantic (NA, green), and continental Europe (EU, blue). Longitude coordinates of the bounding boxes are selected to follow Petzold et al. (2020).

**Table 1.** ERA5 pressure levels (in hPa) and pressure ranges used to collocate the IAGOS observations.

| Pressure level (hPa) | Pressure range (hPa) |
| --- | --- |
| 300 | $275.0 \leq p < 325.0$ |
| 250 | $237.5 \leq p < 275.0$ |
| 225 | $212.5 \leq p < 237.5$ |
| 200 | $187.5 \leq p < 212.5$ |
| 175 | $150.0 \leq p < 187.5$ |

 – measurements are between 325 and 150 hPa

– $r_{P1}$ (w.r.t liquid water) is between 0 and 100 %

While IAGOS has been operated for multiple years, the global horizontal and vertical coverage remains heterogeneous. Figure 1 shows a density plot of all IAGOS measurements from the years 2015 to 2021 fulfilling the above criteria. Due to the history of IAGOS and the contributing airlines, the highest measurement density is found across the North Atlantic domain (Fig. 1, green, 65°W–5°W). A slightly reduced density is found over North America (Fig. 1, red, 105°W–65°W) and Europe (Fig. 1,

blue, 5°W–30°E), particularly towards the western and eastern boundaries of the respective boxes. Outside of the boxes, the coverage is lower and, therefore, we focus our analysis on these three domains. These domains also follow the selection from Petzold et al. (2020).

Figure 2a–e shows the total numbers of measurements per pressure level ($p$-level) and the fractions attributable to the three sub-domains, which can also be understood as a proxy for the altitude distribution of commercial air traffic in the North Atlantic

corridor. The largest number of samples (35.3 %) are found at the 200 hPa level (Fig. 2d). Slightly fewer samples are obtained at the 225 hPa level (Fig. 2c) with 32.0 % and at the 250 hPa level (Fig. 2b) with 26.5 %. Contributions from $p$-level 300 hPa





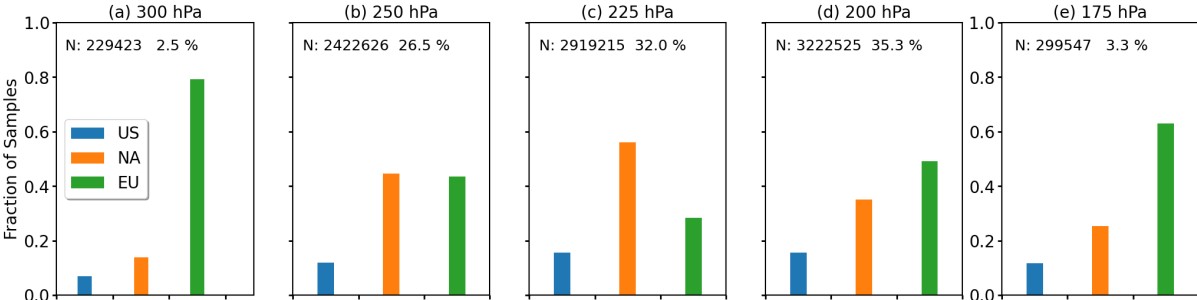

**Figure 2. (a–e)** Fraction of analyzed IAGOS observations by sub-domain separated by pressure level (see Table 1). The total number of samples per pressure level as well as the fraction with respect to the total sample size are indicated at the top.

and 175 hPa are small with 2.5 % and 3.3 %, respectively. Due to the typical flight profiles (for an example see Fig. 10 in Petzold et al. (2015)) the majority of measurements at low $p$-levels (Fig. 2a–b) are sampled over the North Atlantic, where aircraft reach their maximum cruising altitude. For larger $p$-levels, the fraction of samples over the North Atlantic is lower and

measurements from within the EU domain dominate (Fig. 2c–d), corresponding to where the majority of IAGOS-contributing airlines have their main hubs.

The measurement density is a function of longitude, latitude, and $p$-level. In addition, the sampling is biased, i.e., by avoiding severe weather and by avoiding or favoring specific atmospheric circulation patterns, such as the jet stream. The North Atlantic Flight Tracks (routes typically used by aircraft to cross the Atlantic) are selected on a daily basis to avoid (westbound) or take

advantage (eastbound) of the jet stream. This might cause a bias in the sampling of certain atmospheric conditions that might be associated with the jet stream and mid-latitude storm activity (Pasquier et al., 2019).

## 2.2 ERA5

Meteorological data are obtained from the ECMWF Copernicus Climate Data Store (ECMWF CDS, 2021) using output of the High Resolution component (HRES) of ERA5 (Hersbach et al., 2020). The maximal spatial resolution of $0.25° \times 0.25°$ and

maximal temporal resolution of one hour is used. We also use the native vertical resolution with a 50-hPa spacing between 350 and 300 hPa, and a 25-hPa spacing between 300 and 150 hPa. Along-track temperature $T_{\mathrm{ERA}}$, relative humidity $r_{\mathrm{ERA}}$, wind speed $U_{\mathrm{ERA}}$, and cloud fraction $\mathrm{CF}_{\mathrm{ERA}}$ are extracted using the nearest neighbor method, i.e., selecting the ERA5 grid points that are temporally and spatially closest to the IAGOS observations. We prefer to avoid spatial and temporal interpolation as relative humidity is sensitive to the interpolation technique in time and space (Schumann, 2012).

The current version of the ERA5 data set was generated with the ECMWF Integrated Forecasting System (IFS) cycle Cy41r2, which was operational in 2016. Within ERA5 the relative humidity is provided with respect to liquid water or ice depending on $T_{\mathrm{ERA}}$ of the grid box. In general, relative humidity (unitless) is defined as the ratio of the water vapor pressure $e(T)$ to the



saturation water vapor pressure $e_{\text{sat}}(T)$ as:

$$r = \frac{e}{e_{\text{sat}}(T)}. \tag{1}$$

In ERA5, $e_{\text{sat}}(T)$ is given by:

$$e_{\text{sat}}(T) = \alpha \cdot e_{\text{sat,l}}(T) + (1 - \alpha) \cdot e_{\text{sat,i}}(T), \tag{2}$$

with $e_{\text{sat,l}}(T)$ and $e_{\text{sat,i}}(T)$ the saturation water vapor pressure over liquid water and ice, respectively. $e_{\text{sat,l}}(T)$ and $e_{\text{sat,i}}(T)$ are given by:

$$e_{\text{sat}}(T) = a_1 \cdot \exp\left\{ a_3 \cdot \left( \frac{T - T_0}{T - a_4} \right) \right\}, \tag{3}$$

with $a_1$ = 611.21 Pa, $a_3$ = 17.502, and $a_4$ = 32.19 K for liquid water and $a_1$ = 611.21 Pa, $a_3$ = 22.587, and $a_4$ = −0.7 K over ice; and in both cases $T_0$ is 273.16 K (Buck, 1981; Alduchov and Eskridge, 1996; ECMWF, 2020). The scaling factor $\alpha$ in Eq. 2 is a piecewise linear function of temperature $T$ determined by:

$$\alpha = \begin{cases} 0 & \text{for} \quad T < T_{\text{ice}} \\ \frac{T - T_{\text{ice}}}{T_0 - T_{\text{ice}}} & \text{for} \quad T_{\text{ice}} < T < T_0 \\ 1 & \text{for} \quad T_0 \leq T \end{cases} \tag{4}$$

with $T_{\text{ice}}$ = 250.16 K and $T_0$ = 273.16 K. For consistency and comparability with IAGOS all extracted values of relative
humidity are converted to be defined solely over liquid water ($\alpha = 1$) or ice ($\alpha = 0$) and are subsequently referred to as $r_{\text{ERA}}$ and $r_{\text{ERA,ice}}$, respectively. For consistency, IAGOS relative humidity defined over liquid water is labeled with $r_{\text{P1}}$ and defined over ice with $r_{\text{P1,ice}}$.

    The fixed grid resolution of 0.25° of ERA5 does not correspond to a constant longitudinal grid box size in km which depends on the latitude. Considering the three sub-domains between 30°N and 70°N, the spatial resolution of one ERA5 grid-box ranges
between 24 km (30°N) and 14 km (70°N). Therefore, we assume an average grid box size of 19 km. While the IAGOS relative humidity measurements are already smoothed due to the response time of the relative humidity sensor, we additionally smooth the IAGOS measurements by applying a Gaussian filter to estimate the smoothing influence and to compare with ERA5. The standard deviation $\sigma$ of a Gaussian filter is approximated with:

$$\sigma = \frac{k - 1}{6}, \tag{5}$$

where $k$ is the window length. Equation 5 can be regarded as an approximation for a Gaussian distribution as $3 \cdot \sigma$ includes 99.7 % of the Gaussian distribution. With IAGOS measurements every 4 s and assuming an average cruise speed of around 240 m s$^{-1}$, a segment (distance between two measurements) is in the range of 1 km. The average ERA5 resolution of 19 km is matched by setting $\sigma = 3$ to obtain a window length of $k = 19$ (moving average over 19 time-steps).





### 2.2.1 In-cloud representation of supersaturation in ERA5

Previous studies have shown that the upper troposphere is frequently supersaturated with respect to ice under cloud-free (Gierens et al., 1999; Petzold et al., 2020) as well as cloudy conditions (Spichtinger et al., 2004; Dekoutsidis et al., 2023). While ice supersaturation (ISS) in cloud-free conditions is represented in state-of-the-art numerical weather models, they currently lack in the appropriate representation of ISS under cloudy conditions. Often, the ISS is clipped to $r_{\mathrm{ice}} = 100\,\%$, applying the so called "saturation adjustment" (McDonald, 1963). This adjustment is also applied in the IFS ice cloud microphysical

scheme. The adjustment is a necessity of a missing diagnostic variable that would track the time-dependent in-cloud saturation (Tompkins et al., 2007). As a consequence of the adjustment, all available 'excess' water vapor, which is beyond the threshold, is deposited on existing ice particles within one time step, forcing $r_{\mathrm{ice}}$ back to 100 %. While the adjustment approach proved to be suitable for most atmospheric conditions (Gierens et al., 1999; Tompkins et al., 2007; Lamquin et al., 2009), the adjustment results in an underestimation of ISS in the upper troposphere (Gierens et al., 2020), which is problematic for contrail and cirrus

representation.

To compensate for the dry bias in ERA5 for contrail detection applications, $r_{\mathrm{ERA,ice}}$ values are commonly divided by a factor between 0.8 or 0.9 (e.g., Schumann and Graf, 2013; Schumann et al., 2015). An updated scaling-method was proposed by Teoh et al. (2022a, T22 thereafter) that enhances $r_{\mathrm{ERA,ice}} > 100\,\%$ and reduces $r_{\mathrm{ERA,ice}} < 100\,\%$ by a a factor which depends on the original $r_{\mathrm{ERA,ice}}$. The scaling factor is determined by minimizing the Cramér–von–Mises test (Parr and Schucany, 1980)

on collocated ERA5 data and IAGOS flights from the year 2019. Within our study, we use T22-corrected values of $r_{\mathrm{ERA,ice}}$ as a benchmark.

### 2.3 Quantile mapping

In this study we propose to use a quantile mapping (QM) method to remove the lack of ISS in ERA5. QM is a correction method that it is frequently used to correct model biases in comparison to observations in a way that imposes the observed

statistical distribution (Maraun et al., 2010; Maraun, 2012; Cannon et al., 2015; Cannon, 2018). Within our study, the QM technique is applied on ERA5 data and IAGOS measurements, which are regarded as the reference. Subsequently, we provide a brief overview of the mathematical concept of QM for which we follow the notations from Cannon et al. (2015) and Cannon (2018).

The basis of QM algorithms is to consider cumulative distribution functions (CDFs), $F_{\mathrm{o,h}}$ and $F_{\mathrm{m,h}}$, of the observed ($x_{\mathrm{o,h}}$)

and simulated ($x_{\mathrm{m,h}}$) quantity, respectively. The CDFs describe the probability that a certain quantity, for example temperature or relative humidity, exists in the underlying data set. In our case $x_{\mathrm{o,h}}$ are the IAGOS $T_{\mathrm{P1}}$ or $r_{\mathrm{P1}}$ measurements, and $x_{\mathrm{m,h}}$ the corresponding along-track data from ERA5. The subscript 'h' commonly refers to historical data, in our case the reference period from 2018 to 2021. This can also be understood as the training data. Based on the relationship of $F_{\mathrm{o,h}}$ and $F_{\mathrm{m,h}}$, the biased model output $x_{\mathrm{m,p}}(t)$ at any given time $t$ is corrected. The corrected value is represented by $\hat{x}_{\mathrm{m,p}}(t)$ (Cannon et al.,

2015; Cannon, 2018). This is written in mathematical notation as:

$$\hat{x}_{\mathrm{m,p}}(t) = F_{\mathrm{o,h}}^{-1}\{F_{\mathrm{m,h}}[x_{\mathrm{m,p}}(t)]\}. \tag{6}$$





Equation 6 therefore couples a (potentially biased) model output to the most likely value that is observed in reality by the convolution of $F_{\mathrm{m,h}}$ and $F_{\mathrm{o,h}}$. Here we apply the QM technique to the full period, i.e., the inference period, from January 2015 to June 2021, which includes but exceeds the training period.

Equation 6 describes the basic QM bias correction that depends only on one variable. Here, we propose a bivariate QM version for $T_{\mathrm{ERA}}$ and $r_{\mathrm{ERA,ice}}$ as the bias between ERA5 and IAGOS might depend on latitude. Such a multi-variate QM is similar to the suggested versions by Cannon (2016), Cannon (2018), or François et al. (2020).

     For the temperature bias correction, $F_{\mathrm{o,h}}$ and $F_{\mathrm{m,h}}$ are determined at each $p$-level and for two latitude bands. The latitude bands are defined by the outer boundaries of the investigated area with 30°N and 70°N, with the split center point given by the

50 % percentile of the measurements per pressure level. Thus $F_{\mathrm{o,h}}(p,\Phi)$ and $F_{\mathrm{m,h}}(p,\Phi)$ are determined for different classes of pressure $p$ and latitude $\Phi$. $F_{\mathrm{o,h}}(p,\Phi)$ and $F_{\mathrm{m,h}}(p,\Phi)$ spans a temperature range from 190 and 273 K. Similarly, $r_{\mathrm{ice}}$ is corrected with $F_{\mathrm{o,h}}(p,\Phi,T)$ and $F_{\mathrm{m,h}}(p,\Phi,T)$, which are calculated for each $p$-level, two latitude bands $\Phi$, and five temperature bins. As above, $T$ ranges from 190 to 273 K with 5 temperature bins defined by 20 % percentile steps so that each temperature bin contains an equal number of observations at each $p$-level and latitude bin. Consequently, $F_{\mathrm{o,h}}(p,\Phi,T)$ and $F_{\mathrm{m,h}}(p,\Phi,T)$ are

calculated for a total of 80 bins.

     It is important to note that this basic version of QM assumes a time invariant bias between model and observations. We assume the ERA5 data set to be invariant in time as ERA5 is generated with one version of the IFS Cycle 41r2. However, the IAGOS reference observations might vary over time as the spatial distribution of the sampling, instrument calibration, and data post-processing can change. A detailed investigation of the temporal and spatial consistency of ERA5 and IAGOS is provided

in Appendices A and B.

## 2.4   Schmidt–Appleman criterion, potential contrail formation, and contrail persistence

To allow for contrail formation the ambient air must be sufficiently cold and moist. The formation is typically estimated using critical temperature $T_{\mathrm{crit}}$ and relative humidity $r_{\mathrm{crit}}$ thresholds that are derived from the Schmidt–Appleman criterion (SAc, Appleman, 1953). The SAc is based solely on thermodynamic principles and neglects potential effects stemming from the

complex vortex dynamics behind the aircraft. Nevertheless, it proved to be a suitable, first-order approximation. The SAc is a necessary but insufficient criteria for persistent contrails. For contrails to be persistent (lifetime > 10 min), the ambient air must be additionally supersaturated with respect to ice ($r_{\mathrm{ice}}$ > 100 %) in so called ice supersaturated regions (ISSR). However, even under slightly sub-saturated conditions persistent contrails can form. In weakly sub-saturated conditions the dissipation of ice crystals is slowly and, hence, contrails can remain for hours (Li et al., 2023). Within this study, we use the revised version of

the SAc, following Schumann (1996) and Rap et al. (2010). For details on the SAc and equations used to calculate $T_{\mathrm{crit}}$ and $r_{\mathrm{crit}}$ the reader is referred to Wolf et al. (2023).

     The SAc and the requirement for ice supersaturation separate the water-vapor-pressure–temperature diagram (see Fig.2 in Wolf et al. (2023)) in four different areas. The first area represents conditions, where the ambient air fulfills the SAc but is sub-saturated with respect to ice. Contrails that form under these conditions are regarded as non-persistent and are labeled as

NPC. Within the second area the SAc is fulfilled and ambient air is additionally supersaturated with respect to ice and persistent



contrails (PCs) can form. The third area is treated as a special case, in which the ambient air does not fulfill the SAc but is ice supersaturated. Contrails that might have formed under conditions 'R1-NPC' or 'R2-PC' and that are mixed in area 3 may persist and spread. Therefore, area 3 can be understood as a potential 'Reservoir' (R) for contrails (Wolf et al., 2023). The SAc and the ISS threshold are used to flag the IAGOS measurements and the along-track ERA5 for NPC, PC, and R conditions.

Samples that belong to none of these three categories are flagged for no contrail (NoC) formation.

## 3    Results

### 3.1    Distributions of temperature and relative humidity from ERA5 and IAGOS

In a first step, along-track temperature and relative humidity from IAGOS and ERA5 are compared in terms of probability density functions (PDFs), mean values, and mean difference (MD). The performances of the QM-correction and the T22-

correction are further quantified by the root-mean square error (RMSE), the mean absolute error (MAE), the mean square error, and the mean difference (MD). The analysis is limited to $p$-levels 250, 225, and 200 hPa, representing the most frequented $p$-levels (see Fig. 2).

At $p$-levels 200 and 225 hPa, measured $T_{\mathrm{P1}}$ and simulated $T_{\mathrm{ERA}}$ agree well in terms of the MD and the overall shape of the distributions. Only minor deviations in the MD of $-0.4$ K (200 hPa) and $-0.1$ K (225 hPa) are found, with a negative MD

suggesting ERA5 is colder than observed on average ($T_{\mathrm{ERA}} < T_{\mathrm{P1}}$). After the bias correction, the MD is reduced at all $p$-levels to below 0.1 K and also the shape of the PDFs of $T_{\mathrm{ERA}}^{\mathrm{cor}}$ are adjusted to better match the distributions of $T_{\mathrm{P1}}$ (see Fig. 5e).

For all pressure levels, the distributions of $r_{\mathrm{ice}}$ are bimodal although the two modes have different magnitudes. The bimodal shape in the PDFs of upper-air $r_{\mathrm{ice}}$ matches with previous studies, e.g., Ruzmaikin et al. (2014), who used satellite observations from the Atmospheric Infrared Sounder (AIRS). The first mode at low $r_{\mathrm{ice}}$ is caused by dry atmospheric conditions related

to dry air intrusions from the stratosphere into the upper troposphere, e.g., behind frontal zones (Browning, 1997), and flight sections within the lower stratosphere. The second mode at $r_{\mathrm{ice}} = 100\,\%$ is related to regions of high humidity or measurements inside clouds. With the general decrease in $r_{\mathrm{ice}}$ with decreasing $p$, the first mode becomes more and more pronounced, while the second mode flattens and almost vanishes.

Comparing the PDFs of $r_{\mathrm{ERA,ice}}$ and $r_{\mathrm{P1,ice}}$ minor differences are found for the first mode. However, larger differences

appear for the second mode at $r_{\mathrm{ice}} = 100\,\%$, where the occurrence frequency of large $r_{\mathrm{ERA,ice}}$ well exceeds $r_{\mathrm{P1,ice}}$, while $r_{\mathrm{ERA,ice}} > 100\,\%$ are underrepresented. The PDF of $r_{\mathrm{ERA,ice}}$ is characterized by a rectangular shape, the distribution of $r_{\mathrm{P1,ice}}$ is smaller in magnitude, broader in width, and skewed towards $r_{\mathrm{P1,ice}} > 100\,\%$. Furthermore, at all $p$-levels, mean $r_{\mathrm{ERA,ice}}$ (red line, column 3 in Fig. 3), is generally shifted to lower values compared to mean $r_{\mathrm{P1,ice}}$ (black line). This indicates a lack of ISSR in ERA5 that is expected from its use of saturation adjustment (Sec. 2.2.1). The resulting MDs are determined to be

$-4.3\,\%$ (200 hPa), $-3.8\,\%$ (225 hPa), and $-5.5\,\%$ (250 hPa).

Smoothing the IAGOS data, as outlined in Sec. 2.2, leads to mean values of native and smoothed $T_{\mathrm{P1}}$ and $r_{\mathrm{P1,ice}}$ that are similar by a tenth of a degree and 1 %, respectively, with almost identical PDFs (not shown here). Hence, the differences in the





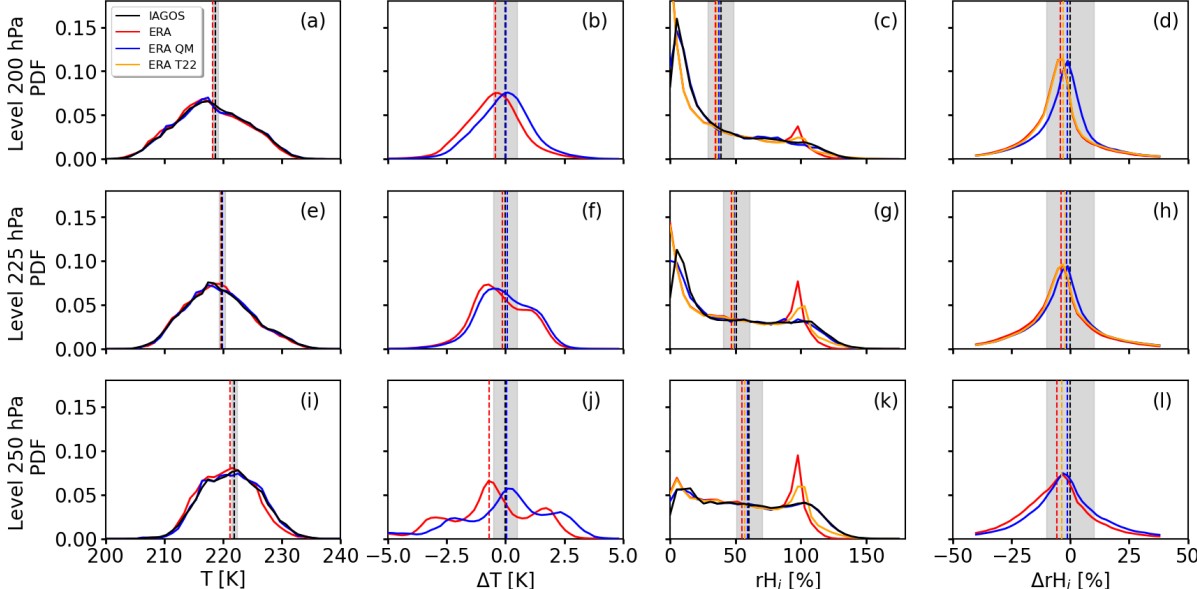

**Figure 3.** Probability density functions (PDFs) of temperature $T$ (in K) and relative humidity $r_{ice}$ w.r.t ice (in %) from IAGOS (black), ERA5 (red), and bias-corrected ERA5 data (blue). From top to bottom, rows represent pressure levels 200, 225, and 250 hPa. The first column shows PDFs of temperature from IAGOS $T_{P1}$, ERA5 $T_{ERA}$, and the bias-corrected ERA5 $T_{ERA}^{cor}$. The second column presents absolute difference of $T_{ERA}$ and $T_{ERA}^{cor}$ with respect to $T_{P1}$. Columns three and four are the same as columns one and two but for relative humidity $r_{ice}$. In addition, bias-corrected $r_{ERA,ice}^{T22}$ using the correction method after T22 is given in orange. Differences are calculated by subtracting the IAGOS reference from the model output. In each plot, the median values of the distributions are indicated by the vertical dashed lines with the black line indicative of the IAGOS data. For reference, the average measurement uncertainties for $T_{P1}$ and $r_{ERA,ice}$ with $\pm 0.5$ K and $\pm 10$ % are indicated around mean $T_{P1}$ and $r_{ERA,ice}$, respectively.

PDFs of ERA5 and IAGOS, as well as the bias in mean $r_{ERA,ice}$ compared to $r_{P1,ice}$ cannot be attributed to differences in the spatial resolutions but to a lack of humidity in ERA5 itself.

IAGOS measurements in the lower stratosphere that are typically characterized by low values of $r$, are subject to a wet-bias. This wet-bias is a non-linear function of $r$ and requires a multi-dimensional regression correction that is currently under development (Konjari et al., 2022). Therefore, this known wet-bias in IAGOS is not corrected in our analysis and it should be kept in mind that subsequent differences between ERA5 and IAGOS for low values of $r$ may also be attributable to artifacts in the IAGOS measurements. However, since the focus of this analysis is to investigate contrail formation and persistence, only

high values of $r$ are relevant. Consequently, the wet-bias for low $r$ values in the IAGOS observations has little impact, if any, on our analysis.

To correct the lack of ISS, the QM-technique is applied. After the QM-correction the MDs are reduced almost by half to $-1.3$ % (200 hPa), $-1.5$ % (225 hPa), and $-0.9$ % (250 hPa), which indicates a remaining slight dry-bias in $r_{ERA,ice}$ as the MD remains negative (see Fig. 5j). However, the QM-correction leads to an adjustment of the entire PDFs such that the shape





**Table 2.** Mean values of temperature $T$ and relative humidity $r_{\text{ice}}$ from IAGOS and ERA5 calculated from the original and the corrected values using the QM-correction and the scaling from T22. The data includes filtered measurements from January 2015 to January 2021. Values in parentheses are the differences relative to IAGOS.

| Pressure level (hPa) | $\overline{T}_{\text{P1}}$ (K) | $\overline{T}_{\text{ERA}}$ (K) | $\overline{T}_{\text{ERA}}^{\text{cor}}$ (K) | |
|---|---|---|---|---|
| 250 | 221.9 | 221.2 (−0.7) | 221.9 (0.0) | |
| 225 | 219.8 | 219.7 (−0.1) | 219.9 (0.1) | |
| 200 | 218.7 | 218.3 (−0.4) | 218.7 (0.0) | |
| Pressure level (hPa) | $\overline{r}_{\text{P1,ice}}$ (%) | $\overline{r}_{\text{ERA,ice}}$ (%) | $\overline{r}_{\text{ERA,ice}}^{\text{cor}}$ (%) | $\overline{r}_{\text{ERA,ice}}^{\text{T22}}$ (%) |
| 250 | 60.4 | 54.9 (−5.5) | 59.4 (−0.9) | 56.8 (−3.7) |
| 225 | 50.6 | 46.8 (−3.8) | 49.1 (−1.5) | 48.6 (−2.0) |
| 200 | 38.8 | 34.5 (−4.3) | 37.5 (−1.3) | 35.8 (−3.0) |

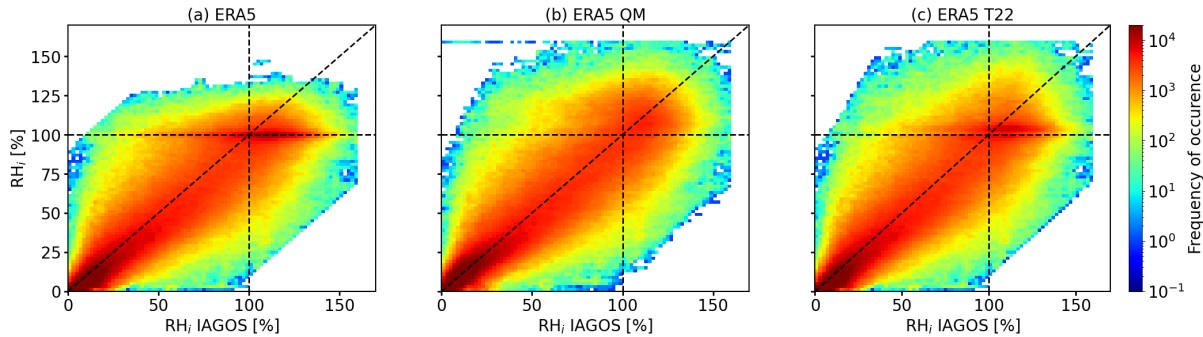

**Figure 4.** (a–c) Bidimensional histogram of original $r_{\text{ERA,ice}}$ (left), QM-corrected $r_{\text{ERA,ice}}^{\text{cor}}$ (center), and T22-corrected $r_{\text{ERA,ice}}^{\text{T22}}$ (right) as a function of IAGOS-observed $r_{\text{P1,ice}}$. Relative humidity is binned in intervals of 2 %. $r_{\text{ice}}$ is given as relative humidity with respect to ice. Pressure levels 250, 225, and 200 hPa are combined. Perfect agreement is indicated by the diagonal dashed line and ice saturation is indicated by the horizontal and vertical dashed lines.

of the PDFs of corrected $r_{\text{ERA,ice}}^{\text{cor}}$ match the IAGOS observations. For comparison, we apply the T22-correction that only partly removes the dry-bias, resulting in MDs between −3.7 % (250 hPa) and −2.0 % (225 hPa) (see Fig. 5j). Furthermore, differences in the second mode in relation to the IAGOS observations remain as the T22-correction cannot modify the shape of the PDF by construction. An overview of the original and corrected mean $T$ and $r_{\text{ice}}$ is given in Table 2.

The individual PDFs of $r_{\text{ice}}$ are used to compile joined 2–dimensional (2D) histograms that are shown in Fig. 4a–c. In general, the frequency distribution of $r_{\text{ERA,ice}}$ and $r_{\text{P1,ice}}$ generally follows the diagonal line of 'ideal' agreement (Fig. 4a). However, the distribution is slightly shifted to below the 1:1-line, indicating a lower $r_{\text{ERA,ice}}$ and therefore dryer conditions in ERA5 compared to the observations. Particularly striking is the elongated feature of the $r_{\text{ERA,ice}}$ distribution positioned close to 100 % (second mode) and a flattening for $r_{\text{ERA,ice}} > 130$ % as a result of the saturation adjustment. Gierens et al. (2020) presented a similar comparison of $r_{\text{ice}}$ between ERA5 and IAGOS, providing only a scatter plot and not a density distribution.




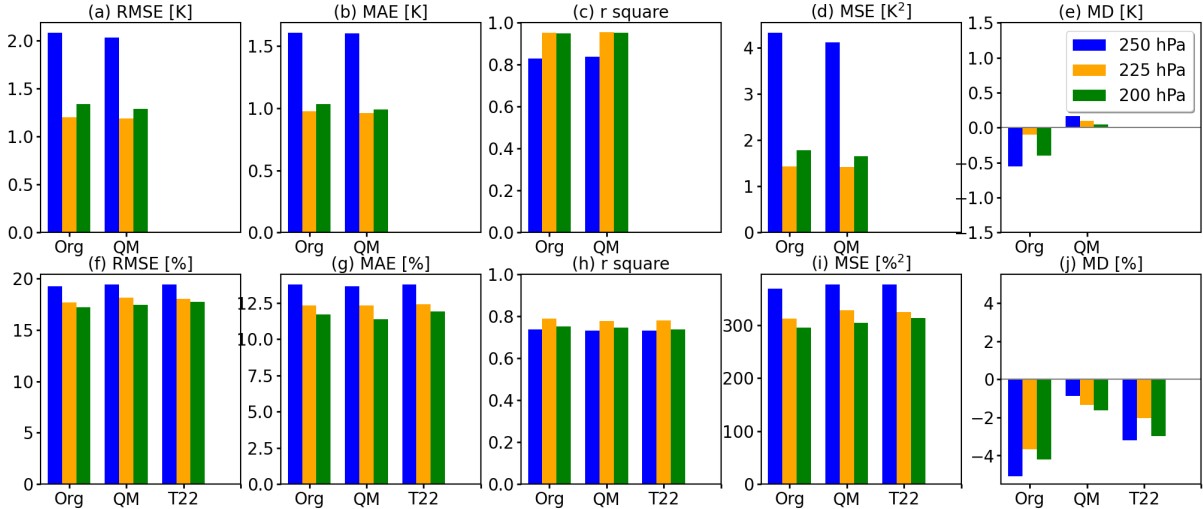

**Figure 5.** Bar plots of: **(a, f)** root mean square error (RMSE); **(b, g)** mean absolute error (MAE); **(c, h)** $r^2$–score; **(d, i)** mean square error (MSE); and **(e, j)** mean difference (MD) of ERA5 against IAGOS. First row shows metrics for $T$ and the second row for $r_{ice}$. The first set of bars represent the original ERA5 output (label Org) while the second set represents the data set after the quantile-mapping correction (label QM). In the second row a third set of bars indicates the T22-correction. The metrics are calculated for pressure levels of 250 (blue), 225 (orange), and 200 hPa (green).

They found a strong scattering around the 1:1-line and described the distribution as "scattered all over the place" with a poor agreement among $r_{ERA,ice}$ and $r_{P1,ice}$. While we agree that the distributions in Fig. 4a–c are subject to scattering, the majority of the points (read to dark-red colors) show a reasonable alignment along the 1:1-line. For the individual pressure fields of 250, 225, and 200 hPa, $r^2$-score of 0.74, 0.79, and 0.75 are determined, respectively (also see Fig. 5h).

After the application of the QM-correction the alignment with the 1:1-line is improved (see Fig. 4b). As expected from 310 Fig. 3, the artificially pronounced second mode in $r_{ERA,ice}$ is removed in $r_{ERA,ice}^{cor}$ and the distribution extends further towards $r_{ice} > 130$ %, better representing the conditions observed by IAGOS. The QM-correction leads to $r^2$-values of 0.73, 0.78, and 0.75 at 250, 225, and 200 hPa, respectively, that are similar to the uncorrected ones (also see Fig. 5h).

For reference, the T22-corrected $r_{ice}$ is compared with observed $r_{P1,ice}$ and shown in Fig. 4c. The scaling of the T22-method enhances $r_{ice}$ that are close or above 100 % and shifts the elongated feature towards higher $r_{ice}$, which is still present. For this 315 correction $r^2$ of 0.73 (250 hPa), 0.78 (225 hPa), and 0.74 (200 hPa) are calculated. So this type of correction leads to a decrease in the $r^2$-score compared to the original ERA5 data.

The top row in Fig. 5 visualizes calculated metrics for the temperature. In general, the 250 hPa $p$-level is characterized by the largest RMSE of 2.1 K, MAE of 1.6 K, and MSE of 4.3 K$^2$ in relation to the other $p$-levels, which is explained by the enhanced natural variability in the temperature field with increasing $p$-level. A larger natural variability leads to larger differences among 320 the IAGOS measurements and the ERA5 grid-box mean values. At the 225 and 200 hPa levels, in a more stratified atmosphere, the RMSE, MAE, and MSE are generally lower and similar for both $p$-levels with values around 1.2 K, 1 K, and 1.5 K$^2$,



**Table 3.** Mean absolute relative humidity (in % with respect to ice) and difference in relative humidity of ERA5 and the corrected ERA5 data with respect to IAGOS as the reference separated for cloud fraction (CF). Calculations are given for a combination of the 250, 225, and 200 hPa levels.

| Cloud fraction (0–1) | Mean $r_{\mathrm{P1,ice}}$ | Mean $r_{\mathrm{ERA,ice}}$ | Mean $r_{\mathrm{ERA,ice}}^{\mathrm{cor}}$ | Mean diff. $r_{\mathrm{ERA,ice}}$ | Mean diff. $r_{\mathrm{ERA,ice}}^{\mathrm{cor}}$ |
|---|---|---|---|---|---|
| $0 \leq \mathrm{CF} < 0.2$ | 41.6 | 36.8 | 39.4 | −4.8 | −2.2 |
| $0.2 \leq \mathrm{CF} < 0.4$ | 90.5 | 95.6 | 103.3 | 5.1 | 12.8 |
| $0.4 \leq \mathrm{CF} < 0.6$ | 98.7 | 100.7 | 109.3 | 2.0 | 10.6 |
| $0.6 \leq \mathrm{CF} < 0.8$ | 103.4 | 100.3 | 108.4 | −3.1 | 5.0 |
| $0.8 \leq \mathrm{CF} \leq 1.0$ | 105.7 | 98.9 | 104.9 | −6.8 | −0.8 |

respectively. The QM-correction leads to a minimal increase in the $r^2$-score at all $p$-levels, while RMSE, MAE, and MSE increase unnoticeable. However, as expected and as it was demonstrated before, the MD is significantly reduced.

Similarly, the bottom row in Fig. 5 visualizes the calculated metrics for the original, the QM-corrected, and the T22-corrected $r_{\mathrm{ice}}$ against the IAGOS observations. As for the temperature, the RMSE, MAE, and MSE are largest for the 250 hPa $p$-level followed by the 225 and 200 hPa $p$-levels. At all $p$-levels, the QM- and T22-corrections lead to a constant or marginally increased RMSE, MAE, MSE, while the $r^2$-score remains almost constant. The increase in RMSE, MAE, and MSE appears counter-intuitive from the results shown in Fig. 4, with an improvement in the mean values and the distributions. However, both correction methods are purely statistical and do not remove differences in temperature and relative humidity of individual data points. Instead singular data points might be falsely adjusted by the QM-correction, which then creates outliers on which the RMSE and MSE respond very sensitively, thus the large RMSE and MSE for relative humidity. In contrast, MAE is less susceptible to outliers.

## 3.2  Influence of ERA5 cloud fraction on the relative humidity distribution

The IFS ISS adjustment depends on the ERA5 cloud fraction $\mathrm{CF}_{\mathrm{ERA}}$ as only cloud-containing grid-boxes are clipped in $r_{\mathrm{ice}}$ (Tompkins et al., 2007). The effect of $\mathrm{CF}_{\mathrm{ERA}}$ on the distribution of the $r_{\mathrm{ice}}$ are investigated by separating $r_{\mathrm{P1,ice}}$ and $r_{\mathrm{ERA,ice}}^{\mathrm{cor}}$ for bins of $\mathrm{CF}_{\mathrm{ERA}}$ that range from 0 to 1 with bin sizes of 0.2.

The $r_{\mathrm{ice}}$ distributions in the left column of Fig. 6 show the formation of the secondary peak at $r_{\mathrm{ice}} = 100$ % when $\mathrm{CF}_{\mathrm{ERA}}$ approaches 1. The distributions of $r_{\mathrm{ERA,ice}}$ also form a secondary mode with a triangular shape, which is not observed by IAGOS data in any way.

Due to the clipping and the method $\mathrm{CF}_{\mathrm{ERA}}$ is determined from $r_{\mathrm{ice}}$ both variables depend on each other. $r_{\mathrm{ice}}$ close to 100 % are likely associated with cloud formation or cloud presence. Furthermore, the sorting of IAGOS observations by means of ERA5 $\mathrm{CF}_{\mathrm{ERA}}$ might be flawed due to the different spatial resolutions of ERA5 and IAGOS. We estimate the IAGOS cloud fraction $\mathrm{CF}_{\mathrm{IAGOS}}$ for each bin of $\mathrm{CF}_{\mathrm{ERA}}$ by counting the number of IAGOS measurements inside a cloud divided by the number of all samples for which measurement from the backscatter cloud probe are available. A cloud particle number concentration $N_{\mathrm{ice}}$ of 0.015 cm$^{-3}$ is used to identify in-cloud measurements (Petzold et al., 2017). Measurement points above



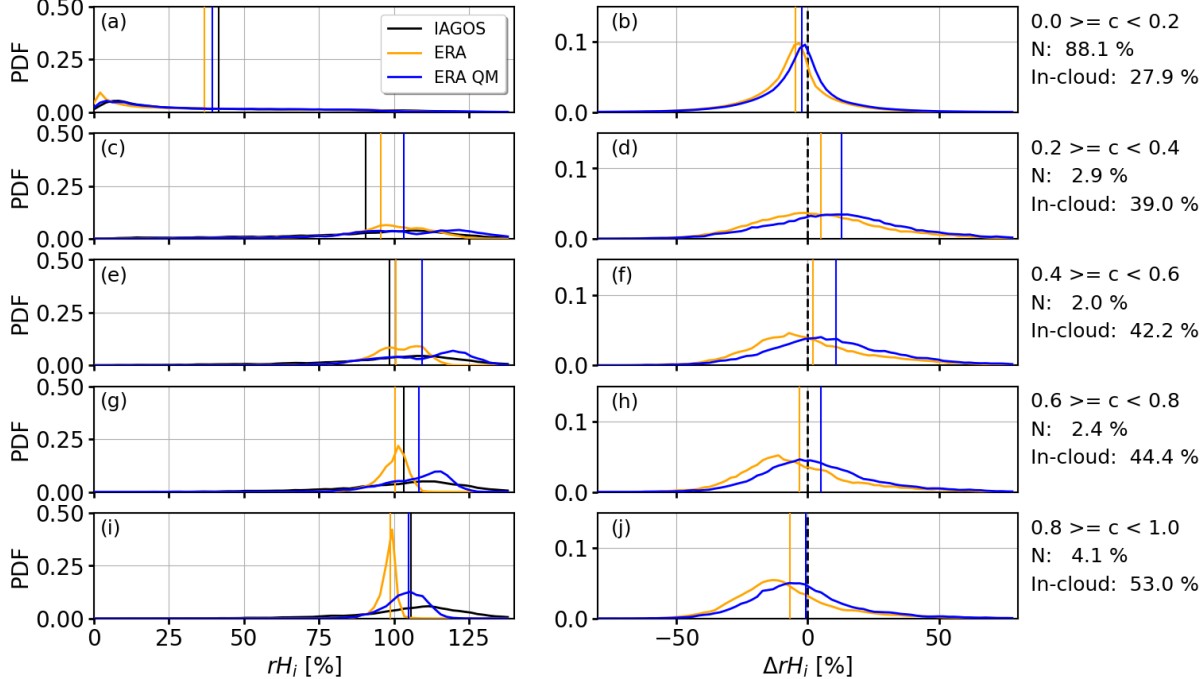

**Figure 6. (a)** Probability density functions (PDFs) of relative humidity (in %) with respect to ice observed by IAGOS (black), simulated by ERA5 (orange), and ERA5 corrected using the quantile–mapping–correction (blue). **(b)** Same as left column but for differences of relative humidity with IAGOS as the reference. Relative humidity is binned for cloud fraction (0–1) with bin sizes of 0.2. Values from $p$-levels 250, 225, and 200 hPa are included.

the threshold are flagged as cloudy. While $CF_{IAGOS}$ is not directly comparable with $CF_{ERA}$, a consistent pattern is identified between both parameters. With increasing $CF_{ERA}$ from CF < 0.2 to $0.8 \leq CF < 1.0$, the fraction of in-cloud measurements from IAGOS represented by $CF_{IAGOS}$ also increases from 27.9 % to 53.0 %. Hence, a correlation between $CF_{ERA}$ and $CF_{IAGOS}$ exists, which demonstrates the representativeness of the IAGOS observations.

For the lowest bin ($CF_{ERA}$ < 0.2) the distributions of $r_{ice}$ from ERA5 and IAGOS are dominated by dry air leading to small peaks close to $r_{ice} = 0$ % and are otherwise flat (Fig. 6a,b). The corresponding mean bias $\overline{\Delta r_{ice}}$ between original ERA5 (orange) and the IAGOS observations is −4.8 %. The QM-corrected values are close to the observations with a remaining mean bias of −2.2 %. The large majority (88.1 %) of all data belongs to this category. For intermediate $CF_{ERA}$ between 0.2 and 0.6 (Fig. 6c–f), $r_{ERA,ice}$ is slightly moister than observed by IAGOS leading to $\overline{\Delta r_{ice}}$ of 5.1 % (0.2 <= $CF_{ERA}$ < 0.4)

and 2.0 % (0.4 <= $CF_{ERA}$ < 0.6). In these two $CF_{ERA}$ bins the QM-correction exaggerates $r_{ice}$, resulting in $\overline{\Delta r_{ice}}$ of 12.8 % (0.2 <= $CF_{ERA}$ < 0.4) and 10.6 % (0.4 <= $CF_{ERA}$ < 0.6). In total, only 4.9 % of all data points belong to these two categories limiting the negative impact of the over correction on the full sample entity. With further increasing $CF_{ERA}$ (Fig. 6g–j), $\overline{\Delta r_{ice}}$ from original ERA5 becomes negative again with values of −3.1 % (0.6 <= $CF_{ERA}$ < 0.8) and −6.8 % (0.8 <= $CF_{ERA}$ < 1.0). In these cases the QM-corrected values are still moister but close to the observations with related $\overline{\Delta r_{ice}}$





**Table 4.** Fractions of measurement points labeled as non-persistent and persistent contrail formation, reservoir conditions, and no contrail formation. The results using the scaling method after Teoh et al. (2022b) is labeled with T22.

|  | IAGOS | ERA5 | | | | |
|---|---|---|---|---|---|---|
|  |  | original | $T$ correction | $r_{ice}$ correction | $T+r_{ice}$ correction | $r_{ice}$ correction T22 |
| Condition |  |  |  |  |  |  |
| NPC | 39.2 | 40.8 | 38.6 | 46.3 | 44.0 | 46.9 |
| PC | 16.9 | 17.5 | 16.8 | 11.4 | 10.9 | 10.5 |
| R | 1.9 | 1.8 | 2.2 | 1.2 | 1.5 | 1.2 |
| None | 41.9 | 39.9 | 42.3 | 41.1 | 43.6 | 41.3 |

of 5.0 % (0.6 <= CF$_{ERA}$ < 0.8) and −0.8 % (0.8 <= CF$_{ERA}$ < 1.0). The categories 0.6 <= CF$_{ERA}$ < 0.8 and 0.8 <= CF$_{ERA}$ < 1.0 contribute by 2.4 % and 4.1 % the total sample size, respectively.

This analysis suggests that the QM correction generally performs well in situations with no (CF = 0) or high cloud cover (CF >=0.8), with the exception of a small subset of data points.

### 3.3   Along-track contrail formation potential and the effect of applied corrections

Along-track time series of uncorrected and corrected ERA5 data and IAGOS measurements are flagged for non-persistent (NPC) and persistent (PC) contrails, and reservoir (R) conditions using the method described in Sec. 2.4. All data points that do not belong to any of the categories are flagged for no contrail formation (NoC). Considering all data points from the years 2015 to 2021 at $p$-levels 250–200 hPa, it is found that 39.2 % of the IAGOS observations show a potential for NPC formation. PC appear to be less frequent with about 16.9 % and R conditions are rare with an occurrence of 1.9 % only. Using the original

along-track ERA5 output, the contrail formation potential for NPC, PC, and R is estimated at 40.8 %, 17.5 %, and 1.8 %, respectively, which is close to the values identified using IAGOS data.

For reference, Teoh et al. (2020) estimated that 18.4 % of the flights form contrails (i.e., at least one contrail section during a flight) with only 7.4 % of the total, analyzed flight distance led to contrails. The latter value is approximately half the distance we identified. There are two main potential sources of disagreement. First, the account of aircraft characteristics is different.

The estimates of our study solely rely on the SAc including constant values for fuel properties (specific heat capacity $Q$) and aircraft-engine specific parameters (efficiency $\eta$). Contrarily, the more elaborate method by Teoh et al. (2020) uses a fleet data set that includes flight specific information of aircraft engine type, thrust settings during flight stages, and estimates of black carbon (soot) emissions. These information were ingested into the contrail cirrus prediction model from (CoCiP,  Schumann, 2012) to determine contrail formation and the related radiative effect. It is noted that CoCiP only considers flight sections as

a contrail, when a certain contrail radiative effect is exceeded, i.e., the ice particle number is larger than $10^3$ m$^{-3}$ and the cirrus optical thickness is larger than $10^{-6}$ (Schumann, 2012). Thus, the results from Teoh et al. (2020) consider the potential for contrail formation, actual aircraft emissions, the synoptic conditions, and the contrail radiative effect. For our approach,





with the IAGOS data set, no such aircraft-performance data are available. Secondly, the way flight distance is counted as contrail-forming is different between the two studies. In this study, the SAc accounts only for thermodynamic properties.

Subsequently, the impact of corrected $T_{\mathrm{ERA}}^{\mathrm{cor}}$ and $r_{\mathrm{ERA,ice}}^{\mathrm{cor}}$ on the along-track classification of NPC, PC, and R is investigated. The individual contributions of $T$ and $r$ are separated by applying the QM-correction separately on $T$ and $r$. The scaling method from T22 is shown as a benchmark.

When applying the QM correction to $T_{\mathrm{ERA}}$ only, the fraction of NPC decreases from 40.9 % to 38.6 % and for PC from 17.5 % to 16.6 %, respectively. This is in good agreement with the IAGOS observations (see Table 4). This correction increases

the mean $T_{\mathrm{ERA}}^{\mathrm{cor}}$ (ambient temperature) allowing fewer ERA5 samples to pass the $T$ and $r_{\mathrm{ice}}$ thresholds for NPC and PC formation. At the same time the fraction of R conditions increases, where supersaturation is achieved but the SAc is not fulfilled.

When applying the QM correction to $r_{\mathrm{ERA,ice}}$ only, the frequency of NPC increases to 46.3 %. At the same time the number of PC drops to 11.4 % and R conditions are reduced to 1.2 %. Thus, this correction leads to an overestimation of NPC and an underestimation of PC and R conditions, likely due to a higher mean $r_{\mathrm{ERA,ice}}^{\mathrm{cor}}$, which allows for more NPC formation but

the saturation is insufficient to reach supersaturation to fulfill the SAc and form PC. At the same time individual values that exceeded the ice supersaturation threshold might got drier and were removed from the PC group.

The scaling-based T22-correction is most similar to the QM correction of $r_{\mathrm{ERA,ice}}$ only. After the T22-correction, 44.0 % of the samples were identified as NPC, which is above the IAGOS reference and similar to the estimated occurrence after QM

correction of $r_{\mathrm{ERA,ice}}$ only. With the T22 correction, PC and R–conditions are found in 10.9 % and 1.5 % of the cases, which is also comparable to the QM correction of $r_{\mathrm{ERA,ice}}$ only.

Applying the QM correction to both $T_{\mathrm{ERA}}$ and $r_{\mathrm{ERA,ice}}$ leads to an increase of NPC to 46.9 %, which is above the frequency determined using IAGOS and the original ERA5 data, but lower than with the QM correction of $r_{\mathrm{ERA,ice}}$ only, due to the simultaneous correction of $T$ and $r_{\mathrm{ERA,ice}}$. PC conditions are determined for 10.5 % of the samples, which is similar compared

to the QM correction of $r_{\mathrm{ERA,ice}}$ only but lower than on the basis of the original ERA5 data. The frequency of R conditions remains almost unchanged. The frequency of NoC increases to 41.3 %, better matching with IAGOS than for the original ERA5 data.

Removing the bias in the $T_{\mathrm{ERA}}$ and $r_{\mathrm{ERA,ice}}$ led to an increase in mean $T$ and $r$, which are counter acting in terms of contrail formation. As shown with the results from the bivariate QM-correction, larger mean relative humidity do not immediately

propagate into an increase of PC and NPC occurrence. Furthermore, Fig. 3 showed that the distributions of $r_{\mathrm{ERA,ice}}$ were modified such that the second mode at $r_{\mathrm{ice}} = 100$ % was removed by redistributing $r_{\mathrm{ERA,ice}}$ towards lower and higher values. Consequently, individual samples that are at $r_{\mathrm{ice}} = 100$ % became drier or moister after the correction. Using the original ERA5 output it is found that 97.1 % of the data points are below the $T_{\mathrm{crit}}$ calculated by the SAc, while after the correction 96.3 % of the samples are below $T_{\mathrm{crit}}$. Before the QM-correction of $r_{\mathrm{ERA,ice}}$, 58.2 % of the data samples exceed the respective $r_{\mathrm{crit}}$,

while after the correction only 54.9 % of the data points still exceed $r_{\mathrm{crit}}$. Thus, the total number of samples passing the SAc and leading to NPC or PC is reduced.





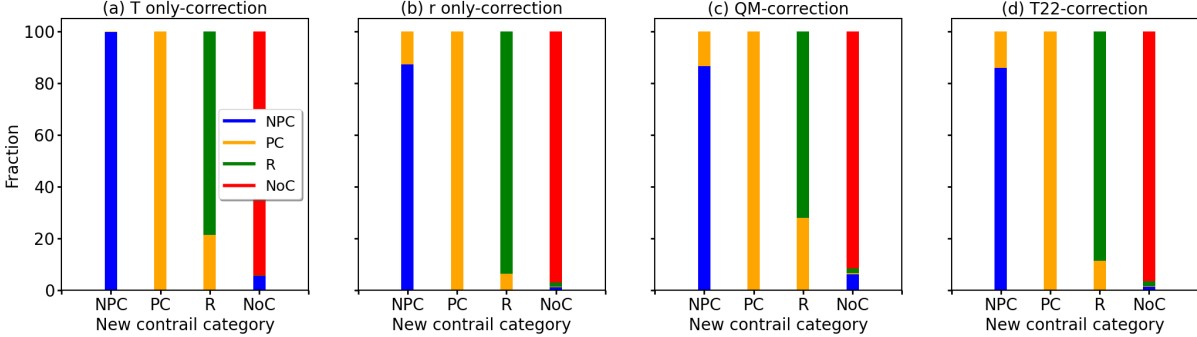

**Figure 7. a–d** Redistributed fractions (in %) of original ERA5 contrail classification with respect to the classification after applying the $T$-only correction, the $r$-only correction, the QM-correction, and the correction after T22, respectively. Non-persistent contrail (NPC) are given in blue, persistent contrails (PC) are given in orange, reservoir conditions (R) are colored in green, and samples that do not allow for contrails formation (NoC) are given in red.

For a detailed understanding on how the QM-correction modifies the classification of NPC, PC, R, and NoC, the redistribution among the contrail categories is quantified by tracking the classification before and after the corrections. Figure 7a–d shows the contribution of the pre-correction categories to the classification after applying a specific correction method. For example, Fig. 7a shows that for the majority of the QM $T_{ERA}$ only corrected ERA5 samples, 99 % of the samples that are now classified as 'NPC' (left bar) have already been NPC before the correction. Only a minority of the new NPC samples were identified as PC or belonged to the NoC category before. However, the QM with $T_{ERA}$ only correction does not influence the PC category significantly (Fig. 7a, second bar from left). The QM $T_{ERA}$ only correction leads to the largest redistribution in the R category. Due to the increase in mean $T_{ERA}$, previous PC-flagged samples now contribute by 21 % to the R category. However, the share of R conditions with respect to the total number of samples with 2.2 % is small. Similarly, samples that have been classified as NPC before contribute to the NoC group by 5 %.

After the application of the QM $r_{ERA,ice}$ only correction (shown in Fig. 7b), the NPC category contains around 13 % previous PC samples. Also a transition from NPC to R is found even though less significant compared to the QM-$T_{ERA}$–correction. The QM $r_{ERA,ice}$ only correction enhanced the mean $r_{ERA,ice}^{cor}$ but certain samples also become dryer and thus previous PC become NPC. The remaining samples in the PC category are unchanged without any significant contribution from the other categories. For the T22-correction a similar redistribution pattern is found (see Fig. 7d). An exception is the category for R conditions, which is now comprised of a larger fraction of PC of 6 %.

The QM-correction is a superposition of the QM $T_{ERA}$ only and QM $r_{ERA,ice}$ only correction correction. Newly classified NPC are comprised of 13 % previous PC samples. Samples that are now in the R–category are composed of 28 % and 72 % from PC and R conditions, respectively. However, the majority of the NoC remains in the category and is comprised of 6 % and 2 % former NPC and R samples, respectively.



**Table 5.** Schematic contingency table for a binary event. Adapted from Stephenson (2000).

| IAGOS | ERA5 detection | |
|---|---|---|
| detection | Yes | No |
| Yes | True Positive (TP) | False Negative (FN) |
| No | False Positive (FP) | True Negative (TN) |

All corrections have in common that the composition of the PC group remains mostly unchanged with no newly formed PC. An exception is the bivariate QM-correction, which contains 0.9 % of former NPC samples in the PC category. The consistency of this category, with no newly formed PC, is of particular interest due to the contrail longevity ($t > 10$ min) and radiative impact. Furthermore, independent of the correction method fewer than 1 % of the NoC data points are converted into either of the contrail classes. Together with the constant composition of the PC group it shows that ERA5 is reliable in terms of overall PC estimation. In previous studies the dry-bias and the use of saturation adjustment in ERA5 was compensated by lowering the ISS threshold from 100 % to 95 %. In the presented approach, with the QM-correction in all its forms, the values of $T_{\mathrm{ERA}}$ and $r_{\mathrm{ERA,ice}}$, which go into the SAc, themselves are corrected, while the threshold for ISS is kept at the physical value of $r_{\mathrm{ice}} = 100$ %. Consequently, the ambient conditions for contrail formation are modified and not just the supersaturation threshold that separates NPC from PC. Therefore, an increase or decrease in $r_{\mathrm{ice}}$ does not directly lead to an increase or decease of NPC and PC. This is particularly true for the PC category, which requires ice supersaturation that itself is temperature dependent.

### 3.4 Analysis of collocated contrail formation potential from ERA5 and IAGOS

The overall contrail formation potential statistics analyzed above are not sufficient for some applications, like contrail rerouting, for which an accurate temporal and spatial representation of $T$ and $r_{\mathrm{ice}}$ is required to accurately diagnose the position of ISSR as well as resulting NPC and PC regions. Subsequently, the collocated temporal and spatial along-track representation of NPC and PC in ERA5 are validated against IAGOS observations using a confusion matrix. A confusion matrix is a special type of a two dimensional contingency table for which a schematic is given in Table 5. In our case the classification is based on: i) the IAGOS observations and ii) the ERA5 data. Each dimension is divided in a binary decision in 'true' and 'false'.

For example, Tompkins et al. (2007) used such a confusion matrix to compare the representation of ISSR in ERA5 with surface observations of PC. To establish comparability, we adapt this approach and use the same metrics. The 'Hit rate' (HR) or 'true positive rate' is defined as:

$$HR = \frac{TP}{TP + FN}, \tag{7}$$

 

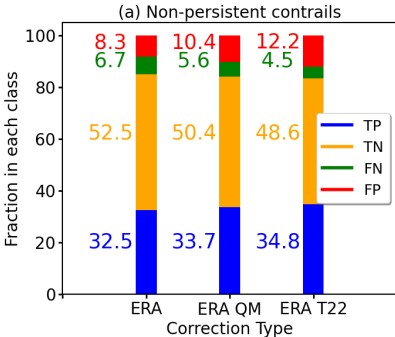 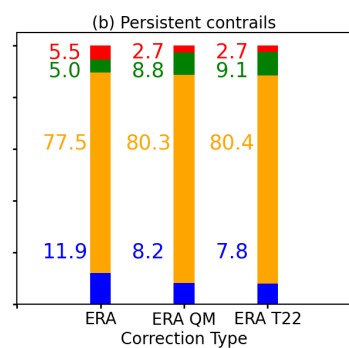 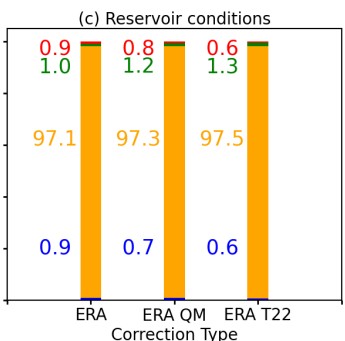

**Figure 8. (a–b)** Fraction of true positive (TP), true negative (TN), false positive (FP), and false negative (FN) predictions from ERA5 data classifications based on IAGOS observations (reference) for non-persistent and persistent contrails, as well as reservoir conditions, respectively. ERA5 data is compared in its original form 'ERA org' (first column), after the QM-correction 'ERA cor2d' (second column), and using the correction after Teoh et al. (2022b) 'ERA T22' (third column).

with TP the number of true positive samples and FN the number of false negative samples in the confusion matrix. The HR is a measure for the probability of a positive test result based on the condition that the individual test is truly being positive. The 'false alarm' (FA) rate or 'false positive rate' is calculated as:

$$FA = \frac{FP}{FP + TN}, \tag{8}$$

with FP the number of false positive and TN the number of true negative values from the confusion matrix. The FA can be understood as how often a true conditions is falsely rejected. In addition, we calculate the 'accuracy' (ACC), which is defined by:

$$ACC = \frac{TP + TN}{TP + TN + FP + FN} \tag{9}$$

and can be interpreted as how close the ERA5 estimated contrail occurrences are to that estimated from the IAGOS observations. HR, FA, and ACC are calculated with Eq. (7)–(9) on the basis of TP, TN, FP, and FN, which are determined for the original as well as corrected ERA5 data against IAGOS estimates (see Table 6 and Fig. 8). The results are compared with Tompkins et al. (2007).

For the original ERA5 data a HR of 0.68 is determined, which is higher than the HR of 0.59 calculated by Tompkins et al. (2007). We also determine a lower FA rate of 0.07 (original ERA5) compared to 0.14 in Tompkins et al. (2007). Tompkins et al. (2007) calculated an ACC of 0.71, while we obtain 0.89. Consequently, the current version of ERA5 has a higher hit rate, lower false alarm rate, and a larger accuracy, which indicates that the overall performance of ERA5, even in the uncorrected form, is at least similar or even has improved since the implementation of the ice cloud microphysical scheme. However, it is noted that the analysis by Tompkins et al. (2007) focused on ice supersaturation and persistent contrails that could be observed from the surface. Therefore, no values for the NPC and R categories are available. In addition, the different nature of the observations might lead to these discrepancies. While Tompkins et al. (2007) used surface observations of actually formed PC we compare



**Table 6.** Hit rate, false alarm, and accuracy calculated from the confusion matrix between IAGOS (reference) and the original ERA5 as well as the corrected ERA5 output. Values for column 'TO07' are from Tompkins et al. (2007) validating persistent contrails in ERA5 with surface observations.

|     |             | ERA5 | ERA5 (3h) | ERA5 QM | ERA5 T22 | T07  |
| --- | ----------- | ---- | --------- | ------- | -------- | ---- |
| PC  | Hit rate    | 0.68 | 0.67      | 0.73    | 0.75     | 0.59 |
|     | False alarm | 0.07 | 0.06      | 0.09    | 0.10     | 0.14 |
|     | Accuracy    | 0.89 | 0.90      | 0.87    | 0.89     | 0.71 |
| NPC | Hit rate    | 0.79 | 0.79      | 0.76    | 0.74     | -    |
|     | False alarm | 0.11 | 0.11      | 0.10    | 0.08     | -    |
|     | Accuracy    | 0.85 | 0.85      | 0.84    | 0.83     | -    |
| R   | Hit rate    | 0.49 | 0.47      | 0.50    | 0.49     | -    |
|     | False alarm | 0.01 | 0.01      | 0.01    | 0.01     | -    |
|     | Accuracy    | 0.98 | 0.98      | 0.98    | 0.98     | -    |

only the PC formation potential. Consequently, the observations used in Tompkins et al. (2007) might have missed PC that were identified in our approach, which leads to the improved metrics.

Applying the QM-correction leads to more correctly detected PC with an increase in the HR from 0.68 to 0.73 but at the cost of an increase of the FA rate from 0.07 to 0.09. The overall accuracy is slightly reduced from 0.89 to 0.87. For the NPC category a slight decrease in HR from 0.79 to 0.76 is observed, with the FA and ACC being constant. The HR for R–conditions

increases to 0.5 but also at the cost of an increased FA rate and a decrease in ACC to 0.98. Consequently, the QM-corrections allows for better estimates of PC formation compared to the original ERA5 data with slightly deteriorated performance for NPC and R.

Finally, we compare the performance of the T22–correction. For the PC category an increase in HR to 0.75 is computed, which is above the HR of 0.68 found for the original ERA5 data and slightly better than obtained by the QM-correction.

Similarly, there is an increase in FA from 0.07 to 0.1, while the ACC remains constant. The performance for NPC is slightly deteriorated compared to original ERA5 and the QM-correction. Considering the R conditions, the performance remains almost unaffected.

Comparing model simulations with observations is an inherently challenging task, particularly measurements in a three-dimensional space from which each sample represents only a small volume of air compared to a large grid-box. Using the

confusion matrix above further mandates an accurate temporal and spatial agreement and does not consider potential measurement and model uncertainties. Hence, even minor spatial mislocations of the temperature or humidity field can lead to a misclassification in the confusion matrix. This might be problematic for atmospheric conditions during which $T$ and $r$ vary on small spatial scales, for example, along frontal zones.

To estimate the effect of potential shift in patterns, we use the three-hour extracted ERA5 data of $T_{\mathrm{ERA}}$ and $r_{\mathrm{ERA,ice}}$.

Evaluating PC formation with the confusion matrix among IAGOS and the coarsened ERA5, the HR marginally decreases



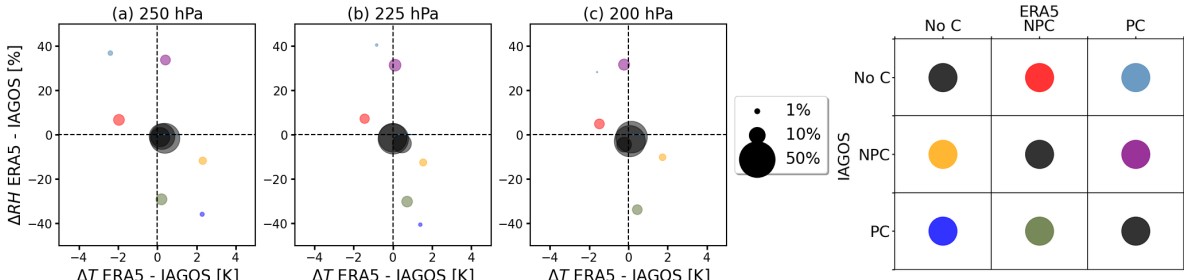

**Figure 9. (a–c)** Mean difference in temperature $T$ (in K) and relative humidity $r_{\text{ice}}$ (in %) between ERA5 and IAGOS for three pressure levels: from left to right, 250, 225, and 200 hPa. Colored dots represent a combination of mean $\Delta T$ and $\Delta r$ for one of the nine categories of the contingency table (right). The area of the dots is proportional to the fraction of measurement–simulation pairs with respect to the total number per pressure level. Colors indicate the classification, using the legend shown in the right-hand panel with categories no contrail formation (NoC), non-persistent contrails (NPC), and persistent contrails (PC).

from 0.68 (original ERA5) to 0.67. FA and ACC remain almost constant with values of 0.06 and 0.9, respectively. Similarly, the values of HR, FA, and ACC for both categories of NPC and R remain almost constant. The decrease in HR and ACC was expected as the difference in temperature and relative humidity among ERA5 and IAGOS get larger. For example, a frontal zone might have shifted its location, which is picked up by IAGOS but is not represented in the 3-hourly ERA5 data. However,
reducing the temporal resolution from one to three hours is shown to be less relevant. Consequently, the sensitivity of the confusion matrix and the calculated metrics is small for temporal and spatial decorrelations that occur within 3h.

### 3.5   Disentangling of classification with respect to temperature and relative humidity

Even after QM-correction, around 16 % of the NPC and 11.5 % of the PC observation–measurement pairs are categorized as 'false positive' and 'false negative'. The sensitivity study using 3 hourly ERA5 data showed that this is unrelated to spatial mis-
matches but is rather due to actual deviations in temperature and relative humidity between IAGOS and ERA5. Subsequently, we aim to quantify the mean differences in temperature and relative humidity that contribute to the misclassification of ERA5 estimates.

     The along track-samples from IAGOS and ERA5, are categorized by a contingency table with groups: NoC, NPC, and PC taking IAGOS as the reference. The created contingency table is visualized in the legend of Fig. 9. The diagonal elements
of the contingency table represent combinations of IAGOS and ERA5 that agree in terms of contrail occurrence, while all off-diagonal values are incorrectly classified. For each of the nine contingency table combinations the corresponding mean differences in the temperature:

$$\overline{\Delta T} = \frac{1}{n}\sum_{i=0}^{n} T_{\text{ERA5,i}} - T_{\text{IAGOS,i}} \tag{10}$$



and relative humidity:

$$\overline{\Delta r_{\mathrm{ice}}} = \frac{1}{n}\sum_{i=0}^{n} r_{\mathrm{ERA5,ice,i}} - r_{\mathrm{IAGOS,ice,i}} \tag{11}$$

are calculated, with $n$ the number of data points in each group. Figure 9a–c presents the 2D–space spanned by $\overline{\Delta T}$ and $\overline{\Delta r}$ for each of the contingency table combinations at $p$-levels 250, 225, and 200 hPa, respectively. In the following, a notation of 'A–B' with $A, B \in \{\mathrm{NoC}, \mathrm{NPC}, \mathrm{PC}\}$ is used as an abbreviation for the classification of A from IAGOS and B in ERA5. For example, a notation of 'NPC-PC' means a combination of IAGOS NPC conditions and ERA5 PC conditions.

In general, $\overline{\Delta T}$ and $\overline{\Delta r_{\mathrm{ice}}}$ are similar at all three $p$-levels and the three $p$-levels are discussed simultaneously and the fraction of each group compared to the total number of samples are given for the middle layer at 225 hPa. 81.9 % of the observation–model combinations are correctly categorized and represented along the contingency table-diagonal. As expected, corresponding $\overline{\Delta T}$ and $\overline{\Delta r_{\mathrm{ice}}}$ (black dots) are close to the origin.

Contrarily, the off-diagonal groups are mostly located in the top-left and lower-right quadrants. Misclassifications for PC–NPC (green, 4.5 %) and NPC–PC (violet, 5.4 %) due to errors in $\overline{\Delta r_{\mathrm{ice}}}$ mostly. Samples in the PC–NPC group (green) were incorrectly categorized due to a too low relative humidity in ERA5, while the NPC–PC samples (violet) were too moist. But of course, since $r_{\mathrm{ice}}$ depends on $T$, mis-classifications are also caused by errors in $T$, even if they do not dominate in these cases.

Misclassifications for the combinations 'NoC–NPC' (red, 3.7 %) and 'NPC–NoC' (yellow, 1.9 %) are mostly due to errors in $T$. For 'NoC–NPC' and 'NPC–NoC', $T_{\mathrm{ERA}}^{\mathrm{cor}}$ was colder or warmer than $T_{\mathrm{P1}}$, respectively.

Least frequent are the misclassifications 'NoC–PC' (light blue, 0.3 %) and PC–NoC (dark blue, 0.5 %). These two groups are subject to the largest $\overline{\Delta T}$ and $\overline{\Delta r_{\mathrm{ice}}}$. Samples in these categories were only found at the 225 hPa $p$-level. It is likely that data points in the two categories result from small scale variations captured by IAGOS but that could not be represented by ERA5 due to temporal and spatial resolution. The absence of NoC–PC and PC–NoC at the 200 hPa $p$-level, where $T$ and $r_{\mathrm{ice}}$ are more homogeneous, supports the hypothesis. Consequently, the misclassification in ERA5 with respect to IAGOS for all classifications is caused by either too dry and warm or too cold and moist conditions.

## 4 Summary and discussion

In this study we proposed a temperature and relative humidity correction method for ERA5 based on a bivariate quantile mapping (QM) technique. The QM-correction was trained on 3.5 years of IAGOS observations and collocated ERA5 data of $T_{\mathrm{ERA}}$ and $r_{\mathrm{ERA,ice}}$. The QM-correction was then applied on 5.5 years of ERA5 data and compared with IAGOS. The target region covers the eastern United States, the North Atlantic, and continental Europe, spanning 30°N to 70°N and 110°W to 30°E for pressure ($p$) levels 250 to 200 hPa, where the majority of IAGOS observations are available (93.8 %).

Alongside the IAGOS data post-processing and the calculation of cumulative distribution functions (CDF) for the QM-correction, the along-track biases in temperature and relative humidity among ERA5 and IAGOS were analyzed. In general, biases in temperature and relative humidity are characterized by a dependence at $p$-level with the largest differences typically for the lowest $p$-level, i.e., 200 hPa. Biases were further separated for their dependencies on latitude and longitude. While the





bias in the temperature was found to be independent of longitude and latitude, the bias in relative humidity was smallest in North America and increased towards continental Europe. The temporal consistency of IAGOS relative humidity measurements was investigated by means of monthly climatologies. A constant bias in temperature and relative humidity between ERA5 and IAGOS was found. An exception are IAGOS relative humidity measurements from the year 2017, when IAGOS observations
tend towards elevated relative humidity observations with respect to the other years, while the bias in temperature remained constant.

Using the bivariate QM-correction, the bias in $T_{\mathrm{ERA}}$ was reduced from $-0.7$, $-0.1$, and $-0.4$ K, at $p$-levels 250, 225, and 200 hPa, respectively, to below 0.1 K at all $p$-levels. The relative humidity bias was reduced from $-5.5$, $-3.8$, and $-4.3$ % to $-0.9$, $-1.5$, and $-1.3$ % at 250, 225, and 200 hPa, respectively. While a slight dry-bias remains, a significant improvement in
terms of the probability density functions (PDFs) of the relative humidity distribution is achieved. PDFs of corrected relative humidity are almost identical in shape with the PDFs determined from the IAGOS observations. A previously existing artificial peak at $r_{\mathrm{ERA,ice}} = 100$ % in the PDFs of ERA5, which is caused by the saturation adjustment in ERA5, was removed. Consequently, corrected values of $r_{\mathrm{ERA,ice}}^{\mathrm{cor}}$ better represent the actual conditions in terms of mean value and frequency of occurrence.

Subsequently, the impact of the QM correction on the detection and classification of NPC, PC, and R with respect to IAGOS was evaluated. Measurements from IAGOS and along-tack ERA5 data were flagged for NPC, PC, R, and NoC conditions. Based on the original ERA5 data set, 40.8, 17.5, and 1.8 % of all data points were identified as NPC, PC, and R, respectively. Similar values were found for IAGOS with 39.2, 16.9, and 1.9 % for NPC, PC, and R, suggesting a good statistical representation of contrail formation conditions in ERA5. After the ERA5 QM correction, 44, 10.9, and 1.5 % of the samples were
identified as NPC, PC, and R conditions, indicating a slight overestimation of NPC and an underestimation of PC and R. Using a parameterized relative humidity correction from Teoh et al. (2022a), here used as a reference for comparison, led to 46.9, 10.5, and 1.2 % of NPC, PC, and R conditions, respectively.

The increase in mean temperature and relative humidity lead to a decrease in PC occurrence. However, the composition of the PC group remained unchanged, which indicated that removing biases in ERA5 does not lead to more or newly formed
PC contrails in ERA5. So from a statistical perspective, that the original ERA5 model output is able to adequately represent contrail formation.

The temporal and spatial representation of NPC, PC, and R in ERA5 with respect to IAGOS was assessed with a contingency table. Based on the contingency table, the hit rate (HR), false alarm (FA), and accuracy (ACC) were calculated. The QM-correction leads to an increase in HR from 0.68 to 0.73 and FA from 0.07 to 0.09. ACC marginally decreased from 0.89 to
0.87. In the NPC category the HR, FA, and accuracy remained almost constant. For R conditions HR, FA, and ACC remained unchanged after the correction. Consequently, the correction had no significant effect on the classification of along-track occurrence of contrail formation. Using the original ERA5 output it was found that 97.1 % of the data points were below the respective $T_{\mathrm{crit}}$, while after the correction 96.3 % of the samples were below $T_{\mathrm{crit}}$. Before the QM-correction of relative humidity, 58.2 % of the data samples exceeded the respective $r_{\mathrm{crit}}$, while after the correction only 54.9 % of the data points still
exceeded $r_{\mathrm{crit}}$.





The contingency table further revealed that 81.9 % of the data samples were coherently flagged in IAGOS and ERA5 after QM-correction. In these cases almost no biases in temperature and relative humidity between IAGOS and ERA5 remain. The remaining 18.1 % of the data points, which were incorrectly classified for NPC, PC, and R conditions by ERA5, are caused by remaining biases in temperature and relative humidity of varying magnitude. The misclassifications were insensitive

to the applied correction method. The magnitude of the individual bias contributions - temperature or relative humidity - depend on the discrepancy of the classification. False classifications of NPC vs PC were primarily dominated by a relative humidity bias, while false classifications of NPC vs NoC were dominated by a bias in the temperature. However, the majority of misclassifications were caused by combinations of temperature and relative humidity biases with ERA5 either being cold-moist or a warm-dry biased. Furthermore, the relative humidity bias between IAGOS and ERA5 was found to depend on the

temperature.

Overall, the presented QM-correction allowed to remove the systematic bias in temperature and relative humidity in ERA5 with IAGOS as the reference. However, the correction did not significantly alter the diagnosed contrail formation potential. This small impact is due to the threshold nature of the SAc, meaning that only the situations that are close to critical $T_{\mathrm{crit}}$ and $r_{\mathrm{crit}}$ are susceptible to be mis-classified. However, for applications that require quantitative estimates of $r$, for example to

calculate the optical thickness or lifetime of PC, the QM corrected ERA5 may be better.

*Code availability.* The python code that was used to perform the analysis and the quantile correction is provided following: https://doi.org/10.5281/zenodo.8418565

*Data availability.* ERA5 data can be obtained from the European Centre for Medium-Range Weather Forecasts (ECMWF) data catalog at https://doi.org/10.24381/cds.f17050d7 (Hersbach et al., 2023).

The IAGOS data can be downloaded from the IAGOS data portal at https://doi.org/10.25326/20 (Boulanger et al., 2020).

## Appendix A: Temporal consistency in temperature and relative humidity of IAGOS and ERA5

Applying the quantile mapping (QM)-correction in the presented form requires a time-invariant bias in temperature ($T$) and relative humidity ($r_{\mathrm{ice}}$) between the IAGOS and the ERA5. The bias among both might vary due to variations in the instrument calibration procedure or changes in the sampling distribution due to seasonal flight schedules.

We tested for time invariance by calculating mean values of $T$ and $r_{\mathrm{ice}}$ from ERA5 as well as IAGOS over all samples for each month spanning January 2015 to 2021 on $p$-levels of 250, 225, and 200 hPa.

Figure A1a shows that monthly mean $T_{\mathrm{ERA}}$ (red) and $T_{\mathrm{P1}}$ (black) agree well, which is expected from the small bias presented in Fig. 3. Furthermore, the monthly mean difference between $T_{\mathrm{P1}}$ and $T_{\mathrm{ERA}}$, given in Fig. A1b, remains constant with values around $-0.5$ K and maximal $-1$ K, except for some individual spikes. Figure A1b also shows that QM-corrected $T_{\mathrm{ERA}}^{\mathrm{cor}}$ (blue)

better match with $T_{\mathrm{P1}}$, which is indicated by maximal differences of $\pm 0.5$ K.



Similarly, Fig. A1c shows monthly mean of $r_{\mathrm{P1,ice}}$ (black), original ERA5 $r_{\mathrm{ERA,ice}}$ (red), and QM-corrected ERA5 $r_{\mathrm{ERA,ice}}^{\mathrm{cor}}$ (blue) ranging between 40 % and 50 % for the majority of the period. An exception is the period after 2020, which is due to low data availability (see Fig. A1e). Figure A1c illustrates that $r_{\mathrm{ERA,ice}}$ (red) follows $r_{\mathrm{P1,ice}}$ (black) with an offset between 3 % and up to −12 % that has been shown before (fourth column in Fig. 3). Like for the temperature correction, Fig. A1d clearly
shows that the QM-correction increases mean $r_{\mathrm{ERA,ice}}^{\mathrm{cor}}$ such that the bias between ERA5 and IAGOS is reduced, bringing the monthly means of $r_{\mathrm{ERA,ice}}^{\mathrm{cor}}$ closer to the 0.

Even though the bias $\Delta r_{\mathrm{ice}} = r_{\mathrm{ERA,ice}} - r_{\mathrm{P1,ice}}$ remains fairly constant for the majority of the presented time series, the differences are particularly pronounced for the years 2016 and 2017. However, their temperature bias $\Delta T = T_{\mathrm{ERA}} - T_{\mathrm{P1}}$ remains constant (Fig. A1b), which suggests that changes in the sampling, e.g., due to modified aircraft operations, are not
the cause, but the IAGOS measurements or the post-processing of $r_{\mathrm{P1,ice}}$ was negatively affected during these years.

In the absence of alternative observations to compare against IAGOS, we turn to the interannual variation in $r_{\mathrm{ice}}$ to confirm that relative humidity measurements for years 2016 and 2017 are anomalous. Multi-year, monthly climatological means of $r_{\mathrm{ERA,ice}}$ and $r_{\mathrm{P1,ice}}$ are calculated spanning the years 2015 to 2021. Using uncorrected $r_{\mathrm{ERA,ice}}$ as the reference, anomalies of $r_{\mathrm{P1,ice}}$, $r_{\mathrm{ERA,ice}}$, and $r_{\mathrm{ERA,ice}}^{\mathrm{cor}}$ are determined by subtracting the monthly mean of an individual year from the multi-year,
monthly climatological mean. Figure A2 shows mean anomalies of $r_{\mathrm{P1,ice}}$ that range from −11.9 % (2017) to 0.8 % (2020). Similarly, mean anomalies of $r_{\mathrm{ERA,ice}}$ range between −4.2 % (2017) and 5.5 % (2020). The mean anomalies between ERA5 and IAGOS are largest for the years 2016 and particularly 2017 the difference between the anomalies for year 2017 exceeds all other years with −6.8 %. Slightly smaller mean anomaly difference between ERA5 and IAGOS anomalies are found to the years 2016 with −4.7 % and 2020 with −4.7 %. Therefore, years 2017 and parts of 2016 are special cases compared to the
other years in terms of anomalies during which $r_{\mathrm{P1,ice}}$ is likely biased towards too moist values.

## Appendix B: Latitudinal and longitudinal dependent deviations in temperature and relative humidity between ERA5 and IAGOS

The bias between IAGOS and ERA5 might depend on the geographic position, e.g., due to characteristic spatial distributions of water vapor in the atmosphere. Such spatial-dependent biases in $T$ and $r_{\mathrm{ice}}$ among ERA5 and IAGOS are identified by
calculating mean differences for bins of 10° in latitude and longitude, at $p$-levels 300, 250, 225, and 200 hPa. The calculations include all samples from years 2015 to 2021 and from within the defined subdomain (30°N–70°N, 105°W–30°E).

First, the longitudinal variation in $\Delta T$ is analyzed (Fig. B1a). In general, a tendency toward more negative $\Delta T$ is found for decreasing $p$-levels, reaching a maximum on the 200 hPa $p$-level, where $\Delta T$ mostly reaches values of up to −2 K. Large $\Delta T$ on 200 hPa westwards of 80° must be cautiously interpreted due to the low number of available samples in this pressure level and
longitude bin (see Fig. B1e). The general negative $\Delta T$ indicates that the mean temperature from ERA5 is predominantly lower than measured by IAGOS. $\Delta T$ at pressure levels 250 hPa (green) and 225 hPa (red) is almost constant over the entire longitude range with $\Delta T$ being smaller than −0.5. An exception is the 300 hPa level, where $\Delta T$ exceeds −0.5 K and reaches values of



up to −1 K east of 50°W. Separating $\Delta T$ for latitudes between 30°N and 70°N does not reveal any latitudinal dependencies. An exception is the 200 hPa $p$-level, where $\Delta T$ increases towards the equator and reaches up to −1.7 K at 30°N.

Analog to $T$, the longitudinal and latitudinal dependence of $r_{\mathrm{ice}}$ is analyzed (see Fig. B1c, d). In general, $\Delta r_{\mathrm{ice}}$ increase from west, with $\Delta r_{\mathrm{ice}}$ around 0 %, towards east, reaching $\Delta r$ of up to −25 % at the 300 and 250 hPa $p$-levels. No systematic offset among the $p$-levels is found. While $\Delta r_{\mathrm{ice}}$ is largest at the 200 hPa level at 110°W, $\Delta r_{\mathrm{ice}}$ is among the smallest levels at 30°E. Conversely, $\Delta r_{\mathrm{ice}}$ is small at the 250 hPa level at 110°W and is the second largest $\Delta r_{\mathrm{ice}}$ at 30°E. Similar to $T$, separating $\Delta r_{\mathrm{ice}}$ for latitudes shows not strong latitudinal sensitivity with the smallest values between −10 % and −4 % at the 225 and

200 hPa $p$-levels. Largest $\Delta r_{\mathrm{ice}}$, of up to $\Delta r_{\mathrm{ice}} = -25$ %, are found at the 300 hPa level particularly between 40°N and 60°N.

Separating biases in $T$ and $r$ clearly shows the necessity to consider the $p$-level in the QM-correction. In contrast, binning for latitudes appears to be of minor importance, which relaxes the requirement for more than two bins in the proposed QM-correction. In contrast, the dependency of $r_{\mathrm{ice}}$ on the longitude is much more pronounced and would required individual cumulative distribution functions but could not be considered for in the QM-correction as dividing the data in three sub-

domains would lead to insufficient data in rarely sampled combinations of $T$ and $p$.

**Appendix C:  Cumulative distribution functions for quantile mapping**

Here we provide an example for calculated cumulative distribution functions (CDFs) of relative humidity $r$ defined with respect to ice. IAGOS CDFs ($F_{\mathrm{o,h}}$) and ERA CDFs ($F_{\mathrm{m,h}}$) are calculated on basis of the observed IAGOS relative humidity ($x_{\mathrm{o,h}}$) and simulated, along-track ERA5 relative humidity ($x_{\mathrm{m,h}}$), respectively, following the description in Sec. 2.3. Figure C1 shows

$F_{\mathrm{o,h}}$ (dashed lines) and $F_{\mathrm{m,h}}$ (solid lines) for individual pressure ($p$) levels between 350 and 200 hPa. As described in Sec. 2.3 the full domain (specified in Sec. 2.1) is subdivided into two latitude bands. The split point is determined by the $50^{\mathrm{th}}$-percentile at each $p$-level such that both latitude bands contain equal numbers of data points. For legibility, only CDFs of the northern most latitude band are shown here. The selection is arbitrary and conclusions are transferable between the two bands.

The black lines in Fig. C1 indicate $F_{\mathrm{o,h}}$ and $F_{\mathrm{m,h}}$ from the quantile mapping (QM) approach that considers only for $p$-level

dependence and the latitude band. For the majority of the $p$-levels $F_{\mathrm{o,h}}$ and $F_{\mathrm{m,h}}$ are similar in shape. An exception is $r$ between 100 and 110 % at levels $350 \le p \le 250$ hPa, where $F_{\mathrm{m,h}}$ (ERA5) show a dominant mode, while $F_{\mathrm{o,h}}$ (IAGOS) remain flat. The mode in $F_{\mathrm{m,h}}$ is primarily caused by the saturation adjustment in ERA5 (see Sec. 2.2.1). This mode becomes less prominent with decreasing $p$ as the atmosphere gets drier with altitudes, so supersaturation is less likely. Simultaneously, the differences between $F_{\mathrm{o,h}}$ and $F_{\mathrm{m,h}}$ increase for $r < 20$ %, where both $F_{\mathrm{o,h}}$ and $F_{\mathrm{m,h}}$ are further characterized by a steep slope. The largest

effect in this regard is found at the 200 hPa $p$-level, where $F_{\mathrm{m,h}}$ contains a larger fraction of high relative humidity values compared to $F_{\mathrm{o,h}}$, indicating an underestimation of $r$ that is not attributable to the saturation adjustment. For example, 50 % of the ERA relative humidity are smaller than around 15 %, while the respective value for IAGOS is around 22 %, indicating a general dry-bias unrelated to the saturation adjustment.

The color-coded lines in Fig. C1 represent the bivariate QM approach, where $r$ is additionally separated for five temperature

($T$) bins that are defined by 20 %-steps. $F_{\mathrm{m,h}}$ and $F_{\mathrm{o,h}}$ that result from the bivariate QM reveal a strong dependence in $T$,



which becomes visible in the deviating shapes of $F_{\mathrm{m,h}}$ and $F_{\mathrm{o,h}}$ at constant $p$-level and latitude band. The systematic order of the colored lines further indicates that $T$-bins with low $T$ (0–20$^{\mathrm{th}}$-percentile, violet lines) are mostly dominated by high relative humidity values, while bins with higher $T$ (80–100$^{\mathrm{th}}$-percentile, red lines) are dominated by low $r$. The CDFs with lower $T$ are generally flat with a continuous slope, while $T$ bins with higher temperatures are dominated by a steep slope for
$r < 10$ %, particularly for $p < 250$ hPa. However, for the bivariate QM correction the actual shape of $F_{\mathrm{m,h}}$ and $F_{\mathrm{o,h}}$ is less relevant but the difference. These difference between $F_{\mathrm{m,h}}$ and $F_{\mathrm{o,h}}$ are increasing with decreasing $p$-level. The importance to consider the $T$-dependence is further highlighted by the fact that the simpler, univariate QM approach (black) and related $F_{\mathrm{m,h}}$ and $F_{\mathrm{o,h}}$ do not consider the shape and the shape difference that is required to adequately correct $r$ under different ambient conditions, particularly with decreasing $p$-level.

*Author contributions.*  **KW** performed the data analysis and prepared the manuscript. **NB** and **OB** contributed equally to the preparation of the manuscript. **SR** and **YL** provided constructive comments and helped with the interpretation of the IAGOS measurements.

*Competing interests.*  The authors declare no competing interest.

*Acknowledgements.*  This research has been supported by the French Ministère de la Transition écologique et Solidaire (N° DGAC 382 N2021-39), with support from France's Plan National de Relance et de Resilience (PNRR) and the European Union's NextGenerationEU.
NB, AP and YL acknowledge funding from the Horizon 2020 project ACACIA (grant no. 875036). Furthermore, we acknowledge the support from Garry Lloyd for post-processing the IAGOS P1 data of the single particle backscattering optical spectrometer.



**Table A1.** Notations

| Symbol | Long-name | Unit |
|---|---|---|
| $\alpha$ | Scaling factor in ERA5 | - |
| $\eta$ | Efficiency of aircraft-engine-combination | 0-1 |
| $\Phi$ | Latitude | ° |
| $\sigma$ | Standard deviation of Gaussian distribution | - |
| ACC | Accuracy | 0–1 |
| $\mathrm{CF_{IAGOS}}$ | Fraction of in-cloud measurements by IAGOS | 0–1 |
| $\mathrm{CF_{ERA}}$ | Cloud fraction from ERA5 | 0–1 |
| $c_\mathrm{p}$ | Isobaric heat capacity of air | $\mathrm{J\,kg^{-1}\,K^{-1}}$ |
| $e(T)$ | Water vapor pressure, temperature dependent | Pa |
| $e_\mathrm{sat,l}(T)$ | Saturation water vapor pressure over water, temperature dependent | Pa |
| $e_\mathrm{sat,i}(T)$ | Saturation water vapor pressure over ice, temperature dependent | Pa |
| EI | Emission index of water vapor for the fuel | $\mathrm{kg\,kg^{-1}}$ |
| $F$ | Cumulative distribution function for quantile mapping | - |
| FA | False alarm | 0–1 |
| HR | Hit Rate | 0–1 |
| $N_\mathrm{ice}$ | Particle number concentration | $\mathrm{cm^{-3}}$ |
| $p$ | Pressure | hPa |
| $\mathcal{P}$ | Probability for contrail occurrence | 0–1 |
| $r_\mathrm{P1,ice}$ | Relative humidity with respect to ice from IAGOS package 1 (P1) | % |
| $r_\mathrm{P1}$ | Relative humidity with respect to liquid water from IAGOS package 1 (P1) | % |
| $r_\mathrm{ERA,ice}$ | Relative humidity with respect to ice from ERA5 | % |
| $r_\mathrm{crit}$ | Critical relative humidity from Schmidt–Appleman criterion | [0–1] |
| $r^\mathrm{cor}_\mathrm{ERA,ice}$ | Relative humidity with respect to ice from ERA5 bias corrected | % |
| $r^\mathrm{T22}_\mathrm{ERA,ice}$ | Relative humidity with respect to ice from ERA5 corrected with method T22 | % |
| $r_\mathrm{ERA,liq}$ | Relative humidity with respect to liquid water from ERA5 | % |
| $r^\mathrm{cor}_\mathrm{ERA,liq}$ | Relative humidity with respect to liquid water from ERA5 | % |
| $t_\mathrm{1-1/e}(T)$ | Temperature dependent sensor response time to adjust to a signal change by 63% | s |
| $T_0$ | Freezing temperature in ERA5 | K |
| $T_\mathrm{ice}$ | Lower temperature limit for scaling of relative humidity conversion in ERA5 | K |
| $T_\mathrm{crit}$ | Critical temperature from Schmidt–Appleman–criterion | K |
| $T_\mathrm{P1}$ | Temperature measured by IAGOS package 1 (P1) | K |
| $T_\mathrm{ERA}$ | Temperature from ERA5 | K |
| $T^\mathrm{cor}_\mathrm{ERA}$ | Temperature from ERA5 bias corrected | K |
| $q_\mathrm{sat,liq}$ | Saturation specific humidity with respect to a liquid water surface | $\mathrm{kg\,kg^{-1}}$ |
| $q_\mathrm{sat,ice}$ | Saturation specific humidity with respect to a ice surface | $\mathrm{kg\,kg^{-1}}$ |
| $Q_\mathrm{heat}$ | Specific heat capacity | $\mathrm{J\,kg^{-1}}$ |
| $\hat{x}_\mathrm{m,p}(t)$ | Transfer function for quantile mapping | |



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





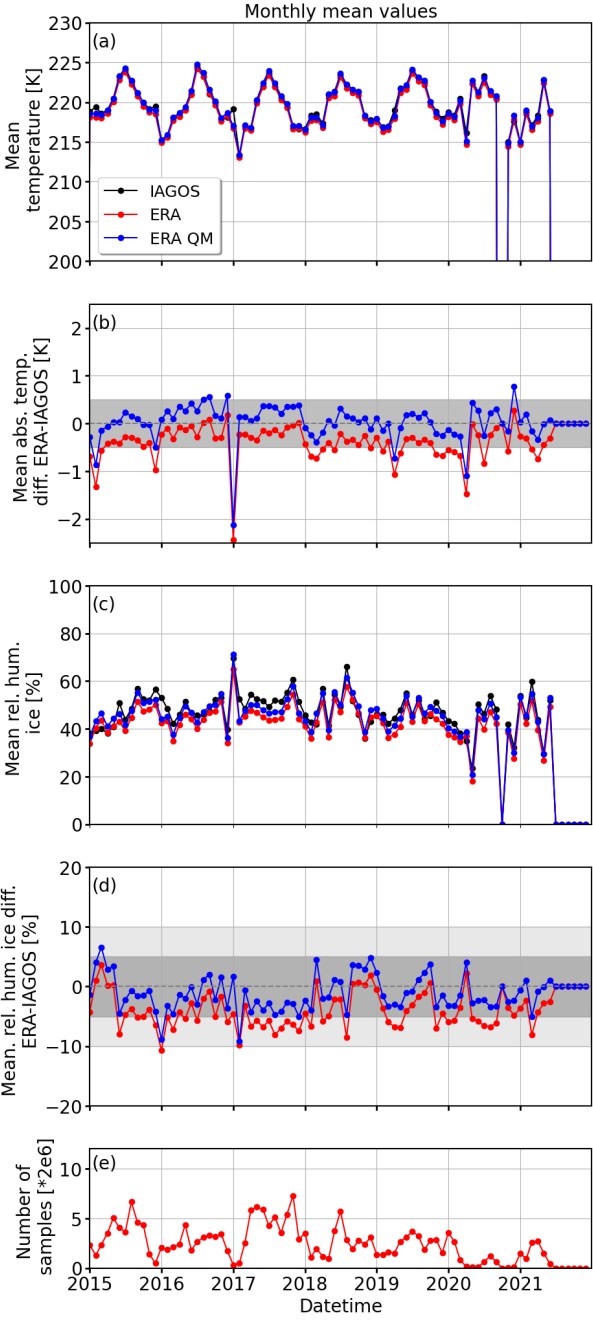

**Figure A1. (a)** Time series of monthly mean temperature (in K) from IAGOS (blue), ERA5 (red), and corrected ERA5 (blue). **(b)** Time series of temperature difference (in K) from ERA5 minus IAGOS (red) as well as corrected ERA5 minus IAGOS (blue). Panels **(c)** and **(d)** are similar to **(a)** and **(b)** but for relative humidity with respect to ice $r_{ice}$ in %. **(e)** Total number of samples available to calculate the monthly mean.



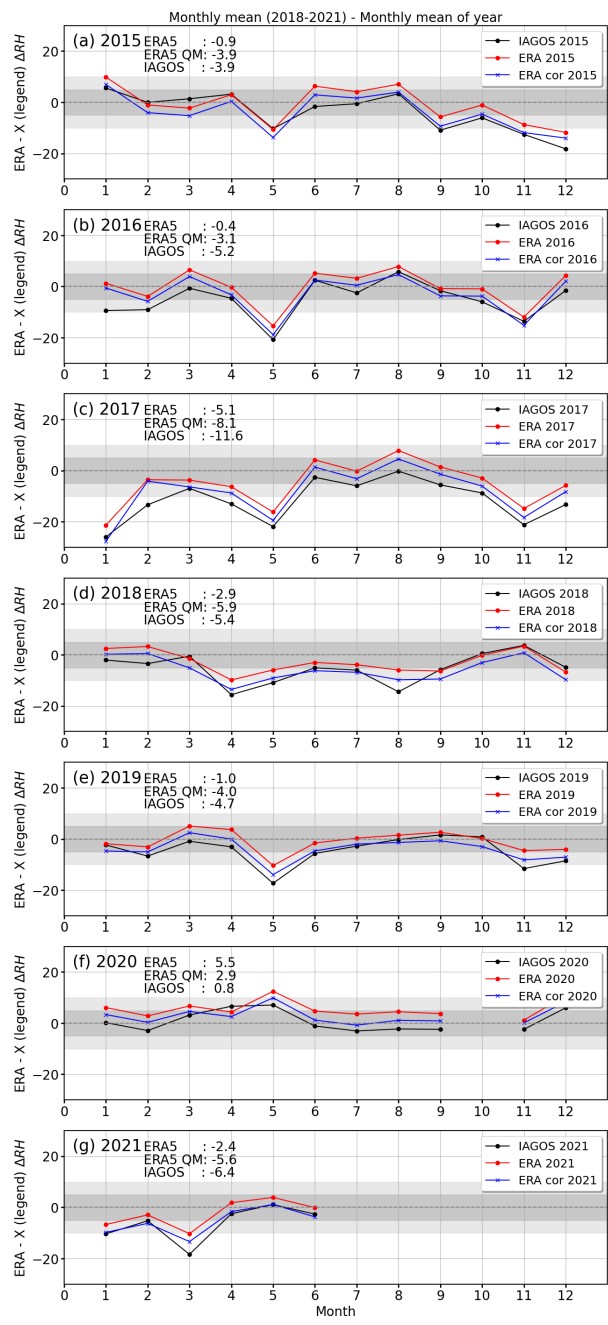

**Figure A2. (a–g)** Anomalies of relative humidity from ERA5 $r_{\mathrm{ERA,ice}}$ (red), corrected ERA5 $r_{\mathrm{ERA,ice}}^{\mathrm{cor}}$ (blue), and IAGOS $r_{\mathrm{P1,ice}}$ (black) with respect to the multi-year $r_{\mathrm{ERA,ice}}$ for the years 2015 to 2021. Differences are given in unit of relative humidity.





**Figure B1. (a–b)** Temperature difference $\Delta T$ (in K) between ERA5 and IAGOS as a function of Longitude and Latitude, respectively. **(c–d)** Same as top row but for difference in relative humidity $r_{\text{ice}}$ (in %). Pressure levels of 300, 250, 225, and 200 hPa are indicated in orange, green, red, and purple, respectively. **(e–f)** Fraction of available samples per longitude or latitude bin with respect to the total number of samples per pressure level.



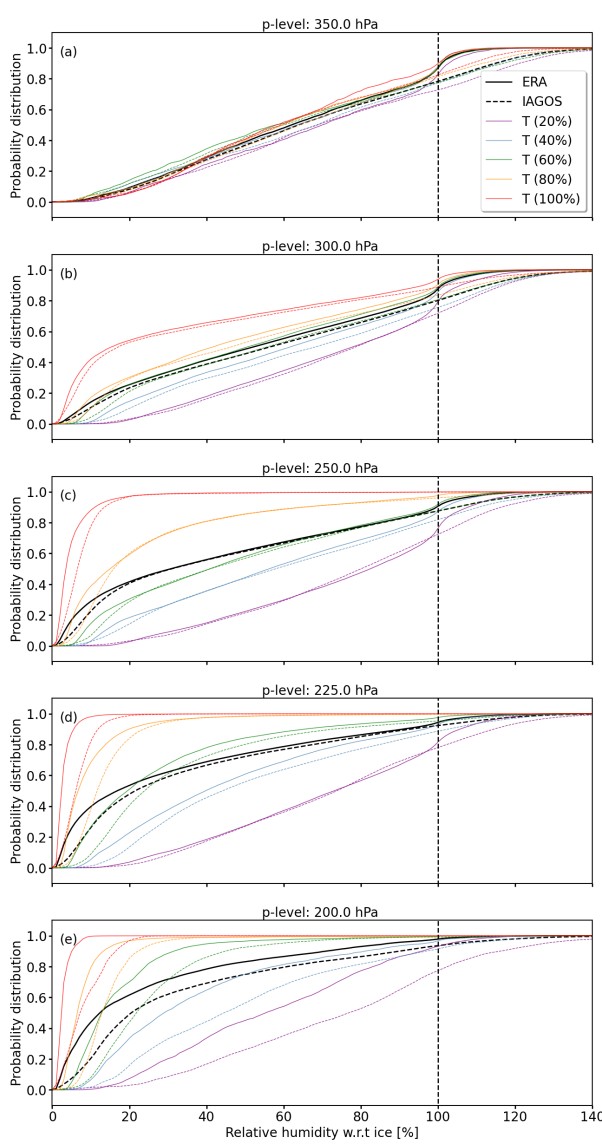

**Figure C1. (a–e)** Cumulative distribution functions (CDFs) $F$ of relative humidity w.r.t ice (in %). CDFs from ERA5 $F_{\mathrm{m,h}}$ and IAGOS $F_{\mathrm{o,h}}$ are given by solid and dashed lines, respectively. The black lines represent $F_{\mathrm{o,h}}$ and $F_{\mathrm{m,h}}$ that depend at $p$-level and latitude $\Phi$. Color-coded are $F_{\mathrm{m,h}}$ and $F_{\mathrm{o,h}}$ that additionally consider for five temperature bins with bin sizes defined by $20^{\mathrm{th}}$-percentiles.