# Peer review of "Correction of ERA5 temperature and relative humidity biases by bivariate quantile mapping for contrail formation analysis."

_EGUsphere, 2023_

## Author Response (AR1)

**Reply to Reviewer #1**

(Referee comment on "Correction of temperature and relative humidity biases in ERA5 by bivariate quantile mapping: Implications for contrail classification" by K. Wolf et al. (egusphere-2023-2356), https://doi.org/10.5194/egusphere-2023-2356-RC1, 2023)

We thank the Reviewer for the time she/he spent on the manuscript and for the useful comments. We have addressed all the comments from the Reviewer, which has improved the manuscript.

For better legibility, the Reviewer's comments are highlighted in **bold** and changes in the manuscript are in *italic*.
* * *
**General Comments:**

**The paper evaluates the ability of ERA5 to correctly represent temperature and relative humidity in the upper troposphere using data from IAGOS as reference. Biases in both fields are detected and characterised and a correction method is applied that corrects both T and then RHi using the so-called quantile mapping method. It is found that this method indeed is able to reduce the biases. Unfortunately it turns out that in spite of this improvement the prediction of contrail formation or contrail persistence is not improved. But at least, for other quantities that are relevant for the assessment of aviation climate impacts, e.g. the optical thickness of contrails, improved estimation seems possible (not individually, but in a statistical sense).**

**While the method leads certainly to a significant improvement of the RHi-statistics of ERA5, the investigations and analyses in the paper are somtimes a bit lengthy and, at least to my view, not always necessary. The differences between the original ERA5, and after the application of the QM method of the correction of Teoh et al are often minor and it is not sure whether they are always real or simply caused by statistical noise. But even if they are real in certain cases, it is not always clear to me why the reader needs to know the potential causes of these differences. I have the feeling that the reader can easily get lost in these details and that the straight way from the analysis to the results and conclusions becomes unclear. So, I think, there is potential to make the paper more concise and to clearly convey a message.**

Multiple sections and paragraphs of the manuscript have been shortened and revised to make the manuscript more concise. For specific modifications to the manuscript we would like to point the Reviewer to the revised manuscript with track-changes and to the specific answers to major comment number 2 as well as to minor comments numbers 14 and 16.

**This paper needs at some places more elaborateness in its formulations. For**

**instance, the sentence (Page 1) "IAGOS flight trajectories are used to extract co-located meteorological conditions - temperature, relative humidity, and wind speed - and are compared with the IAGOS measurements" is obviously a faulty. In short, it says that IAGOS trajectories are compared with IAGOS measurements, which is evidently nonsense.**

Following the Reviewer's comment the sentence has been corrected.

*"IAGOS flight trajectories are used to extract co-located meteorological conditions from ERA5, namely temperature and relative humidity, which are compared with the IAGOS measurements."*

**Also "representation of contrail occurrence in ERA5" (P 1) is misleading, since contrails are not represented in the ECMWF model and reanalysis.**

The sentence has been rephrased to be clearer.

"*To estimate potential contrail formation on the basis of ERA5, data points from IAGOS as well as corrected and uncorrected data points from ERA5, temperature and relative humidity, are flagged for contrail formation using the Schmidt-Appleman criterion*."

**Therefore I recommend a major revision of this paper to make its messages stronger and clearer.**

**Major issues:**

**1.) Reading the abstract it seems that the evaluation merely uses bulk statistics, which I deem insufficient. An good indication of this is figure 5, where, apart from MD, all other measures are very insensitive to the applied corrections. The r² hardly change (and these tiny changes may be insignificant on a p=0.05 level). The r², like the other insensitive statistics, are bulk measures, that say nothing to a point-by-point comparison. The authors acknowledge this in the first sentence of section 3.4.**

The Reviewer is right that we primarily use bulk statistics. However, these bulk statistics do rely on a point-by-point comparison after careful sampling of the ERA5 data. The proposed quantile mapping technique is commonly applied to remove biases in data sets, which is the goal of this study. It is not designed to improve the other statistics. Our study is nevertheless an improvement over the state-of-the-art. Earlier correction methods of relative humidity from ERA-interim and ERA5 estimated the contrail occurrence based on a simple scaling technique, which, in principle also aims to compensate a potential bias in the dataset. Our method is a more elaborate way of applying a bias correction as we not only use an established method but we also apply it as a function of temperature.

A more sophisticated method aiming at exploiting more of the ERA5 variables and/or their three-dimensional structure remains to be developed. This is the subject of ongoing work.

**2.) Section 3.4: The selection of the score values is an unlucky choice to my view, since the TN cases dominate in this data set. Thus, if one would ignore everything and always predict "no ISS", one would already get an impressive accuracy and FA rate. The statement "which indicates that the overall performance of ERA5, even in the uncorrected form, is at least similar or even has improved" is thus misleading. It may be that the author made their choice to compare with Tompkins et al. (2007). If the goal of the paper is to provide better data on the occurrence of ISS, this might be ok, although I still think, that scores that downweight the default TN would be better. If the ultimate goal is, however, forecast of contrail persistence, I think, that the authors should better compare with Gierens et al. (2020) and also use the ETS score described there**.

We followed the suggestion of the Reviewer and computed the equitable threat score (ETS) based on the four entries in the contingency table. Following this work, we replaced the calculation and discussion of hit rate,  false positive rate, and  accuracy with the ETS, which is indeed more informative. The calculated ETS is given in Table 5. Accordingly, the entire sections "Along-track contrail formation potential and the effect of applied corrections" and "Analysis of collocated contrail formation potential from ERA5 and IAGOS" were revised.

in the course of the revision of the metrics, we found a mistake in the contrail flagging algorithm, which has now been corrected. This leads to slightly different results in the contrail categorization and the contingency table in Figures 7 and 8 as well as Tables 3 and 5. The quantile mapping itself remains unchanged but the conversion from RHi w.r.t to ice and liquid water was unnecessarily applied in succession, leading to a cumulative error in the conversion of RH between ice and liquid water. The error was dominant for samples with low temperature, where the saturation curves of liquid water and ice are close to each other and small errors had a significant effect on the calculated RH (Ambaum, 2020).

The revised analysis and the evaluation of the impact of the bias correction on the contrail categorization with the ETS now shows an improvement of NPC, PC, and R detection from ERA5 data against the IAGOS reference observations. A similar improvement was identified for the scaling method in Teoh et al. (2022).

We refer the Reviewer to sections "Along-track contrail formation potential and the effect of applied corrections" and "Analysis of collocated contrail formation potential from ERA5 and IAGOS" in the revised manuscript and the manuscript with track-changes.

**Minor issues:**

**1.) Lines 34-38: I suggest that you state that RF is a global (or at least regional) quantity, averaged over a long time period. On first reading, it was not so clear to me whether you refer to single contrails or contrails in general. Furthermore, aren't the quoted values ERF values (in Lee et al.) rather than RF values?**

To be consistent within the text, we removed the citation from Lee et al. (2021) and only kept the references from Boucher et al. (2021) and Burkhardt and Kärcher (2011), who

give estimates for the radiative forcing.

*"The influence of a perturbation, e.g., clouds, aerosols, or gases, on the Earth's atmosphere and its radiative transfer is quantified by the radiative forcing (RF). By definition, RF is defined as the difference in the net irradiance at the top of atmosphere under perturbed and unperturbed conditionsI(Ramanathan et al., (1989)). In the context of climate studies the RF is understood as the difference in the Earth energy budget due to a contributor to climate change (Bickel et al., 2020). For example, the aviation-induced global CO2-related RF is estimated to be around 30 mWm$^{-2}$ (Boucher et al., 2021). Contrail RF is estimated to be stronger, at about 60 mWm$^{-2}$, but is subject to much larger uncertainties (Burkhardt and Kärcher, 2011)."*

**2.) L 48 and following: This text mixes up different things which is not good. In order to avoid contrails, one needs a precise PREDICTION of where they WILL occur. Knowledge of their occurrence is insufficient, it can only result in a kind of climatology. Schumann's use of the roof-camera is a bad example for predictive purposes, as it shows contrails that already exist and perhaps exist already quite some time. The same comment can be made for satellite or other observations of already present contrails. Then, contrail simulations in a climate model are not intended for contrail avoidance or prediction. CoCiP is the only example here where contrail prediction is the intention, but of course only, if it is fed with actual weather forecasts, not with ERA5.**

We understand the objection of the Reviewer and rephrased the paragraph in the manuscript. We agree that there is only one way to avoid contrails, and this is by using numerical weather prediction models that predict regions that need to be avoided. However, it is useful to obtain a statistical database that documents the spatial and temporal distribution of regions where contrails are likely to form. The paragraph has been rephrased as follows:

*"To lower the climate impact of aviation it is important to reduce CO2 as well as non-CO2 effects. An approach to minimize non-CO2 effects is active flight re-routing to avoid areas where contrails are likely to form and persist, which would require accurate numerical weather predictions. A useful prerequisite is to identify and document flight levels and regions of the Earth's atmosphere that are particularly prone to contrail formation due to meteorological and dynamical conditions that favor contrail formation. Such a statistical data base might be obtained in four different ways.*

*The first approach builds on ground-based observations. For example, [...]"*

**3.) L 75: It is a bit surprising that a "fourth approach" is now mentioned. I believe that the mixed-up list from above (L 48 ff) is now continued, but that is not sufficiently clear. Again, this is not an approach to prediction and should thus not be mentioned as something that has to do with contrail avoidance. I find also, that this section interrupts the logic of the argumentation. It would be better if the**

**paragraph that introduces various correction attempts would directly by followed with tha paragraph explaining the goal of the present paper.**

We agree with the Reviewer and rearranged the paragraphs for a better logical flow. Please also see the answer to minor comment 2.
We now clearly state that the primary way to avoid contrails is active re-routing of flights by using numerical weather prediction and by circumventing contrail prone areas. However, creating a statistical database of contrail formation potential can be achieved by the four methods that we listed. (The list is not necessarily exhaustive.) Due to the length of the paragraphs we direct the Reviewer to the updated manuscript and the manuscript with track-changes.

**4.) Fig. 2 and corresponding text: it seems that either the information in the figure is useless or something is wrong. For instance, why are there so many 175hPa data for EU, where I would expect a lot of landing and departures (i.e. low altitudes)? How do I have to read the figures? It seems, the interpratation is: on 300 hPa most flights are over EU etc..... and on 175 hPa again most flights are over EU. Is this, because most flights are over EU anyway? Then the information is useless. Should't it rather be the following: In EU most flights are on lower altitudes because of a lower fraction of cruise, similar in US but perhaps not as strong, and over the NA all flights are in cruise, therefore a predominance of high altitudes (or low pressures). It seems, the data should be organised the other way, i.e three panels "EU", "NA", "US" and then showing the fraction of pressure levels in the threee panels**.

As described in section "Quantile mapping", the CDFs have been calculated on individual pressure levels that are determined by the ERA5 pressure levels. Individual levels have been considered because the temperature and humidity biases do not have to be consistent in the vertical (across pressure levels) nor in the horizontal (across sub-domains); see Fig. B1 in the appendix. Considering the spatial dependencies of the biases, it is most important to be aware of the distribution of the IAGOS data.

By separating into individual pressure levels (Fig. 2) we intended to clearly show the respective contributions of each region to measurements for a given pressure layer. It should be noted that the total number of samples is given for each sub-panel of the Figure. The majority of samples are obtained at the 200 hPa level, which corresponds to the cruise level.
The Reviewer pointed to the 175 hPa layer, which she / he assumes to be incorrect. It is true that the majority of IAGOS flights depart and arrive in the EU. However, the fraction of samples in the EU domain on the 175 hPa layer is high because aircraft reach their maximum altitude at the end of the flight (when the fuel tanks are almost empty).

**5.) Section 2.1: Please indicate whether all data are used along a flight (it seems so) or a subset to avoid autocorrelation which might spoil the statistics. If autocorrelation has not been avoided, a check should be made whether this affects the results or at least good arguments for this should be given.**

All IAGOS data was filtered for data quality using the provided quality flags provided by the IAGOS post-processing. After filtering, around 90 % of all the data was usable.
The analysis that is presented in the manuscript has been performed with all remaining 90% of the data but also for a second time, where only every fourth data point (one data point approx. every 4 km) was used (second analysis not shown in the manuscript). No significant differences in the results were found.

The data extraction method, i.e., selecting and sampling collocated data from ERA5 based on IAGOS flights in an identical manner, might lead to autocorrelation, which then exists in both extracted data sets and, therefore, should not influence the analysis. Autocorrelation might be a problem if trends within a single time series are analyzed. Autocorrelation is expected to be an issue for the temperature field, which is relatively homogeneous, but will not be an issue with the relative humidity field, which is subject to greater tempo-spatial variability.

To further elaborate on the question about the impact of spatial averaging of IAGOS data and potential autocorrelation we prepared the two plots below. The plot at the top shows IAGOS data at the original 4-s resolution and the lower plot shows temporally smoothed data (60 s). The obtained distributions change little between the original and the smoothed data and, thus, we argue that averaging IAGOS only has a second-order impact on the obtained statistics and subsequent analysis.

[Figure]

**6.) L 153/154: I suggest to delete this statement since the procedure is better explained below from L 173 on. Anyway, I think the argument is weak (I don't remember Schumann's reasons for this) and refers rather to interpolation of specific humidity than relative humidity.**

Interpolation of relative humidity is not straightforward as it depends on the underlying temperature and absolute humidity fields, and is determined on the basis of the exponential Clausius–Clapeyron relationship. Due to the nature of the Clausius–Clapeyron equation, linear interpolation, for example, leads to incorrect values of relative humidity. We rephrased the paragraph and removed the reference to Schumann et al. (2012). The end of the paragraph reads as follows:

*"Spatial and temporal interpolation of relative humidity is not done because the relative humidity depends on the underlying temperature and absolute humidity field, which are both related through the Clausius–Clapeyron relationship.. Due to the exponential nature of the Clausius–Clapeyron equation, linear interpolation, for example, would lead to incorrect values of relative humidity."*

**7.) L 210/211: The statement "The CDFs describe the probability that a certain quantity, for example temperature or relative humidity, exists in the underlying data set" is wrong. I think, we know which quatities are in the data sets, therefore the probability is either zero or one. Please consult a textbook on probability and correct the sentence or leave it out. The CDFs describe the probability that a certain value of a quantity, for example temperature or relative humidity, exists in the underlying data set.**

We agree with the Reviewer that the sentence was poorly phrased. The sentence has been rephrased to the following:

*"The CDFs describe the probability that the value of a quantity (or random variable) X, for example temperature or relative humidity, has a value that is lesser or equal to x."*

**8.) L 239/240: You should delete the second part of the sentence. Contrail formation takes place a few tenths of a second after exhaust. It has nothing at all to do with the vortex phase which starts at, perhaps 20 seconds. I also dislike the next sentence. The SAC has been tested on many flights long ago, and it works excellently. There is a figure in the 1999 IPCC report that shows this (I think, the figure has been taken from a paper by Kärcher).**

The Reviewer is right and we rephrased the sentence to be more precise.

*"The SAc is based solely on thermodynamic principles and has been tested to be a valid approximation although it does not inform on the fate of the contrail, which is a more complicated function of the ambient conditions but also the interactions of the vortex phase with the environment."*

**9.) Section 2.4: I miss information on your choice of an overall propulsion efficiency.**

This is correct and the propulsion / engine efficient is now mentioned in the text.

*"Calculations are performed for kerosene with a fuel specific energy $Q = 43.2$ MJ kg$^{-1}$ and an emission index of water $EIH20 = 1.25$. The overall engine efficiency $\eta$ is set to a typical value of 0.3 (Rap et al., 2010)."*

**10.) L 272: is there really a general decrease of r_ice with p? Why then occurs ISS mostly directly below the tropopause?**

In general, absolute humidity decreases with altitude (e.g. Kiemle et al., 2012; Kaufmann et al., 2018, Kruegger et al 2022). However, local maxima of relative humidity can occur just below the tropopause, e.g., in regions with strong vertical updraft or advection of humidity, which then leads to cloud formation. We followed the suggestion of the Reviewer and rephrased the sentence as following:

*"With the general decrease in absolute humidity and possible intrusion of dry air from the stratosphere, the first mode becomes more and more pronounced with decreasing p, while the second mode flattens and almost vanishes."*

**11.) L 297/298: it is not clear to me why the T22 correction cannot modify the shape of the pdf. Please explain.**

The sentence has been rephrased and is now less definitive. We intended to say that scaling all humidity values above a certain value by a constant increases the values as a whole and shifts the peak of the distribution to higher relative humidity values.

*"Furthermore, differences in the second mode in relation to the IAGOS observations remain as the T22-correction only scales values above a certain threshold, which primarily shifts the bulk of data points from 100 % to higher $r_{ice}$."*

**12.) L 370 ff: To my view the discussion of the differences between the QM and T22 is not convincing. For instance, the quoted thresholds from CoCiP are very very low, so they are not really a constraint. To me, the first question is, whether these differences are statistically robust. If a subset of four out of the five years is used, how large is the change of the quoted values? If it is much smaller than the difference between QM and T22, then a real difference seems more plausible. Second, it might be that you compare apples with oranges, i.e. for instance contrail distance with relative frequency of occurrence.**

To answer the first question of the Reviewer, we want to make clear that we gave the estimate from Teoh et al. (2020) to provide a reference for the reader and to set our results in the context of existing literature. The differences between the results using our approach and the approach by Teoh et al. (2020) are clearly stated in the manuscript. Differences in approaches can partly explain the differences. We do agree with the Reviewer that the

absolute numbers given in our manuscript and in Teoh et al. (2022) might be subject to spatial and temporal effects. At the same time the differences between our calculations of potential contrail occurrence and the calculations from Teoh et al. (2022) must not be overinterpreted. They only indicate the uncertainty due to different approaches.

Regarding the second question of the Reviewer, we think that the comparison is valid as the study by Teoh et al. (2020) entitled "Mitigating the Climate Forcing of Aircraft Contrails by Small-Scale Diversions and Technology Adoption" provides the percentage of flights that form contrails rather than the fraction of time it produces contrail. We direct the Reviewer to Table 1 in Teoh et al. (2020).

**13.) L 408 ff: It is not clear to me how you interpret the SAc. Why does an rcrit occur? As far as I understand it, the SAc gives a threshold temperature which is the maximum T where a contrail could in principle form (at 100% RH). Is rcrit the RHi values along the tangential mixing line between the T threshold (rcrit=100% RH) and the lower temperature below which contrails are always formed (rcrit=0%). This is not completely clear to me.**

The critical threshold $r_{crit}$ is always 100 % (w.r.t liquid water). This was incorrectly formulated in the last version of the manuscript. In course of the manuscript revision the sentence in line 408 was also removed.

**14.) Figure 7 and corresponding text: This text contains a lot of details and numbers that I find not necessary. What is the take-away message that the reader learns from these details? Again, how robust are these values, if you leave one your out, then another one, etc.?**

Following the suggestion by the Reviewer the text has been shortened by removing unnecessary sections. Figure 7a is discussed in detail and the influence of the QM correction (Fig.7c) is presented. We focused on Fig.7 a and c as both are essential to see if the improved correction has an impact on the estimated contrail formation conditions. Furthermore, the discussion of the differences between the two methods is used to explain the rationale behind Fig. 7. Figure 7b and d are only briefly mentioned but not discussed in detail. We further highlight that Fig. 7 and the given numbers should be interpreted in a qualitative and not quantitative way, as they can change slightly between years. The section now ends with a summary (take-away message) that states the robustness of the ERA5 estimated PC occurrence with respect to the temperature and humidity correction. The minor impact of the corrections on the potential PC occurrence implies that ERA5 is suited to estimate regions that are prone for contrail formation. We would like to direct the Reviewer to the revised manuscript to see the changes.

**15.) L 482: "Applying ... more correctly DETECTED...", please reformulate. The QM method does not change any detection.**

The sentence has been rephrased and now reads as:

*"The application of the QM-correction modifies the distributions of temperature and relative humidity in such a way that locations of PC are detected more often where they are supposed to occur and were missing before, resulting in an increase in ETS from 0.27 to 0.36."*

**16.) L 493-506: This is an interesting consideration. But it comes too late. Perhaps a lot of not-so-important numbers could be saved if this consideration would be presented earlier.**

We agree with the Reviewer that this is an important point to mention. Therefore, the paragraph has been moved earlier in the section. The paragraph, which mentions the potential mismatches and that investigates the 3h time resolution, is now used to explain the baseline and sensitivity of the scores on the temporal-spatial misalignment. This provides a reference for the scores that have been calculated on the different correction methods. Differences that exceed the baseline are thus truly attributable to incorrect values of temperature and relative humidity in ERA5 or the corrections, and are therefore of relevance.
We also followed the Reviewer's major comment 2 and calculated the ETS. The ETS is provided in Table 5 and we also added a paragraph to the section, where we briefly mention the impact of the corrections on the ETS.
We would like to direct the Reviewer to the revised manuscript to see the changes in Section "Analysis of collocated contrail formation potential from ERA5 and IAGOS".

**17.) Section 3.5 and Figure 9: please state whether you use for the analysis original or QM corrected ERA5 data.**

The section and the figure are based on QM-corrected ERA5 values as we intended to evaluate the remaining errors after the correction that are associated with a certain classification. The following sentences have been added at the beginning of the section. The caption of the Figure has also been modified.

*"Subsequently, we aim to quantify the mean differences in temperature and relative humidity that remained after the QM correction and that contribute to the misclassification of potential contrail formation. Within the following section all ERA5 values are QM-corrected."*

**18.) L 536-539: If that happens only on 225 hPa, is it then really likely that small scale variations cause these differences? Don't such variations occur on the neighbouring pressure levels as well? In which sense are T and rice "more homogeneous" on 200 hPa? What does that mean and from where do you derive such a conclusion?**

Thank you for pointing this out. The text has been revised and rephrased as follows:

*"Least frequent are the misclassifications 'NoC–PC' (light blue, 0.3 %) and PC–NoC (dark blue, 0.5 %). These two groups are subject to the largest ΔT and Δrice. Samples in these categories were only found at the 250 and 225 hPa p-level, while the PC–NoC (dark blue) is not found at the 200 hPa level. It is likely that data points in the two categories result from small scale variations captured by IAGOS that are not represented by ERA5 due to temporal and spatial resolution."*

**19.) L 539/540: Isn't this a trivial statement? Did you expect something else? Perhaps there are more interesting results in this section that could be more elaborated. For instance, which error usually dominates, is it the error in T or the error in rice. It would also be interesting to see how the differences in absolute humidity contribute to the misclassifications. What is here the conclusion? What needs to be fixed in the ECMWF model more urgently, is it T or the water vapour field? What is the effect of the QM correction in this analysis? I suggest you add a second set of points (squares or empty points), perhaps with arrows, to show how the QM correction shrinks the errors. Please think also on getting rid of the "default" class in the figure. It seems as if a couple of dots are hidden behind the big black dot in the middle.**

We agree with the Reviewer that the sentence is trivial, but we meant to express that it is worth identifying whether the misclassification in ERA5 with respect to IAGOS is most often due to biases in temperature or in humidity. The sentence and paragraph have been rephrased and extended in this regard. The paragraph now reads as follows:

*"It is worth identifying whether the misclassification in ERA5 with respect to IAGOS is most often due to biases in temperature or in humidity. Focusing on the PC representation in ERA5, the primary reasons for a misclassification after the correction is the deviation in rice. This is visualized by the proximity of the violet and green dots to the y-axis (small ΔT), while the differences in rice are larger than ±20 %. Hence, the underestimation (green dot) or overestimation (violet dot) of potential contrail formation is primary related to the underlying humidity field in ERA5."*

**20.) L 575: "So from a statistical perspective, that the original ERA5 model output is able to adequately represent contrail formation." First, the sentence is gramatically incorrect. Second, please explain what means that contrail formation is adequately represented FROM A STATISTICAL PERSPECTIVE? Assume, the model would only predict ISS and PC over Antarctica and nowhere else, but with the correct frequency of occurrence averaged globally, then one could also state that the formation is represented adequately from a statistical perspective. So my question is, whether your statement makes sense**.

In the course of revising the manuscript this sentence has been removed.

**21.) L 590/591: Isn't it the other way, that is, the special bias combination leads to the corresponding entry in the contingency table?**

Following the suggestion of the Reviewer the sentence has been removed as it is a trivial statement. The sentences that follow in the paragraph are sufficient.

**22.) Section 4: should not be entitled "Summary and discussion", since it does not contain any discussion.**

"Discussion" has been removed from the subtitle.

**Miscellaneous**

**23.) Line 3: upper tropospheric**

The sentence has been rephrased as follows*:*

*"The skill of the atmospheric reanalysis ERA5 from the European Centre for Medium-Range Weather Forecasts (ECMWF) at simulating temperature and relative humidity in the upper troposphere and lower stratosphere is assessed by using five years of In-service Aircraft for a Global Observing System (IAGOS) observations."*

**24.) L 19: underlying**

The typo has been corrected.

**25.) L 30: delete "bonds"**

"Bonds" was deleted.

**26.) L 31: What is WC?**

This was a typo and is now corrected to "WV", which is defined on the line before.

**27.) L 41-42: The word "defined" is too strong. In fact, this was never defined. There is only a vague understanding that contrails that survive the vortex phase are somehow persistent.**

The sentence was rephrased as follows:

*"For a contrail to be persistent (with the common meaning that is has a lifetime longer than 10 minutes), the ambient air has to fulfill the SAc and must also be supersaturated with respect to ice."*

**28.) L 67: "due to the high temporal and spatial distribution of WV" ???**

The sentence was rephrased as follows:

*"Slightly less accurate is the prediction and re-analysis of relative humidity, which is generally challenging due to the high temporal and spatial variability of WV."*

**29.)L 73: What is "contrail estimation"?**

To be clearer the sentence has been rephrased as follows:

*"To mitigate the dry bias under conditions close to ice-supersaturation in ERA-interim and ERA5, studies have applied either multiplication factors (Schumann et al., 2013, Schumann et al., 2015) or parameterized corrections (Teoh et al., 2022)."*

**30.) L 126: Replave "multiple" by "many". I believe "multiple years" is nonsense.**

Multiple has been replaced by many.

**31.) L 199: The expression "minimizing the C... test" is confusing. How can a test be minimized. I think this sentence can be dropped since it does not provide essential information.**

We meant that the test statistic was minimized. However, we followed the Reviewer's suggestion and removed the sentence as details about how the parameters were detected can be found in Teoh et al. (2022).

**32.) L 241: criterion (singular).**

Thank you for pointing this out. The word has been corrected.

**33.) L 244: slow**

The typo was corrected.

**34.) L 335: r_ice IS**

The typo was corrected.

**35.) L 433: remove one instance of correction**

The word was removed.

**36.) Table 6: the caption says "TO07" while the table headline says "T07"**

Based on the suggestion from the Reviewer we use the equitable threat score to compare the measurements. Doing so removed the reference to Tompkins.

**37.) L 513: delete comma after ERA5**

The comma was removed.

**38.) L 658: require**

The typo was corrected.

**39.) L 671: showS and remainS**

Both words have been corrected.

**40.) Table A1: eta should be labelled "overall propulsion efficiency". It is not necessary (and not good) to introduce new notions.**

We agree with the Reviewer and added "overall propulsion efficiency" in the text and unified it in the table.

**41.) Figure A1(e): The y-label contains [*2e6]. Please check.**

The label is correct and scales the y-axis, which shows the number of individual IAGOS measurements (every 4 seconds) within a given month.

**References**

Appleman, H.: **The formation of exhaust condensation trails by jet aircraft**, *B. Am. Meteor. Soc.*, 34, 14–20, 1953.

Busen, R. and Schumann, U.: **Visible contrail formation from fuels with different sulfur contents**, *Geophys. Res. Lett.*, 22, 1357–1360, 1995

Heymsfield, A., Baumgardner, D., DeMott, P., Forster, P., Gierens, K., and Kärcher, B.: **Contrail microphysics**, *B. Am. Meteor. Soc*., 91, 465–472, 2010.

Jensen, E., Toon, O., Kinne, S., Sachse, G., Anderson, B., Chan, K., Twohy, C., Gandrud, B., Heymsfield, A., and Miake-Lye R.: **Environmental conditions required for contrail formation and persistence**, *J. Geophys. Res*., 103, 3929–3936, 1998

Kärcher, B., Peter, T., Biermann, U. M., and Schumann, U.: **The initial composition of jet condensation trails**, *J. Atmos. Sci*., 53, 3066–3083, 1996.

Kaufmann, S. / Voigt, C. / Heller, R. / Jurkat-Witschas, T. / Krämer, M. / Rolf, C. / Zöger, M. / Giez, A. / Buchholz, B. / Ebert, V. / Thornberry, T. / Schumann, U. , **Intercomparison of midlatitude tropospheric and lower-stratospheric water vapor measurements and comparison to ECMWF humidity data**, 2018, *Atmos. Chem. Phys*. , Vol. 18, No. 22, p. 16729-16745

Kiemle, C. / Schäfler, A. / Voigt, C. , **Detection and analysis of water vapor transport.**

2012, *Atmospheric Physics: Background -- Methods – Trends,* Springer Berlin Heidelberg: Berlin, Heidelberg, p. 169-184

Krüger, K. / Schäfler, A. / Wirth, M. / Weissmann, M. / Craig, G. C., **Vertical structure of the lower-stratospheric moist bias in the ERA5 reanalysis and its connection to mixing processes**, 2022, *Atmos. Chem. Phys.* , Vol. 22, No. 23, p. 15559-15577

Penner, J. E., Lister, D. H., Griggs, D. J., Dokken, D. J., and McFarland, M. (Eds.): **Aviation and the global atmosphere,** *Cambridge University Press*, Cambridge, UK, 1999

Schmidt, E.: **Die Entstehung von Eisnebel aus den Auspuffgasen von Flugmotoren**, Schriften der Deutschen Akademie der Luftfahrtforschung, 44, 1–15, 1941.

Schumann, U., Ström, J., Busen, R., Baumann, R., Gierens, K., Krautstrunk, M., Schröder, F. P., and Stingl, J.: **In situ observations of particles in jet aircraft exhausts and contrails for different sulfur-containing fuels**, *J. Geophys. Res.*, 101, 6853–6869, 1996.

Schumann, U., Arnold, F., Busen, R., Curtius, J., Kärcher, B., Kiendler, A., Petzold, A., Schlager, H., Schröder, F., and Wohlfrom, K.-H.: **Influence of fuel sulfur on the composition of aircraft exhaust plumes: The experiments SULFUR 1–**7, *J. Geophys. Res.*, 107, 4247, doi:10.1029/2001JD000813, 2002.

Schumann, U.: **Formation, properties and climate effects of contrails,** *C. R. Phys*., 6, 549–565, 2005.

Teoh, R. / Schumann, U. / Majumdar, A. / Stettler, M. E. J., **Mitigating the climate forcing of aircraft contrails by small-scale diversions and technology adoption**, 2020, *Environ. Sci. Technol.* , Vol. 54, No. 5, p. 2941-2950

Teoh, R., Schumann, U., Gryspeerdt, E., Shapiro, M., Molloy, J., Koudis, G., Voigt, C., and Stettler, M. E. J.: **Aviation contrail climate effects in the North Atlantic from 2016 to 2021**, *Atmos. Chem. Phys*., 22, 10 919–10 935, 10.5194/acp-22-10919-2022, 2022.

**Reply to Reviewer #2**

(Referee comment on "Correction of temperature and relative humidity biases in ERA5 by bivariate quantile mapping: Implications for contrail classification" by K. Wolf et al. (egusphere-2023-2356), https://doi.org/10.5194/egusphere-2023-2356-RC2, 2023)

We thank the Reviewer for the time she/he spent on the manuscript and for the useful comments. Addressing the comments has improved the manuscript.

For better legibility, the Reviewer's comments are highlighted in **bold** and changes in the manuscript are in *italics*.
* * *
**General Comments:**

**The authors compare ERA5 temperature and relative humidity at pressure levels of 175hPa to 300hPa with collocated IAGOS observations. A slight cold bias and a dry bias are found that peak at 200hPa and 250hPa, respectively. A bias correction method is applied that reduces the cold and dry bias. Finally, the impact of the correction on the probability of contrail formation is analyzed and compared with simpler corrections used in the literature. The probability of non-persistent and persistent contrail formation in uncorrected ERA5 data are found to agree quite well with the respective estimates from IAGOS data. This agreement improves very slightly for persistent contrail formation (correction reduces the probability of contrail formation) and deteriorates for non-persistent contrail formation after the correction of ERA5 data.**

**My main objection to the paper is that I am not convinced of the UT dry bias in ERA5 (see my point 1 below). I appreciate that the authors are careful to compare ERA5 and IAGOS data on a similar scale but I worry that the resulting (close to in-cloud) RH PDF (their figure 6) is unphysical. Nevertheless, as the authors point out, many studies in the literature have used a 'correction', often multiplying ERA5 RH with a constant factor. The factor is often determined by comparing ERA5 RH interpolated onto the aircraft track to IAGOS. The present study is designed to call into question such approaches and additionally shows that if the comparison between ERA5 and IAGOS is performed on a similar scale, the correction that can be inferred has close to no impact on contrail formation.**

We thank the Reviewer for this faithful summary of our work.

**Major comments:**

**1.)The ERA5 dry bias in extratropical UT RH:**

**1.1)The authors cite Dyroff et al. (2015) and Bland et al. (2021) as showing the LS moist and cold bias and Kunz et al., 2014; Gierens et al., 2020; Schumann et al., 2021 as presenting the UT underestimation of water vapor concentrations and ice supersaturation in ERA-interim and ERA5.**

**1.2) Kunz et al. (2014) analyzed ERA-interim and operational IFS data from the years 2001 to 2011 and finds a LS moist and UT dry bias. They note that the agreement with measurement data was improving within the analyzed time period due to model**

**updates. Bland et al. (2021) evaluate the ERA5 UT humidity fields using radiosonde data and find a slight moist bias or no bias depending on the type of radiosonde. Krüger et al. (2022) report a slight moist bias in the ERA5 UT when comparing to the active Differential Absorption Lidar (DIAL) WAter vapour and Lidar Experiment in Space (WALES; Wirth et al., 2009). This slight moist bias in water vapor mixing ratio coupled with the cold bias should lead to an UT moist bias in relative humidity (RH). Gierens et al., 2020 and Schumann et al., 2021 both compare ERA5 to MOZAIC / IAGOS RH data partly at relatively low resolution and interpolate ERA5 data to the flight track which leads to comparing observed and simulated RH at different resolutions.**

**1.3) Based on the above literature I am not convinced that there is an UT dry bias in ERA5.**

Subsequently, we address Reviewer 2's comments 1 through 1.3.

Obtaining upper tropospheric / lower stratospheric measurements is an inherently challenging task and inter-comparisons of different instruments and measurement principles exist. However, it is not so easy to compare different studies because the observations or the meteorological data (ERA-interim, IFS operational cycles, ERA5) or the method for the comparison (e.g., use of a tropopause-relative coordinate in Krüger et al. (2022) vs the use of pressure levels in our study) differ. To answer the question of the Reviewer, it is not clear whether ERA5 data is too dry or too moist for conditions close to saturation and/or high cloud cover. To our knowledge, there is no consensus as inter-comparisons lead to contradictory results. We modified the introduction in this regard in several places and would like to direct the Reviewer to the revised manuscript with tracked changes.

We referenced the studies by Kunz et al. (2014), Dyroff et al. (2015), Bland et al. (2021), and Gierens et al. (2020) to show previous attempts that evaluated the performance of ECMWF products (ERA-interim, and the operational forecast). In those studies various observations have been used, ranging from aircraft measurements from dedicated campaigns, radiosonde observations, to satellite products.

Indeed, the recent paper by Krüger et al. (2022) shows a wet bias in ERA5 on average on a wide range of meteorological conditions and seasons. However, Krüger et al. (2022) also found a minor wet bias below the thermal tropopause, which would support our results. Having said that, caution is needed when comparing our study to Krüger et al. (2022) because they use the thermal tropopause as the vertical reference, while our investigation uses fixed pressure levels. Variations in the altitude of the thermal tropopause mean that it is difficult to directly compare both studies. One also needs to consider potential uncertainties and biases in the LIDAR / WALES retrieved relative humidity that is presented in Krüger et al. (2022). Lidar is a remote sensing technique, and the retrieved relative humidity relies on temperature fields that are inferred from ERA5 reanalysis data (Groß et al., 2014), which itself could introduce a small bias.

In this regard we would like to mention the study from Reutter et al. (2020), who compared IAGOS with the ERA5 predecessor ERA-interim, and found a slight dry bias below the tropopause, which would also support our findings. They also found that a reduction in the ice-supersaturation threshold from the physical 100% (w.r.t liquid water) to 85% (lower stratosphere) or 95% (upper troposphere) – to compensate a bias at around 100% - lead to an improved estimation of the potential contrail formation.

Another sampling-specific factor that needs to be considered are targeted measurement campaigns. While Krüger et al. (2022) investigated a large variety of meteorological conditions, the investigated measurement campaigns still target specific weather patterns. This selected sampling might bias the data set to certain atmospheric and cloud conditions. A similar problem might influence the IAGOS data set, as pilots tend to avoid clouds during the flight (Petzold et al., 2020).

While we cannot rule out that IAGOS data themselves are biased, comparisons to more exact instruments do not indicate a bias. Here we refer to the validation study by Petzold et al. (2020), who compared CARIBIC and MOZAIC measurements (both predecessor of IAGOS) with dedicated humidity measurements. Petzold et al. (2020) demonstrated a good statistical agreement of CARIBIC and MOZAIC measurements with respect to the dedicated humidity measurements. They found only small differences around 100 % relative humidity, which they argue can be explained by differences in sampling (Petzold et al., 2020). A more recent study by Konjari et al (2022) showed that, from the instrumentation and data quality side, IAGOS measurements agree very well with other instruments' data, such as the airborne Fast In-situ Stratospheric Hygrometer (FISH, Meyer et al. 2015), in the upper troposphere. We also acknowledge that IAGOS is known to have a wet bias in the lower stratosphere and now explicitly mention this in the manuscript. Due to the agreement in the upper troposphere and in spite of the bias in the lower stratosphere, we argue that IAGOS data in the upper troposphere is reliable.

*"IAGOS measurements in the lower stratosphere that are typically characterized by low values of relative humidity (≈ rP1 < 20 %), are subject to a moist bias. This moist bias is a non-linear function of the relative humidity and requires a multi-dimensional regression correction that is currently under development (Konjari et al., 2022). Therefore, this known moist bias in IAGOS is not corrected in our analysis and it should be kept in mind that subsequent differences between ERA5 and IAGOS for low values of relative humidity may also be attributable to artifacts in the IAGOS measurements. However, since the focus of this analysis is to investigate contrail formation and persistence, only high values of relative humidity are relevant. Consequently, the moist bias for low relative humidity values in the IAGOS observations has little impact, on our analysis."*

In summary, there is a strong rationale for comparing and bias correcting ERA5 with IAGOS because: i) IAGOS data has shown to be reliable; ii) IAGOS samples temperature and relative humidity at exactly at the locations and pressure levels that are relevant for aviation studies; and iii) ERA5 is often used for predicting potential contrail formation.

**2.) When comparing IAGOS and ERA5 data you filter the IAGOS data because of their higher resolution. In order to determine the resolution of the comparison you estimate the resolution of the ERA5 data from the grid point distance of the Gaussian grid at the latitude of interest. In a spectral model, such as IFS, this is not the model resolution. The model resolution (grid length at the equator) of ERA5 is 31km. Even then the ERA5 grid box value is of course representative of the 3D volume and the IAGOS observation of a small subset of this volume so that ERA5 should be expected to display a lower variability.**

We partly agree with the Reviewer. We now mention that ERA5 is generated from a spectral model that internally operates with an approximate resolution of 31 km. Indeed, smoothing of the IAGOS along-track data lowers the natural variability and reduces extreme values, e.g., low and high relative humidity. However, the applied smoothing

modifies the calculated mean values only slightly. The text sections have been rephrased and now read as follows:

*"The fixed (Cartesian) grid resolution of 0.25° of ERA5 does not correspond to a constant longitudinal grid box size in km which instead depends on the latitude. Considering the three sub-domains between 30°N and 70°N, the spatial resolution of one ERA5 grid-box ranges between 24 km (30°N) and 14 km (70°N). Therefore, we assume an average grid box size of 19 km. However, it is emphasized that ERA5 is a spectral model with an internal Gaussian resolution of around 31 km and, thus, the effective resolution is coarser than the Cartesian grid resolution (Hersbach et al. (2020))."*

*"Smoothing the IAGOS data, as explained in Section 2.2, leads to mean values of $T_{P1}$ and $r_{P1,ice}$ for the native and the smoothed data that are similar by 0.1°C and 1 %, respectively. As the smoothing did not change the mean values significantly, the differences in the PDFs of ERA5 and IAGOS, as well as the bias in mean $r_{ERA,ice}$ compared to $r_{P1,ice}$ cannot be attributed to differences in the spatial resolutions. However, the smoothing of the IAGOS data leads to a reduction in the variability as well as in the extreme values in measured $T_{P1}$ and $r_{P1,ice}$ (not shown here)."*

It should be noted that we use ERA5 data that is interpolated from the model levels onto pressure levels (e.g. 250 hPa). We do not think it is appropriate to say that temperature and humidity data are representative of a volume. In this case the vertical resolution is not the same as the vertical sampling. However we compare measurements made in a given pressure layer centered onto a pressure level to this particular pressure level.

To further answer the question about the impact of spatial averaging of IAGOS data, we present IAGOS data at the original 4-s resolution and temporally smoothed data (60 s) in the plots below. The obtained distributions change little between the original (top) and the smoothed (bottom) data and, thus, we argue that averaging IAGOS does not have a significant impact on the obtained statistics.

[Figure]

[Figure]

1 min resolution

**3.) The only reason given for the dry bias in ERA5 RH is saturation adjustment within clouds (section 2.2.1 and in lines 278-279). If this is your hypothesis then I would suggest using the IAGOS particle number concentration N_ice and splitting up the data set into cloudy and cloud free measurements and doing the same for ERA5 (using cloud cover) and then comparing the RH CDF for cloudy and cloud-free instances separately. The comparison could also be extended to include the data of Krämer et al (2009, 2020) that show an increased probability at RH = 100% (unlike IAGOS data). Corrections could and should be done for cloudy cases only.**

**3.1) You do something similar in section 3.2 but sort both ERA5 and IAGOS data dependent on ERA5 cloudiness into in-cloud and cloud-free data. Since you do not check in figure 6 whether IAGOS measurements are representative of cloudy or cloud-free conditions (see also my comment 4) the differences between ERA5 and IAGOS PDFs worsen with increasing ERA5 cloud cover. Since you have calculated already a 'IAGOS cloud cover' it is not clear to me why you use it only in order to correlate the IAGOS and ERA cloud cover and don't do the IAGOS in-cloud with ERA in-cloud comparison.**

**3.2) Nevertheless, it is also not clear to me why a correction of in-cloud RH is needed since contrail formation studies are mainly interested in contrail formation within cloud free air.**

In this section, we answer comments 3 through 3.2.

Intrigued by the suggestion of Reviewer 2, we performed additional analysis and modified Figure 6 and its corresponding discussion in Section 3.2 "Distribution of relative humidity under cloud-free and in-cloud conditions".

We now separate ERA5 simulations by cloud fraction (CF), where CF < 0.2 are considered as cloud-free, 0.2<CF<0.8 as intermediate, and 0.8<CF as cloudy or in-cloud. Similarly, the IAGOS observations are separated for cloud-free, intermediate, and in-cloud measurements using the detection threshold for cloud particle number concentration given by Petzold et al. (2017).

We replaced the old figure with a new plot that is also given here. Filtering for cloud-free conditions results in similar PDF for ERA5, corrected ERA5, and IAGOS measurement for RHi< 90%. Larger RHi are observed much more frequently compared to ERA5. The QM-correction partly enhances RHi such that the PDF approaches the one from IAGOS. For intermediate cloud conditions the distributions form ERA5 and QM-corrected ERA5 are similar, however, there is a better agreement of ERA5 QM-corrected data with the IAGOS observations. Both, ERA5 and ERA5 QM-corrected data miss the variability towards RHi< 80%. For data points considered to be in-cloud, the PDF of IAGOS is much broader compared to ERA5 and ERA5 QM-corrected data. The QM-correction broadens the distribution of enhancing the right wing of the distribution towards larger RHi values. Considering the three different categories, there is an improvement in the statistical distribution of RHi after the QM-correction. Particularly important is the improvement in RH representation in the intermediate category, where clouds are most likely to form.

The newly added analysis is described in section "Distribution of relative humidity under cloud-free and in-cloud conditions". Due to the length of the revised section, we refer the Reviewer to the revised manuscript with track changes.

[Figure]

*Figure 6. a–c Probability density functions of relative humidity rice w.r.t ice (in %) from IAGOS (black), original ERA5 (red), and corrected.*

**4.) I believe there could be another reason for differences in the PDF of RH between IAGOS and ERA5 which is connected with sampling. Petzold et al. (2020) discuss sampling issues in their figure 5 when comparing MOZAIC measurements with measurements from research flights (Krämer et al. 2009). They say that the reason why the research flights show a much higher probability of RH around 100% than IAGOS is because campaign measurements often target clouds in which RH is often around 100%. In the same way, it is likely that IAGOS pilots tend to avoid clouds and rather fly through cloud free air or very thin clouds while ERA5 data represent cloudy, partly cloudy and cloud free situations purely based on their probability of occurrence.**

We agree with the Reviewer. In addition to mentioning the sampling issue in the introduction and the IAGOS data section, we further highlight this potential shortcoming later in the text.

The following text was added to the introduction.
*"In situ measurement campaigns are a potential fourth approach, during which contrails are directly probed and contrail properties are investigated. Dedicated measurement campaigns, for instance by Krämer et al. (2009, 2020) and Voigt et al. (2017), are rare.*

*Furthermore, they may lack spatial representation by targeting specific atmospheric features as well as cloud conditions, which may bias the results (Petzold et al., 2020). "*

The following text was added to the section "In-service Aircraft for a Global Observing System":
*"In addition, the sampling is biased, i.e., by avoiding severe weather and by avoiding or favoring specific atmospheric circulation patterns, such as the jet stream (Petzold et al. (2020))."*

**5.) Figure 6 shows that the ERA QM data display a maximum probability of RH of ~105% for close to full cloud cover and of ~115% for cloud cover of between 0.6 and 0.8. Uncorrected ERA data show that RH = 100% has the highest probability in cloudy situations in line with Krämer et al. 2009 and 2020 and in line with theory that predicts significantly large in-cloud supersaturation only if supersaturation forcing is very large or if ice crystal number concentrations are very small. Both conditions are not the most probable inside clouds. To be fair, the other methods that you mention within your paper, e.g. scaling up RH with a constant factor, will have exactly the same (or possibly a worse) impact on in-cloud RH.**

To answer this comment, we would like to direct the Reviewer to the answers to comment 3 and the revised Section 3.2.

While it might be true that such conditions are rare, the conclusions given in Krämer et al. (2009, 2020) are based on sophisticated measurements in terms of instrumentation and accuracy. In contrast, IAGOS are routine measurements onboard passenger aircraft that are reliable but cannot achieve the same accuracy. They sometimes provide information on particle number but only some IAGOS aircraft were equipped with particle counters. Selecting only those measurements where the particle number concentration is provided would drastically reduce the available dataset. The advantage of IAGOS data is the large number of measurements and the relatively uniform spatial distribution, considering the constraints given by Petzold et al. (2017).

**6.) I am surprised that the averaging of IAGOS data does not change the RH PDF. Averaging should always decrease the variability of data and I would expect the probability of very high ice supersaturation to be reduced. Since this result is used as an argument to claim that mixing ratios are too low I think the figure should be included. Note also that the result that averaging of IAGOS does not improve the comparability to IAGOS data is in contradiction to Reutter et al. (2020) who finds that the IAGOS data show a very high percentage of small-scale ice supersaturated areas which ERA-interim cannot resolve but that ERA-interim and IAGOS fit well once the IAGOS resolution is reduced to the resolution of ERA-interim.**

To answer this question, we would like to direct the Reviewer to our answer to major comment 2 and the provided figure. The presented figure shows distributions of relative humidity for the raw measurements with 4-s temporal resolution as well as distributions based on 1-minute averages. Even after averaging over such a long period the distributions did not change significantly. There is an exception for extreme values but they contribute only a very small fraction to the total number of samples.

**7.) Despite the fact that I am not convinced of the need to 'correct' ERA RH data, I still think that your work points out an important point in relation to the earlier attempts to study contrail formation using ERA5 data. You show that if you take**

**care to compare ERA5 and IAGOS data on a similar scale that correction has hardly any impact on the contrail formation probability. This means that the ERA5 RH scaling of earlier studies that is calculated by interpolating ERA5 data to the aircraft track and comparing to IAGOS data, lead to an increase in contrail formation probability that is not supported by IAGOS data.**

We thank the Reviewer for this comment. Concerning the bias in ERA5 we would like to direct the Reviewer to our answers to comments 1 and 3.

**Minor:**

**1.) The text is generally difficult to read because it is often not clear which figure is being discussed. E.g. at the beginning of section 3.1 you mention fig. 2 and fig. 5e in the first 10 lines but actually you are discussing probably figure 3 which you fail to say. This problem is repeated in other places.**

We partly agree with the Reviewer. The reference to Fig. 2 is correct. In the sentence before the reference we discuss the number of flights / measurements per pressure layer. The distribution of flights per pressure level is shown and indicated in Fig. 2.
The reference to Fig. 5e is also correct as we discussed the reduction in the mean deviation of temperature, which is given in Fig 5e. However, we added an additional reference to Fig 3. The sentence now reads the following:

"*[...] distributions of $T_{P1}$ (see second column in Fig. 3 and Fig. 5e).*"

Several further annotations have been made in the text by adding references to direct the reader to the respective figures.

**2.) line5 (abstract): You extract wind speed but I can't find the place where you use this variable.**

The Reviewer is right. "Wind speed" accidentally remained from an earlier revision of the text. Wind speed has been removed and the sentence reads as following:

*"IAGOS flight trajectories are used to extract co-located meteorological conditions from ERA5, namely temperature and relative humidity, which are compared with the IAGOS measurements."*

**3.) Line 31: You did not define 'WC'.**

"WC" was a typo and is now corrected. The correct abbreviation is WV, which is defined in the line above.

**4.) Line 59: Bickel et al. (2020) discusses the difference between radiative forcing and effective radiative forcing. The model that is used in Bickel et al. is described in Bock and Burkhardt (2016).**

We thank the Reviewer for pointing this out and we took the suggestion to better explain the different definitions of radiative forcing earlier in the manuscript. The revised and new section now reads:

*"[....]The influence of a perturbation, e.g., clouds, aerosols, or gases, on the Earth's atmosphere and its radiative transfer is quantified by the radiative forcing (RF). By definition, RF is defined as the difference in the net irradiance at the top of atmosphere under perturbed and unperturbed conditions (Ramanathan et al., (1989)). In the context of climate studies the RF is understood as the difference of the Earth energy budget due to a contributor to climate change (Bickel et al., 2020). For example, the aviation-induced global $CO_2$-related RF is estimated to be around 30 $mWm^{-2}$ (Boucher et al., 2021). Contrail RF is estimated to be stronger, at about 60 $mWm^{-2}$ but is subject to much larger uncertainties (Burkhardt and Kärcher, 2011)."*

**5.) Line 67: Instead of the word 'distribution' you mean probably 'variability'.**

The Reviewer is correct and the words have been exchanged. The sentence now reads:

*"[...] which is generally challenging due to the high temporal and spatial variability of WV."*

**6.) Line 68 ff: Please include Krüger et al (2021) and improve the description of findings of the papers (see above).**

We revised the section of the text and added the reference to Krüger et al. (2022). The section has been modified to the following:

*"Specific issues have been identified in the upper troposphere and lower stratosphere, as well as with the general representation of ice supersaturation. For example, Bland et al. (2021) compared radiosonde observations with operational ECMWF IFS weather forecast and identified a lower stratosphere moist bias. Similarly, Krüger et al. (2022) compared measurements from a differential absorption Lidar with ECMWF ERA5-reanalysis data (on a relative-tropopause coordinate) and identified a slight moist bias in the upper troposphere. A moderate to significant moist bias was found in the lower stratosphere. Contrarily, studies that compared water vapor concentrations and ice supersaturation in ERA-interim and ERA5 with aircraft in-situ observations found that ice supersaturation is insufficiently represented (in frequency and magnitude) in those re-analysis products (Kunz et al., 2014; Dyroff et al., 2015; Gierens et al., 2020; Reutter et al., 2020; Schumann et al., 2021). Consequently, there is no consensus whether ECMWF re-analysis products are subject to a moist or dry bias in the upper troposphere."*

**7.) Line 75 ff: You may want to include Krämer et al. (2009, 2020)**

We followed the suggestion of the Reviewer and added the two references to the manuscript.

**8.) Line 196: The multiplication of ERA5 RH with a factor is only common to the studies that you cite afterwards and not generally common.**

We followed the Reviewer's comment and made this point clearer by highlighting that scaling is primarily applied in studies from, e.g., Schumann et al. (2013) or Schumann et al. (2015), and is not common practice.

*"To compensate for the dry bias in ERA5 for contrail detection applications, $r_{ERA,ice}$ values are scaled sometimes in some studies by multiplication factors between 0.8 or 0.9, particularly in Schumann and Graf (2013) and Schumann et al. (2015)."*

**9.) Line 276-277: I am not sure what you are talking about. I don't see any rectangular shapes.**

Thank you for mentioning this. The word rectangular was replaced by triangular and the text now reads:

*„The PDF of $r_{ERA}$,ice close to 100 % is characterized by a triangular shape, while the distribution of [...]"*

**10.) Line 340-341: 'r_ice close to 100% are likely associated with cloud formation' – no this is not the case. Cirrus formation happens at high relative humidity relative to ice.**

This section the sentence has been removed during revisions.

**11.) Line 366-367: 'All data points that do not belong to any of the categories are flagged for no contrail formation (NoC).' Isn't you R category also no contrail formation?**

Agreed. We removed the word "formation" from the sentence that was highlighted by the Reviewer. The sentence now reads:

*"The SAc and the ISS threshold are used to flag the IAGOS measurements and the along-track ERA5 for NPC, PC, and R conditions. Samples that belong to none of these three categories are flagged as "no contrails" (NoC)."*

**12.) Line 395-396: please reformulate the sentence 'but the saturation is insufficient to reach supersaturation to …'**

In the course of the revision of the manuscript this sentence has been removed.

**13.) Line 671-672: 'The mode in Fm,h is primarily caused by the saturation adjustment in ERA5 (see Sec. 2.2.1)'. As said above, Krämer et al shows that the mode around 100% RH is a naturally occurring mode and not a modelling feature.**

We agree and have rephrased the sentence to the following.

*"The mode in Fm,h is a superposition of two effects. While the peak is of natural origin, as reported by Krämer et al. (2016, 2020), it is also caused by the saturation adjustment in ERA5 (see Section 2.2.1)."*

**References**
Groß, S. / Wirth, M. / Schäfler, A. / Fix, A. / Kaufmann, S. / Voigt, C. , **Potential of airborne lidar measurements for cirrus cloud studies**, 2014, *Atmos. Meas. Tech.* , Vol. 7, No. 8, p. 2745-2755

Meyer, J. / Rolf, C. / Schiller, C. / Rohs, S. / Spelten, N. / Afchine, A. / Zöger, M. / Sitnikov, N. / Thornberry, T. D. / Rollins, A. W. / Bozóki, Z. / Tátrai, D. / Ebert, V. / Kühnreich, B. / Mackrodt, P. / Möhler, O. / Saathoff, H. / Rosenlof, K. H. / Krämer, M., **Two decades of water vapor measurements with the FISH fluorescence hygrometer: a review**, 2015, *Atmos. Chem. Phys.* , Vol. 15, No. 14, p. 8521-8538

---

## Author Response (AR2)

**Reply to Reviewer #2 (2nd round)**
(Referee comment on "Correction of temperature and relative humidity biases in ERA5 by bivariate quantile mapping: Implications for contrail classification" by K. Wolf et al. (egusphere-2023-2356), 2023)

We thank the Reviewer for the time she/he spent on the manuscript and for the useful comments. Addressing the comments has further improved the manuscript.

For better legibility, the Reviewer's comments are highlighted in **bold** and changes in the manuscript are in *italics*.
* * *
Before addressing the specific comments of Reviewer 2 we would like to highlight that we have decided to make a small but important change to the title of the manuscript. The previous title "*Correction of temperature and relative humidity biases in ERA5 by bivariate quantile mapping: Implications for contrail classification*" has been changed to "*Correction of temperature and relative humidity biases in ERA5 by bivariate quantile mapping for contrail prediction and classification*". With this title change we want to clarify that we do not want to make a universally applicable correction of humidity in ERA5 but only provide a corrected humidity to enable better estimates of contrail occurrence consistently with IAGOS measurements.

**I very much appreciate the considerable effort by the authors to improve the paper. Nevertheless, there are a few points that should be considered in order to improve the manuscript even further.**

- **I appreciate the modifications of the text connected with the question if ERA5 has a dry bias. We agree that 'there is no consensus whether ECMWF re-analysis products are subject to a moist or dry bias in the upper troposphere'. I am happy with the changes made in the introduction. But I am surprised that the rest of the paper completely ignores those important changes. It is not acceptable to first say that there is no consensus on a moisture bias and then talk in the remainder of the paper about the moisture bias and how it can be corrected.**

  As noted in our responses to Reviewer 2 (1st Round), there is no consensus in the literature as to whether ERA5 is generally too dry or too moist in the UTLS. However, by analyzing the IAGOS data we found a dry bias in ERA5 with respect to the IAGOS measurements. Since IAGOS has been found to be in good agreement against more accurate in situ measurements, there is a strong basis to say that ERA5 has a dry bias in the regions (locations and pressure levels) sampled by IAGOS aircraft, and by extension in the corresponding air traffic regions. Here, and as already stated in the first round of replies, we are particularly interested in correcting ERA5 temperature and relative humidity for predicting potential contrail formation and relying on IAGOS data allows to do so exactly at the locations and pressure levels that are relevant for aviation studies. We have added an additional sentence to justify our approach and the rationale for correction ERA5 against IAGOS. The additional sentence reads as:

  "*It is important to stress that we do not seek to make a universally applicable correction of humidity in ERA5 but rather provide a corrected humidity to enable better estimates of contrail occurrence. Relying on IAGOS data allows us to do so*

*exactly at the locations and pressure levels that are relevant for aviation studies."*

To provide further clarification, qualifiers have been added to manuscript to explicitly indicate that the identified biases in temperature and RH are in relation to IAGOS.

- **Furthermore, from your answers to my comments I gather that we agree that pilots appear to avoid clouds. This would lead to IAGOS sampling clouds with close to 100% RH less often and, accordingly, higher RH values more often. This would mean that a certain bias between ERA5 and IAGOS needs to be expected. I have not seen text connected with this within the paper – I am sorry if I overlook connected text. There are several places within the paper where the impact of avoiding to fly through clouds should be discussed (but I did not find it mentioned). This argument should be included in the introduction right after your sentence 'Contrarily, studies that compared water vapor concentrations and ….. in ERA … with aircraft in-situ observations ….. This should also be mentioned in the results section e.g. when comparing the PDF of ERA5 and IAGOS (lines 336-338) and in section 3.2.**

We agree that pilots tend to avoid deep convective clouds (such as cumulonimbus) and other clouds that are potentially indicative of dangerous weather and/or turbulence. However, cirrus clouds are typically not avoided because they do not pose a threat to flight safety. More importantly, cirrus clouds are often optically thin and barely visible or even invisible to the pilots, when viewed from flight altitude. Therefore, we are not convinced that a potential sampling issue with respect to cirrus clouds plays a role in our diagnosis of a dry bias of ERA5 against IAGOS data. Even if cloud avoidance was introducing a dry bias, we do not need to correct for it because our objective is to go from ERA5 humidity to humidity as sampled by commercial aircraft. It is not obvious to us that cloud avoidance would lead to higher RH values, as suggested by the Reviewer. Even if cirrus cloud were subsaturated more often than they are supersaturated, which is not a given (e.g., see Li et al (2023)), sampling supersaturated cloud free areas is also infrequent.

Even though we raised awareness for the sampling issue, we added the following sentence to the introduction at the position suggested by the Reviewer.

*"[...]Contrarily, studies that compared water vapor concentrations and ice supersaturation in ERA-interim and ERA5 with aircraft in-situ observations found that conditions of ice supersaturation are not frequent enough in those reanalysis products, suggesting a dry bias (Kunz et al., 2014; Dyroff et al., 2015; Gierens et al., 2020; Reutter et al., 2020; Schumann et al., 2021). Consequently, there is no consensus whether ECMWF re-analysis products are subject to a moist or dry bias in the upper troposphere. It is noted that in-situ observations are potentially biased by avoiding deep-convective clouds and the outflow of such clouds. However, cirrus clouds are typically not avoided (Petzold et al., 2020) and, therefore, a potential sampling issue with respect to cirrus clouds plays more than a minor role"*

The following sentence was added to the second paragraph in the summary:

*"It is again noted that in-situ observations from IAGOS are potentially biased by*

*avoiding deep-convective clouds and their outflow, while cirrus clouds are generally not avoided."*

- **The PDFs of RH (figure 3) are compared and the 'lack of ISSR in ERA5' is suggested to be connected with the use of saturation adjustement within IFS (line 341) which is explained in section 2.2.1. The whole section 2.2.1 is dedicated to 'in cloud ice supersaturation'. The possibility of a lack of ISS coming from cloud-free areas appears to not be discussed. As far as I can see figure 3 does not contain any information on if the difference in ERA5 and IAGOS RH comes from cloudy or cloud-free areas. Nevertheless, it is discussed as a problem stemming from the cloudy areas. In section 2.2.1 the authors write that ' they (the models) currently lack in the appropriate representation of ISS under cloudy conditions'. You do not mention that the models furthermore lack in the appropriate representation of subsaturation under cloudy conditions due to the saturation adjustment. In the last little while in-situ measurements pointed at contrails often persisting in ice subsaturated air. The analysis of in-situ measurements from Krämer et al 2009 show a large probability of subsaturation in cloud free air. Dekoutsidis et al. 2023 show that the most probable relative humidity within midlatitude cirrus clouds is 96% and only 34.1% of in-cloud RH values are supersaturated. Given those analysis I am surprised that in section 2.2.1 it isn't even mentioned that in-cloud subsaturation is underestimated as well. So, I doubt that the peak at 100% humidity needs shifting over to higher ISS. Based on Dekoutsidis et al. you may want to claim that it needs to be shifted to RH=96%. But I think 100% is not too bad an approximation.**

We agree with the Reviewer that cirrus clouds may be subsaturated or supersaturated given the slow kinetics of condensation on or evaporation from ice crystals. Neither process is well represented in large-scale models. Moreover, ERA5 does not aim for an accurate representation of relative humidity within clouds. Instead, it provides information regarding whether a grid-boy is cloud-free or cloudy, as well as the cloud coverage. Additionally, (relative) humidity is generally complicated to determine as it is influenced by many processes, such as atmospheric dynamics and temperature fields. The objective of our study is not to identify and quantify the individual sources of humidity errors in ERA5.

Section 2.2.1 is dedicated to the discussion of in-cloud representation of supersaturation in ERA5 as the in-cloud saturation is technically clipped to 100% relative humidity. Clipping to 100% is a necessary workaround, as the temporal evolution of in-cloud humidity is not tracked within ERA5. With our proposed QM technique we want to remove the effects of the RH clipping concerning contrail estimation.

A discussion of the distributions and PDFs of humidity inside and outside of clouds as simulated by ERA5 and measured by IAGOS is given 3.2, where we actually do see a good agreement for in RH between IAGOS and ERA5 for intermediate cloud conditions.
Furthermore, we would like to direct the Reviewer to Fig. 6 in the paper by Sanogo et al (2024). Figure 6 in their paper shows the distributions of RHi as measured by IAGOS. RHi are separated by threshold of ice particle number concentration, which is used as proxies to differentiate between measurements in clouds and outside of

clouds. Their paper includes a detailed discussion of RHi determined from in-cloud and cloud free measurements.

- **In line 354-355 you say that smoothing leads to a reduction in extreme values in RH. Comparing grid box mean values and in-situ observations therefore will have very different extrema and differences should not be interpreted as a dry bias. Ignoring this difference means that you disregard the subgrid variability of humidity within IFS. We know that this is a bad approximation.**

We agree with the Reviewer that smoothing leads to a reduction in extreme RH values. However it does not affect the mean value and therefore has no impact on a dry or wet bias. Furthermore we do compare RH at comparable resolution between the model and the observations. We are not sure about the relevance of this comment and, as a consequence, no change was made to the text.

- **The analysis in figure 6 is interesting. But the analysis cannot fully support the conclusions:**

  **a) In the introduction it was mentioned that IAGOS pilots often choose to fly outside of clouds which leads to a sampling issue when comparing ERA5 and IAGOS RH data. I am missing a discussion of the impact of this sampling issue in the discussion connected with figure 6. E.g. in the RH PDFs for lower cloud cover including the RH from cloudy patches vs not including them makes a difference of the RH PDF and ISSR estimates.**

As stated above, pilots typically avoid deep convective clouds, mesoscale convective systems, and the outflow thereof. However, cirrus clouds are typically not avoided as they do not threat flight safety. More important,  cirrus clouds are often optically thin so that they are barely or not visible when viewed from flight altitude. Therefore, we have no evidence that the sampling issue with respect to cirrus clouds plays more than a minor role.

  **b) I am also missing a discussion of the surprising RH PDF from IAGOS for full cloud coverage. Do we really expect values of above 150% in the main air traffic areas and do we really expect them within cirrus? Aren't values like that a tropical UT issue? I expect to see RH > 140% extremely seldomly when sampling fresh nucleation events. The frequency with which we see 140% is only one order of magnitude lower than seeing 100%? I think it would be good to compare the results to Dekoutsidis et al. 2023.**

Our manuscript does not address the investigation of supersaturated or sub-saturated cirrus. IAGOS measurements have been successfully validated against other high-precision measurements, for example, the Fast In-situ Stratospheric Hygrometer (FISH). However, due to the response time of IAGOS observations they are not very well suited to investigate humidity fluctuations and humidity gradients between cloudy and cloud-free air masses. The response time was already explained and quantified in the first version of the manuscript. As our objective is to

validate ERA5, a large-scale model with an average spatial resolution of approximately 19 km, we do argue that IAGOS data are still suited for this kind of analysis.

In response to the Reviewer, we prepared a plot to compare with Dekoutsidis et al. (2023). The appended plot (see below) shows the density distribution with temperature on the x-axis and RHi on the y-axis. The measurements are separated for measurements inside and outside of clouds. The transition between both is called "intermediate". Cloud-free conditions are defined by ice crystal number concentrations N lesser than 0.001 cm$^{-3}$, while cloudy conditions are defined by N > 0.015 cm$^{-3}$. Intermediate conditions are defined by 0.001 cm$^{-3}$ < N < 0.015 cm$^{-3}$. The x and y-axis are scaled such that they use the same value ranges as in Dekoutsidis et al. (2023). A quantitative analysis of our plot indicates that the distributions observed are similar to the distributions reported by Dekoutsidis et al. (2023). Other than stated by the Reviewer we do see a differences by two orders of magnitude between RHi at 100 % and RHi at 140 %.

[Figure]

*Figure 1: Density distributions with temperature on the x-axis and relative humidity (w.r.t. ice) on the y-axis.*

Again, we would also like to direct the Reviewer to Fig. 6 in the paper from Sanogo et al (2024). Figure 6 in their paper shows distributions of RHi measured by IAGOS. RHi are separates for different thresholds of ice particle number concentration that are used as proxies to separate between measurements in clouds and outside of clouds. Their paper includes a extensive discussion of Rhi of in-cloud and cloud free measurements.

**c) A short sentence about the stability of the in-cloud RH PDF based on less than 1% of observational data would be helpful.**

We have added a sentence to say that the in-cloud RH PDF is less robust because of the smaller amount of data.

*"The relatively limited number of samples (<1 %) from within clouds causes a less robust PDF compared to the PDF based on measurements conducted outside of clouds."*

- **As said above, the summary should also include the open issues and their possible impact on the conclusions. Is there actually a dry bias or do we just correct RH to have a more convenient estimate for contrail formation? What does the sampling problem, stemming from pilots avoiding clouds, mean for**

**the analysis? How good is our knowledge of in-cloud RH? Can we really call the corrections improvements or are they only improvements when trying to find the suitable input parameters for estimating contrail formation.**

Please see our reply above.

Following the remarks of the Reviewer, several modifications have been made to the text. The individual lines are copied below. We also direct the Reviewer to the provided track-changes file:

*"The QM method allows the removal of biases based on the statistical distributions of an observed and modeled quantity, for example temperature and relative humidity, with the aim to better estimate the contrail formation potential in air traffic regions."*

*"In this study we proposed a temperature and relative humidity correction method for ERA5 based on a bivariate quantile mapping (QM) technique to better estimate the contrail formation potential."*

- **Line 19: It should say 'for non-persistent and persistent' and not the other way round.**

  The Reviewer is correct. The text has been corrected to read as follows:

  *"The original ERA5 analyses show corresponding numbers of 50.3% and 7.9% for non-persistent and persistent contrails, respectively"*

- **Line 26: The sentence is not clear: Are you talking about the remaining bias after the bias correction? Or are you saying that regionally there are still some deviations?**

  The sentence has been rephrased to the following:

  *"Despite this improvement, differences in contrail occurrence persist after the correction, which are traced back to the underlying biases in temperature and relative humidity, as well as to the non-linearities in the Schmidt-Appleman criterion."*

- **Line 45 + 47: It is not clear to me why Lee et al is not cited anymore. The citation would fit perfectly.**

  In the first round of reviews, Reviewer 2 argued (see Minor issues number 1) that Lee et al. 2021 present ERF values instead of RF values.
  Original comment: *"1.) Lines 34-38: I suggest that you state that RF is a global (or at least regional) quantity, averaged over a long time period. On first reading, it was not so clear to me whether you refer to single contrails or contrails in general. Furthermore, aren't the quoted values ERF values (in Lee et al.) rather than RF values?"*
  Our response to the comment: *"To be consistent within the text, we removed the citation from Lee et al. (2021) and only kept the references from Boucher et al. (2021) and Burkhardt and Kärcher (2011), who give estimates for the radiative*

*forcing."*
As mentioned in the first round of replies and above, we want to be consistent in the compared values.

- **Line 72: You cite Bickel et al. when talking about the implementation of a contrail scheme within a climate model. Bickel et al does not include a description of the modelling approach. The description can be instead found in Bock and Burkhardt (2016).**

  We thank the Reviewer for this comment. Following the Reviewer's suggestion, we changed the citation to Bock and Burkhardt (2016).

Li, Y., Mahnke, C., Rohs, S., Bundke, U., Spelten, N., Dekoutsidis, G., Groß, S., /Voigt, C., Schumann, U., Petzold, A., Krämer, M. Upper-tropospheric slightly ice-subsaturated regions: frequency of occurrence and statistical evidence for the appearance of contrail cirrus, 2023, Atmos. Chem. Phys. , Vol. 23, No. 3, p. 2251-2271

Sanogo, S., Boucher, O., Bellouin, N., Borella, A., Wolf, K., Rohs, S., Variability in the properties of the distribution of the relative humidity with respect to ice: implications for contrail formation, 2024, Atmos. Chem. Phys. , Vol. 24, No. 9, p. 5495-5511

---

## Author Response (AR3)

**Reply to Reviewer #1 (3rd round)**
(Referee comment on "Correction of temperature and relative humidity biases in ERA5 by bivariate quantile mapping: Implications for contrail classification" by K. Wolf et al. (egusphere-2023-2356), 2023)

We thank the Reviewer for the time she/he spent on the manuscript and for the useful comments. Addressing the comments has further improved the manuscript.

For better legibility, the Reviewer's comments are highlighted in **bold**, comments from our side are in normal font, and changes in the manuscript are in *italics*.
* * *
**This is my first review of "Correction of temperature and relative humidity biases in ERA5 by bivariate quantile mapping: Implications for contrail classification" by Kevin Wolf et al. The authors present a thorough analysis of a correction of ERA5 relative humidity and temperature using a quantile mapping approach, with the aim to obtain an improved assessment of the contrail formation potential. The presented figures are nice and clear and thoroughly explained. Especially the illustration of the effect on the contrail formation potential based on the Schmidt-Appleman criterion is well-made.**
**After the preceding rounds of reviews the content of the paper is in good shape. However, I have a few comments that I hope can help the authors to improve their manuscript and, especially, to make it easier to access the relevance of the work.**

**1) I think you should better clarify the purpose of the work and add more information about the added value. In some places this information is simply missing (abstract, summary) and in others unprecise formulations caused confusion (introduction).**
**The first sentence of the abstract implies a different topic to be addressed. In combination with the word "prediction" in the title, I anticipated that forecast skill of T and RH is addressed (see also comments below), which is not the case. Only a few lines later the topic shifts to contrails. I recommend better introducing the key topic of this work. After that, partly redundant details are linked without a conclusion. So, I suggest better connecting these sentences and adding a precise conclusion about the added value of this work, which I believe is a more reliable identification of the contrail regions using post-processed ERA5 T and TH fields.**
We agree with the Reviewer. We have restructured and rephrased the abstract to avoid giving the impression that we are trying to correct ERA5 and state clearly that we instead are post-processing the ERA5 temperature and relative humidity fields to obtain a better estimate of potential contrail formation.
We also had to significantly reduce the length of the abstract to meet ACP's word limit of 250 words.
Due to the comprehensive changes, we would like to direct the Reviewer to the track-changes version of the revised manuscript to review the modifications.

**The title "correction (…) in ERA5" may imply that ERA5 is improved within the model, which is not the case. It is rather a post processing with the aim to improve temperature and rHi to better analyse the potential of contrail formation. In addition, I struggle with the word "prediction" as no forecast data are used in this study. I would be careful when using "prediction" in the context of NWP. Hence, I would suggest changing the title to e.g. "Correction of ERA5 temperature and relative**

**humidity biases (…) for contrail analysis and classification”.**
We followed the suggestion of the Reviewer and changed the title to the following:
*"Correction of ERA5 temperature and relative humidity biases by bivariate quantile mapping for contrail formation analysis."*

**In the introduction a clear introduction of the approach and goals are missing. Different approaches to investigate contrail formation/impact are listed and mixed with the explanation of the chosen approach. Especially after L62 the structure is not clear: Approach three on contrail modeling is followed by ERA5 and its biases. The fourth approach, which starts with campaign/IAGOS measurements, is abruptly followed by ERA5 correction attempts (L92 "Comparing and bias correcting"). I do not understand the causality to i, ii, and iii. Additionally, both approaches overlap somehow. The introduction contains all information, but should be better structured and more precisely formulated with a focus on your own work and open questions. For what is the correction actually needed?**
The Reviewer is right and we restructured the introduction to be clearer. We also have refrained from mentioning different approaches as it is better to separate into measurements (in-situ and remote) and simulations by models.
We also added a section at the end of the introduction to better articulate the goal of the manuscript and the application of the QM-correction. We would also like to direct the Reviewer to the track-changes version of the revised manuscript for a full overview of the modifications.
The added text reads as following:
*"The QM-correction aims to remove possible temperature and humidity biases in ERA5 post-processed data to better estimate the contrail formation potential beyond the common locations of the IAGOS flight tracks. The advantage of such a correction is that ERA5 data away from the IAGOS flight tracks can be used to estimate the large-scale contrail formation potential, thus providing a broader perspective on potential contrail formation in space and time over the Atlantic. Potential applications include the study of temporal and spatial patterns of contrail formation, and the development of rerouting options based on statistical distributions of contrail formation potential."*

**In the summary section, I wonder about the added value of the presented corrected ERA5 data and the applied methods? Can the method be used to correct grid points away from IAGOS flight routes and to actual prediction data in order to support flight routing to avoid contrail formation? What is the added value to the IAGOS observations alone? Who could make use of such data? What are the ways forward?**
We do agree with the Reviewer and added the following sentences to the summary:
*"[…] Overall, the presented QM–correction allows to remove the systematic bias in temperature and relative humidity in ERA5 using IAGOS as a reference. Therefore, the method can be applied to ERA5 data to estimate the contrail formation potential away from IAGOS flight tracks under the constraint that the correction is applied to grid points within a specified domain between 105°W and 30°E and from 30°N to 70°N. This provides a broader perspective on potential contrail formation in space and time over the Atlantic region. This allows to study of temporal and spatial patterns of contrail formation over the North Atlantic region to develop statistically based rerouting options."*

**2) I think that the ERA5 reanalysis and applied methods could be explained more clearly:**
**In the introduction (L62ff) the explanation of ERA5 is unprecise (e.g. "the prediction and reanalysis"). ERA5 is based on a particular version of the IFS that is kept constant over the time period from 1940. The constant version including the data assimilation system allows trends to be better accessed and separated from model changes. Make sure that the reader understands the difference between analyses, reanalysis and predictions (see also L94). I suggest in restricting the discussion to the terminology reanalysis. It is not clear how the accuracy of relative humidity and temperature can be compared ("less accurate").**
All instances, where the word "prediction" was used incorrectly, were removed. We also removed the statement about the lower accuracy of relative humidity and rephrased the sentences to the following:
*"Higher uncertainties with respect to IAGOS observations were found in the re-analysis of relative humidity, which is generally challenging due to the high temporal and spatial variability of WV. Specific issues have been identified in the upper troposphere and lower stratosphere, as well as with the general representation of ice supersaturation."*

**In L68ff, there is an abrupt transition from the simulations of contrail occurrence to "A frequent source of information (…)". What does the latter mean and how are these parts connected? I think ERA5 has no dedicated information about contrails and that should be clarified. Avoid contrail representation and speak e.g. of ability to estimate the contrail formation.**
We followed the suggestions of the Reviewer. All instances, where prediction was used incorrectly, have been changed to "estimation".
The discussion about contrails models and the use of ERA5 data is now better linked. Furthermore, the role of ERA5 is better explained. The following text has been added to the introduction.
*"Offline models such as CoCiP require meteorological data, e.g., temperature and humidity, as input. A well established data set of meteorological data is provided by ERA5 (Hersbach et al., 2020), which stems from a state-of-the-art global modeling system of the European Centre for Medium-Range Weather Forecasts (ECMWF) and a large number of observational data streams. The ERA5 output is based on simulations with a specific, constant version of the Integrated Forecasting System (IFS) of ECMWF. Thus, the ERA5 data set from 1940 to present can provide some insight into trends in the Earth's atmosphere."*

**In the discussion of model biases in ERA5 (LL72ff) absolute and relative humidity biases are mixed. Studies like Bland et al. or Krüger et al. and possibly a few others discuss absolute humidity biases. This cannot lead to a conclusion "no consensus" with studies looking at rHi in ISSRs. I think there is no knowledge about systematic absolute humidity errors in ISSRs. I find it also confusing that throughout the paper you simply talk about a dry bias (L98, L231). It should be made clear that and where you refer to rHi.**
We only partly agree with the Reviewer that we mix the discussion about absolute and relative humidity biases in the introduction. While it is true that the studies compared absolute, specific, and relative humidity, within the the cited studies either absolute, specific, or relative humidity from ERA5 were compared with respective observations. Hence, equal properties were compared. Dry and moist biases were identified in absolute, specific, and relative humidity. Therefore, we argue that there is no real consensus on the

sign of the bias. Furthermore, it is known that the supersaturation adjustment in ERA5 has drawbacks in the representation of ice clouds in the subsaturated and supersaturated state (e.g., ECMWF, IFS Docuementaion – Cy47r3, Part IV, 2021).
Based on the available ERA5 data and the IAGOS observations that are available to us, we identified a slight but systematic rHi dry-bias in ERA5 with respect to IAGOS. While we aimed to bias correct rHi in general, the main focus is on ISSRs and to improve the relative humidity distribution in the post-processed ERA5 data to better identify potential contrail formation regions.

The Reviewer is right that we need to be clearer and have to mention that we refer to a bias in relative humidity. We now explicitly call it relative humidity in the text, where we previously wrote about a moisture bias. Due to the scattering of the changes over the length of the manuscript, the individual corrections are not copied here. Instead we like to direct the Reviewer to the track changes version of the revised manuscript.

**In the description of ERA5 (Sec 2.2), the "native vertical resolution" should be revised. Neither is the spacing between the pressure levels equal to the resolution nor is the pressure level data native data.**
The word "native" is removed and the sentence is rephrased as follows:
*"We make use of the full vertical resolution with a 50-hPa spacing between 350 and 300 hPa, and a 25-hPa spacing between 300 and 150 hPa."*

**In fact, I wonder why the authors did not use the full model level data.**
Previous contrail related studies that used ERA5 data in one way also used pressure level data, for example Agarwal et al. (2022), Wilhelm et al. (2022), and Hofer et al. (2024).
Also from a aircraft perspective it is convenient to use pressure levels as aircraft fly along constant pressure levels.

**Related to that, I cannot follow the justification of the nearest neighbor sampling. RH is not a prognostic variable and I do not understand why T and q cannot be interpolated due to the C-C-Eq. I guess the same is done when interpolating model levels to pressure levels!? I miss a discussion about the treatment of the vertical sampling or interpolation. You talk about levels but sometimes rather mean layers. I guess that a vertical difference of up to 25 hPa, when the vertically nearest grid point is used, may relate to quite a difference in RH as it is known that these ISSRs are shallow and vertical gradients near the TP can be strong. May some of the differences between model and observations be related to that fact? Were sensitivity tests made for different sampling / interpolation strategies? There is no "current version of ERA5" instead ERA5 is, as you mention, using a constant model cycle for creating the reanalysis data set.**
We did not perform a sensitivity study on the sampling method in the vertical nor in the horizontal direction.
In section "ERA5" we added "(horizontally and vertically)" to clarify that we used the closest ERA5 value on the Longitude-Latitude grid as well as the value from the closest pressure level. The respective sentence now reads as follows:
*".. selecting the ERA5 grid points that are temporally and spatially (horizontally and vertically) closest to the IAGOS observations."*

We also removed "current version of ERA5" and the sentence now reads as follows:
*"The ERA5 data set was generated with the ECMWF Integrated Forecasting System (IFS)*

*cycle Cy41r2, which was operational in 2016."*

**In the description of the quantile mapping (2.3) I do not understand the discussion in L265-268. What is the purpose of this? What do you mean with ERA5 is invariant? Do you mean potential biases?**
The presented version of the QM correction relies on a constant bias between model (ERA5) and observation (IAGOS). This means that the bias between ERA5 and IAGOS must be invariant in time. ERA5 is constant in the sense that only one IFS Cycle is used to generate the data set. Thus, the same implementation of dynamics, cloud microphysics, and others are used. However, there may be potential changes due to changes in the observation system that feeds into the 4D-VAR assimilation procedure and those due to a changing climate. Only the latter changes would also be represented in the IAGOS measurements. What may also change over the years are the IAGOS measurements themselves, e.g., through calibration and maintenance procedures, and updated post-processing algorithms.
We have rephrased the text to make this clearer.

*"The presented version of the QM-correction assumes a time-invariant bias between model and observations. On the model side, we assume that the ERA5 data set is constant in time, since it is generated only with the IFS Cycle 41r2 and therefore has the same implementation of the dynamical core, cloud microphysics representation, etc. However there may be some changes in the quantity and quality of observations feeding into the ERA5 data assimilation system. The IAGOS reference observations may also vary over time due to changes in instrument calibration and maintenance procedures."*

**It would be good to have some more information about the training data set and the verification period. What is the strategy behind selecting these time periods? (Why) Is the training and verification period overlapping?**
Several sentences in the section "Quantile mapping" have been rephrased to be more precise in the selection of the training and validation data set.
*"[…] The subscript 'h' commonly refers to historical data, which can also be understood as training data. The training data makes use of IAGOS measurements from January 2018 to June 2021, as this period was considered to be stable in the IAGOS post-processing. [...]"*

Concerning the selected, overlapping periods we added the following sentences :
*"[…] The QM technique is applied to the entire reference period from January 2015 to June 2021, which includes but exceeds the training period. The periods were chosen to a) verify the general applicability of the bi-variate QM-correction method with the same data set and to b) test the stability of the bias correction in years outside the training period."*

**What data period is shown throughout Section 3?**
We have clarified the time period that is analyzed at the beginning of section 3. The first sentence now reads:
*"In a first step, along-track temperature and relative humidity from January 2015 to June 2021 from IAGOS and ERA5 are compared in terms of probability density functions (PDFs), mean values, and mean difference (MD). [...]"*

**Can the QM method only be used to correct at observed locations or also used at other model grid points?**

The model can be used outside the IAGOS flight tracks, but within the study area, which extends from the Eastern US across the North Atlantic to the central EU. We have made this clearer by restructuring and rephrasing the abstract, introduction, and summary. We direct the Reviewer to our responses given below and to the track changes file.

**Sometimes you talk about "grid boxes" (e.g. Sec. 3.2) and I wonder if these are the nearest grid points?**

The manuscript was screened for the use of the word "grid-box(es)" and, if incorrectly used, the word grid-box(es) was replaced with the word "data points" or an equivalent word. We direct the Reviewer to our responses given below and to the track changes version of the revised manuscript.

**I think I understand the purpose of analyzing differences of cloudy and cloud-free condition in Sec. 3.2. However, the reader would profit from a few clarifications. Is it correct that you compare PDFs from completely different data sets? How does a comparison of data points look where ERA5 and IAGOS both have no cloud simulated/observed? To what extent are results influenced by situation of cloud simulated but not observed and vice versa? In addition, I wonder about the use of cloud-coverage? Wouldn't ice water content be more straightforward to compare cloudy situations.**

The Reviewer is partially correct in the sense that we compared different data sets, whereas we would prefer to say that we compared different data points from the same data set.

However, even by requiring cloud-free, intermediate, or cloudy conditions to be present at the same time does not mean that same data is compared as cloud conditions derived from IAGOS and ERA5 do not have to agree in space and time.

However, we have followed the Reviewer's suggestion and repeated the analysis so that only data points are compared where in-cloud and cloud-free conditions were determined in ERA5 and based on the IAGOS measurements. For comparison, we first provide the plot of the previous version of the manuscript:

[Figure]

and below the updated version with the new analysis:

[Figure]

The distributions for the cloud-free conditions (panel a) remain largely unchanged compared to the plot in the previous version of the manuscript. Differences appear for the intermediate (b) and the cloudy conditions (c). In panel (b), the broad distribution from IAGOS becomes narrower and more similar to the distributions of the original and QM-corrected ERA5 data. While the medians of the original and the QM-corrected ERA5 data remain unchanged compared to the plot in the last version of the manuscript, the median from IAGOS is slightly shifted to lower values by about 3%. In panel (c), the distributions of the original and corrected ERA5 data remain unchanged compared to the plot in the last version of the manuscript. Similar to panel (b), the distribution of IAGOS rHi becomes narrower. However, the IAGOS distribution is still much broader compared to both ERA5 distributions, indicating that neither the original ERA5 nor the QM-corrected ERA5 can represent the natural variability in rHi observed by IAGOS. The medians of the original and QM-corrected ERA5 data are slightly shifted to lower values of rHi shown in the plot in the previous version of the manuscript.

We decided to use the cloud fraction as a criterion, since IAGOS does not provide the ice water content. Furthermore, the saturation adjustment is applied depending on the presence or absence of a cloud. We selected a cloud fraction value threshold of 0.2 to determine cloud-free samples, assuming that these have a high probability of being cloud-free. Similarly, we chose 0.8 to select points with a high probability of cloud cover.

**Minor comments:**
**L82: The sentence "It is noted (…)" should be revised. I guess you don't want to say that the observations have a bias, right?**
We rephrased the sentence to be more specific and to say that the spatial sampling might be biased and not necessarily the measurements itself.
*"It is noted that in-situ aircraft observations are potentially biased in terms of spatial sampling because aircraft typically avoid deep convective clouds and the outflow of such clouds."*

**L229: Please revise "problematic for contrail and cirrus representation". Contrails**

**are not at all represented in ERA5. What is the problem for Cirrus?**

The sentence has been clarified and rephrased to:

*"While the adjustment approach proved to be suitable for most atmospheric conditions (Gierens et al., 1999; Tompkins et al., 2007; Lamquin et al., 2009). However, the use of ERA5 relative humidity data, which is subject of the adjustment in the context of contrail formation analysis, leads to an underestimation of ISSR in the upper troposphere (Gierens et al., 2020)."*

**L396 "revels" should be "reveals"?**

The typo has been corrected.

**L410 What means "smaller"?**

The word has been replaced by "narrower", which is the correct term to describe the width of a distribution. The sentence now reads as following:

*"All three distributions of $r_{ice}$ are narrower compared to the cloud-free or intermediate conditions, with $r_{P1,ice}$ being broadest."*

**L495: This sentence makes no sense.**

The sentence has been rephrased to the following:

*"A confusion matrix is a table that is used to visualize the classification performance of an algorithm. Table 4 provides a schematic for a binary event. "*

**L577-582: Be more specific about the identified RH biases, especially, whether it is a cold/warm or moist/dry bias. Avoid redundant information.**

The Reviewer is correct and we now specify the bias. Redundant information was removed. Due to the length of the section, we would like to direct the Reviewer to the track changes version of the revised manuscript.

Agarwal, A. / Meijer, V. R. / Eastham, S. D. / Speth, R. L. / Barrett, S. R. H., Reanalysis-driven simulations may overestimate persistent contrail formation by 100%–250% 2022-01, Environ. Res. Lett. , Vol. 17, No. 1, IOP Publishing, p. 014045

ECMWF, IFS Documentation CY47R3 - Part IV Physical processes, 2021, IFS Documentation CY47R3 , No. 4, ECMWF

Hofer, S. / Gierens, K. / Rohs, S., How well can persistent contrails be predicted? An update, 2024, Atmos. Chem. Phys. , Vol. 24, No. 13, p. 7911-7925

Wilhelm, L. / Gierens, K. / Rohs, S., Meteorological conditions that promote persistent contrails, 2022, Applied Sciences , Vol. 12, No. 9